# ADDQ: Adaptive Distributional Double $Q$-Learning

**Leif Döring** [1]  **Benedikt Wille** [1]  **Maximilian Birr** [1]  **Mihail Bîrsan** [2]  **Martin Slowik** [1]

## Abstract

Bias problems in the estimation of $Q$-values are a well-known obstacle that slows down convergence of $Q$-learning and actor-critic methods. One of the reasons of the success of modern RL algorithms is partially a direct or indirect overestimation reduction mechanism. We propose an easy to implement method built on top of distributional reinforcement learning (DRL) algorithms to deal with the overestimation in a locally adaptive way. Our framework is simple to implement, existing distributional algorithms can be improved with a few lines of code. We provide theoretical evidence and use double $Q$-learning to show how to include locally adaptive overestimation control in existing algorithms. Experiments are provided for tabular, Atari, and MuJoCo environments.

## 1. Introduction

A fundamental building block of many modern reinforcement learning (RL) algorithms is Watkins' $Q$-learning (QL) (Watkins & Dayan, 1992). In each round the agent observes a new reward signal and updates the currently estimated state-action function by combining the new reward signal with the best currently estimated action in the next step. Unfortunately, the update rule involves a maximum and maxima suffer both from overestimation bias and function approximation uncertainty. Thus, estimated $Q$ values are initially way too large. Although convergence of $Q$-learning and its variants for tabular cases can be proved rigorously, the convergence can often be seen only after millions of iterations. In the context of $Q$-learning we refer to the seminal paper (Thrun & Schwartz, 1993). The overestimation effect is harmful not only for simple QL and variants with function approximation such as DQN (Mnih et al., 2015), but also for critic estimation in actor-critic methods such as

soft actor-critic (SAC) of (Haarnoja et al., 2018). Motivated by statistical approaches to the estimation of the expectation of maxima of random variables the concept of double $Q$-learning (DQL) was introduced in (van Hasselt, 2010). Instead of using one set of random variables two independent sets are used. One is used to detect the maximal index, the other to evaluate the random variable corresponding to the maximal index. For $Q$-learning this translates to keeping track of two copies of the $Q$-matrix that are alternated in order to detect the best action and evaluate the corresponding $Q$-value. DQL and its actor-critic variants reduce the overestimation (see for instance Figure 2 in (Fujimoto et al., 2018)) and sometimes even underestimate $Q$-values. This, for example, can be seen in a simple chain MDP (Example 6.7 in (Sutton & Barto, 2018) and also (Lan et al., 2020) for the underestimation effect in the same example). (Fujimoto et al., 2018) argue that overestimation should be addressed particularly in state-action regions with high uncertainty. Indirectly, this is taken into account by ensemble methods such as Maxmin QL (Lan et al., 2020). These methods use ensembles of more than two $Q$-estimators. Different ensemble estimators take the minimum, full ensemble averages (Anschel et al., 2017), or random ensemble averages (Chen et al., 2021). A related line of research uses uncertainty-based RL, also with the goal of reducing overestimation, see for instance (Wu et al., 2021), (Ghasemipour et al., 2022).

There is no rule whether QL, DQL, or any ensemble variant works well for a given environment. Algorithms sometimes perform well and sometimes fail. This article proposes a novel approach. We propose to take QL and (at least) one other method and try to control if the QL update should be replaced or mixed with another underestimating method. The control must be locally adaptive, and the need to manage the bias depends on the local uncertainty (aleatoric and epistemic randomness, function approximation). For an approach similar in spirit to ours, see (Dorka et al., 2021).

- We show theoretically how distributional RL helps the agent identify the need for overestimation control.

- As a test case, we combine QL and DQL using a local weighting that we call ADDQ.

- Convergence of ADDQ is proved, experiments are performed on tabular, Atari, and MuJoCo environments.

[1]Institute of Mathematics, University of Mannheim, Germany [2]Department of Mathematics and Computer Science, Freie Universität Berlin, Germany. Correspondence to: Leif Döring <doering@uni-mannheim.de>.

*Proceedings of the 42nd International Conference on Machine Learning*, Vancouver, Canada. PMLR 267, 2025. Copyright 2025 by the author(s).

## 2. $Q$-learning and the overestimation problem

### 2.1. Tabular $Q$-learning

Let us fix a (discrete) Markov decision model $(\mathcal{S}, \mathcal{A}, \mathcal{R}, p)$, where $\mathcal{S}$ is a finite state-space, $\mathcal{A}$ a finite space of allowed actions, $\mathcal{R}$ the reward space, and $p$ a transition kernel describing the distribution of the reward $r$ and the new state $s'$ when action $a$ is played in state $s$. Given a time-stationary policy $\pi$, a Markov kernel on $\mathcal{S} \times \mathcal{A}$, there is a Markov reward process $(S_t, A_t, R_t)$ with transitions

$$\mathbb{P}^\pi(R_t = r, S_{t+1} = s', A_{t+1} = a' | S_t = s, A_t = a)$$
$$= \pi(a' : s')p(r, s' : s, a).$$

The goal of the agent in reinforcement learning is to use rollouts of the MDP to find a policy that maximizes $Q^\pi(s, a) = \mathbb{E}^\pi[\sum_{t=0}^\infty \gamma^t R_t | S_0 = s, A_0 = a]$, the expected discounted reward. The discounting factor $\gamma \in (0, 1)$ is fixed. In the discrete setting with $\mathcal{S}$ and $\mathcal{A}$ finite it is well-known that optimal stationary policies exist and can be found as greedy policy obtained by the unique solution matrix $Q^*$ to $T^*Q = Q$. The non-linear operator $(T^*Q)(s, a) = r(s, a) + \sum_{s' \in \mathcal{S}} p(\mathcal{R} \times \{s'\} : s, a)\gamma \max_{a' \in \mathcal{A}} Q(s, a')$ is called Bellman's optimality operator. Bellman's optimality operator is a max-norm contraction on the $\mathcal{S} \times \mathcal{A}$ matrices. Using Banach's fixed point theorem, the solution can in principle be found by iteratively applying $T^*$ to some initial matrix $Q_0$. The drawback of this approach is the need to know the operator $T^*$, thus having knowledge about the transitions $p$. Using standard stochastic approximation algorithms the fixed point $Q^*$ can be approximated by

$$Q(s, a) \leftarrow (1 - \alpha)Q(s, a) + \alpha(r + \gamma \max_{a'} Q(s', a')),$$

called $Q$-learning (QL). The state-action pairs can be chosen synchronously or asynchronously using rollouts. Typically, to update at $(s, a)$ a one-step sample $s', r$ is obtained from $p(\cdot : s, a)$ and the step-sizes $\alpha$ are assumed to satisfy the Robbins-Monro conditions. The exploration (choices of $(s, a)$ to be updated) can be on-policy (using the $Q$-estimates) or off-policy (using a behavior policy). The only requirement is infinite visits for all state-action pairs. The recursively defined matrix-sequence $(Q_t)$ was proved to converge in the tabular setting, see e.g. (Tsitsiklis, 1994).

### 2.2. The overestimation problem

Even though QL converges to $Q^*$ for the number of updates going to infinity, the convergence is very slow. One of the known sources is the so-called overestimation problem of QL. The algorithm does not provide unbiased estimates of $Q^*(s, a)$. Instead, the estimates $Q_t(s, a)$ tend to overestimate $Q^*(s, a)$. A statistical explanation is based on the simple fact that the point estimator $\max\{\hat{X}_1, ..., \hat{X}_n\}$ is not an unbiased estimator of $\max\{\mathbb{E}[X_1], ..., \mathbb{E}[X_n]\}$ but

the estimator is positively biased. Thus, the update targets $r + \gamma \max_{a'} Q_t(s', a')$ can be seen as overestimating the true Bellman optimality operator at each step. The consequences of overestimation are less obvious than it seems at first sight. Shifting $Q$-values the same amount globally has no effect, neither for $Q$-based exploration purposes nor for best action selection. It is the local difference in overestimation that must be avoided to not confuse the agent. Thus, it is crucial to understand the root causes of overestimation to then mitigate by taking alternative updates to QL if needed.

There is little quantitative understanding of the overestimation; for some rough bounds see (van Hasselt, 2011). We used tools from probability theory to compute estimation bounds for a simple explanatory example. The example is

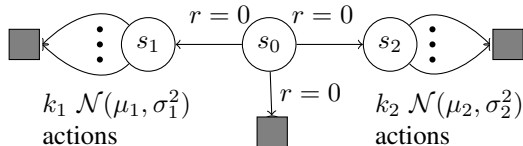

Figure 1. Two-sided bandit MDP, start in $s_0$, gray boxes terminal

intriguing, as it is simple but hard to learn - even more so if one side is replaced by a chain of decisions. Each side gives a reward (for simplicity 0) followed by a Gaussian reward from one of $k$ actions. QL and DQL can both fail badly (each on one side) for the same parameter configuration. If $\mu_1 > 0 > \mu_2$, then the optimal action in $s_0$ is "left". If $\sigma_2$ (and/or $k_2$) is large compared to $\sigma_1$ (and/or $k_1$) then the overestimated $Q$-values of QL will confuse the agent and lead him to believe that "right" is optimal. Similarly, underestimating can make the agent believe "down" is optimal. Our approach of learning locally to use QL or DQL (or another variant) can mitigate that problem by learning to use QL for one side and DQL for the other. The method is motivated by two theoretical results, a lower bound on overestimation (Proposition 2.1) and a computation with DRL to connect estimated sample variances and overestimation (Proposition 2.2). If step-sizes are chosen in the common way as $\alpha_t(s, a) = \frac{1}{T_{s,a}(t)}$, with $T_{s,a}(t)$ the number of visits at $(s, a)$ up to time $t$, then results on sums and maxima of Gaussian random variables can be used to prove a lower bound on the expected overestimation of the true value $\gamma\mu$.

**Proposition 2.1.** *If the left side has been explored $Nk_1$ times and the exploration was sufficiently exploratory (see Theorem A.2), then the $Q$-estimate at $(s_0, "left")$ has bias at least $\frac{\gamma}{\sqrt{\pi \log(2)}} \frac{\sigma_1 \sqrt{\log(k_1)}}{\sqrt{N}}$ and analogously at $(s_0, "right")$.*

The lower bound quantifies the idea that uncertainty forces overestimation and the bias only decreases slowly over time. A proof is given in Appendix A. Since the estimate is relatively tight one gets a feeling for how many reward samples

are needed to get sufficiently precise estimates of the $Q$-values so the agent makes the right decision.

There has been considerable interest for the past years to understand the sources of uncertainty in RL. While many sources of uncertainty exist, they are often categorized as either aleatoric or epistemic. Aleatoric uncertainty is model given and cannot be improved by more data or learning. In RL this is uncertainty implied by random variables governing rewards and transitions. Epistemic uncertainty refers to the uncertainty that could potentially be reduced using more data and better algorithms (including better function approximation). Epistemic uncertainty in RL is induced by all random variables to run the learning procedure (exploration, replay buffer, etc.) and function approximation in deep RL. Keeping in mind the different sources of uncertainty is useful in order to identify algorithmic potential for improvement but also theoretical limitations. It is also important to realize that aleatoric and epistemic randomness strongly influence each other. If a reward has large variance (aleatoric uncertainty) then the estimation with samples creates more epistemic uncertainty. For estimating expectations, this is due to the central limit theorem.

In deep RL one of the major sources of epistemic uncertainty is function approximation. As in (Thrun & Schwartz, 1993) we could add to our analysis independent error-noise modeling the function approximation. If the error noise is assumed Gaussian then our results readily extend, by adding additional noise-variance to our results. Since the modeling assumption as independent noise is rather special we restrain from studying the effect of function approximation on epistemic uncertainty. Instead, we will now show how to use additional information from distributional QL to deal with overestimation in a local way.

### 2.3. Tabular distributional $Q$-learning

To make this article as self-contained as possible we give a minimal overview on DRL. For a concise treatment we refer to the recent book (Bellemare et al., 2023). Given a Markov decision model and a stationary policy $\pi$, (Rowland et al., 2018) define the return distribution function as

$$\eta^\pi(s,a)(B) := \mathbb{P}^\pi\Big(\sum_{t=0}^\infty \gamma^t R_t \in B \Big| S_0 = s, A_0 = a\Big)$$

for $B \in \mathcal{B}(\mathbb{R})$. The expectation over the measure $\eta^\pi$ is the classical state-action value function $Q^\pi$. In contrast to ordinary RL, the target in DRL is to learn return distributions instead of only return expectations. There have been a lot of theoretical articles on DRL (Bellemare et al., 2017; Dabney et al., 2018; Rowland et al., 2018; Lyle et al., 2019; Bellemare et al., 2023; Rowland et al., 2023b;a) establishing distributional Bellman operators, contractivity, convergence proofs of dynamic programming,

etc. It was shown in (Bellemare et al., 2017; 2023) that the return distribution function is the unique solution to $\eta^\pi = T^\pi \eta^\pi$, where $T^\pi : \mathcal{P}(\mathbb{R})^{\mathcal{S} \times \mathcal{A}} \to \mathcal{P}(\mathbb{R})^{\mathcal{S} \times \mathcal{A}}$ is the distributional Bellman operator defined as $(T^\pi \eta)(s,a) = \sum_{r,s',a' \in \mathcal{R} \times \mathcal{S} \times \mathcal{A}} b_{r,\gamma} \# \eta(s',a') \pi(a';s') p(s',r;s,a)$ with bootstrap function $b_{r,\gamma}(z) = r + \gamma z$ and push-forward of measures $f \# \nu(B) := \nu(f^{-1}(B))$. In essence, sample based dynamic programming can also be carried out in a distributional sense replacing the classical Bellman operator by its analogue in the distributional sense. Distributional QL proceeds similarly to classical expectation QL:

$$\eta(s,a) \leftarrow (1-\alpha)\eta(s,a) + \alpha\big(b_{r,\gamma} \# \eta(s',a^*)\big),$$

with $a^* = \operatorname{argmax}_{a'} Q(s',a')$, where $Q(s',a')$ are the expectations of the probability measures $\eta(s',a')$. In order to work algorithmically with DRL parametrizations $\mathcal{F}$ of measures need to be used. Distributional learning algorithms in practice then work similarly to deep learning algorithms, alternating Bellman operators and function class projection, see Algorithm 1. There are two simple

---

**Algorithm 1** Distributional $Q$-learning update step

---
**Require:** Proxy $\eta$ for $\eta^*$ and pair $(s,a)$ to be updated
    Determine step-size $\alpha$
    Sample reward/next state $(r,s')$
    # Compute target
    $a^* \leftarrow \operatorname{argmax}_a \mathbb{E}_{Z \sim \eta(s',a)}[Z]$
    $\hat{\eta}_* \leftarrow b_{r,\gamma} \# \eta(s',a^*)$
    # Project target back onto support
    $\hat{\eta} \leftarrow \Pi_{\mathcal{F}}(\hat{\eta}_*)$
    # Move $\eta(s,a)$ towards the target, for tabular RL e.g.
    $\eta(s,a) \leftarrow (1-\alpha)\eta(s,a) + \alpha\hat{\eta}$

---

parametrization that have been used frequently. The categorical parametrization (fixing number of atoms with variable weights at fixed locations) and the quantile parametrization (fixing number of atoms with fixed weights but variable locations). For the categorical parametrization a set of $m$ evenly spaced locations $\theta_1 < \cdots < \theta_m$ needs to be fixed, the categorical measures are then defined by $\mathcal{F}_{C,m} = \big\{ \sum_{i=1}^m p_i \delta_{\theta_i} \,\big|\, p_i \geq 0, \sum_{i=1}^m p_i = 1 \big\}$. That is, measures in $\mathcal{F}_{C,m}$ are parametrized by an $m$dim probability vector with weights for the $m$ fixed atoms. In contrast, the quantile parametrization $\mathcal{F}_{Q,m} = \big\{ \sum_{i=1}^m \frac{1}{m} \delta_{\theta_i} : \theta_i \in \mathbb{R} \big\}$ fixes the weights to $\frac{1}{m}$ with variable atom locations.

### 2.4. Overestimation mitigation using distributional RL

We follow an insight from Proposition 2.1 that large uncertainty, aleatoric (large $\sigma$) as well as epistemic (small $N$ - additional $\sigma$ if function approximation is modeled as Gaussian error), implies large overestimation of $Q$-values. While we skip function approximation for theoretical considerations we are as precise as possible in the simplest tabular

settings. The estimate from Proposition 2.1 suggests to replace QL-updates in $(s, a)$ if uncertainty is large (compared to other actions). Unfortunately, in standard QL the agent has no direct access to such information in order to adjust the update rule at $(s, a)$. This is where our idea comes into play. DRL gives the agent exactly the needed information using distribution insight into the return estimate $\eta_t(s, a)$. It is crucial to note that DRL learns random measures as $\eta(s, a)$ is a probability measure that depends on the random samples used in the updates so it has an expectation and a variance that are both random in terms of the random samples. The situation becomes tricky as we are going to take variances of the variances. To avoid confusion an analogy to statistics is used. We speak of sample averages $M$ (resp. sample variances $S^2$) of $\eta(s, a)$ and expectation $\mathbb{E}$ (resp. variance $\mathbb{V}$) for the integrals against the randomness induced by the probability space behind all random variables.

We now use the bandit MDP from above to explain why the DRL agent has access to uncertainty estimates during the learning process. To allow concrete computations with distributions, we use a particularly simple QL update mechanism (the one also used for estimates in (van Hasselt, 2011)). First explore all actions $N$ times, then propagate to $(s_0, "left")$. In fact, this is nothing but distributional QL with cyclic exploration and target-matrix trick (Mnih et al., 2015) as we explain in Appendix A. The obtained estimate of $\eta^*(s_0, "left")$ will be denoted by $\hat{\eta}(s_0, "left")$.

**Proposition 2.2.** *The sample variance of $\hat{\eta}(s_0, "left")$, analogously for "right", after $Nk_1$ steps is $\frac{\sigma_1^2}{N-1}\chi_{N-1}^2$-distributed. The expectation is $\sigma_1^2$, the variance is $\frac{2\sigma_1^4}{N-1}$.*

A proof is given in Appendix A. It is surprising that the sample variance distribution can be identified explicitly as chi-squared, unlike the sample expectations (maxima of independent Gaussians). This is a consequence of the well-known fact in statistics that sample variances of Gaussians are independent of sample expectations. Thus, if return estimates are Gaussian the max-operation of QL (with respect to sample averages) is only delicate for sample expectations, not for sample variances. We emphasize that in contrast to QL the agent in distributional QL does have access to the $\frac{\sigma^2\chi_N^2}{N}$-distributed sample variance by computing sums of the atoms! Hence, the agent can make use of an unbiased estimate for the aleatoric uncertainty $\sigma^2$ which is conflicted by epistemic uncertainty that decreases with $N$.

**Implications from of our theoeretical considerations:** We propose the following locally adaptive overestimation mitigation method. (i) Use distributional QL. (ii) At every update compute the sample variance of the current return estimate $\eta_t(s, a)$. (iii) If the sample variance is large replace the QL update by another update (e.g. DQL or an ensemble

update or a mixture). Since "large" has no absolute meaning in RL we will compare variances among all actions in $s$ and reduce according to the relative sample variance.

**Exploration and overestimation control with ensembles and double $Q$-learning:** Known algorithms that directly try to better estimate the $Q$-values require to set up a particular algorithmic architecture. Most use an ensemble of $Q$-copies which are then combined by taking minima (Lan et al., 2020), averages (Peer et al., 2021), or averages with random choices (Chen et al., 2021). Ensemble methods are promising in theory (assuming independent ensembles) but more problematic for deep RL as storage problems force ensembles to be parametrized by the same neural network. The optimal number of copies (a hyperparameter) varies for different environments, some choices work well, others fail.

Our approach is different. We suggest to use QL when it works well (small sample variance) and another algorithm where QL fails (large sample variance). We could combine QL with ensemble methods but for this article decided to combine QL with DQL a bit in the spirit of weighted DQL (Zhang et al., 2017) but with completely different weights - we compare to weighted DQL in Appendix C.2. For DQL (van Hasselt, 2010) two copies $Q^A$ and $Q^B$ are stored. The update mechanism is similar to QL where the matrix to be updated is chosen randomly in every step. The main difference is the target used, matrices $Q^A$ and $Q^B$ are flipped:

$$Q^{A/B}(s, a)$$
$$\leftarrow (1-\alpha)Q^{A/B}(s, a) + \alpha\big(r + \gamma Q^{B/A}(s', z^*)\big),$$

with $z^* = \max_{a'} Q^{A/B}(s', a')$. Here and in the following we use $Q^{A/B}$ to allow either the choice of $Q^A$ or $Q^B$. Double $Q$-learning reduces the overestimation strongly but sometimes leads to severe underestimation.

The use of sample variance of estimated return distributions is not new to RL. For bandits the simplest example is the UCB exploration bonus for unknown variances that uses this for exploration. In the context of exploration in RL uncertainty dependent exploration has been applied using distributional RL (see for instance (Mavrin et al., 2019), (Moerland et al., 2018)). The use of sample variances in distributional RL to locally mitigate the overestimation is new to the best of our knowledge. In order to not mix up effects we stick to the standard exploration choices.

## 3. ADDQ: Tabular setting

Based on the theoretical insight we will now introduce a concrete method. We use DQL-updates as an alternative to QL-updates when the agent expects QL-updates to be harmful (large estimated uncertainty by means of sample variance). The weighted DQL-approach we suggest is readily implemented into existing DRL implementations.

## 3.1. Weighted DQL: From SARSA-trick to ADDQ

To derive the algorithm let us recall the SARSA convergence proof of (Singh et al., 2000) that was also used to prove convergence for DQL (van Hasselt, 2010) and variants such as clipped $Q$-learning (Fujimoto et al., 2018) or Maxmin (Lan et al., 2020). The idea is to add a clever 0, adding and subtracting what is missing to the QL update, thus, writing the algorithm as QL with a bias. If the bias can be proved to disappear, a comparison to QL implies convergence to $Q^*$:

$$
\overbrace{Q^{A/B}(s,a) \leftarrow (1-\alpha)Q^{A/B}(s,a) + \alpha\big(r + \gamma Q^{A/B}(s', z^*)\big)}^{Q\text{-learning update}}
$$
$$
\underbrace{+ \alpha(\gamma Q^{B/A}(s', z^*) - \gamma Q^{A/B}(s', z^*))}_{=:b^{A/B}(s,a)},
$$

with $z^* = \arg\max Q^{A/B}(s, a)$. Taking this point of view there are plenty of possibilities to modify DQL to achieve more or less over-/underestimation. As an example, choosing the negative bias-terms $b_{\text{clip}}^{A/B} = \alpha \min\{\gamma Q^{B/A}(s', z^*) - \gamma Q^{A/B}(s', z^*), 0\}$ yields so-called clipped $Q$-learning introduced as part of TD3 in (Fujimoto et al., 2018). As motivated in Section 2.4 we suggest a locally adaptive overestimation control. The main insight is as follows. One can locally interpolate between QL and DQL by multiplying bias terms with local adaptive weights:

$$
\bar{b}^{A/B}(s,a) := \underbrace{\beta^{A/B}(s,a)}_{\text{new}} b^{A/B}(s,a).
$$

Replacing bias terms $b$ by $\bar{b}$ generalizes the update. The aggressive choice $\beta = 1$ results in QL updates (overestimation), $\beta = 0$ results in DQL updates (tendency of underestimation). Choices of $\beta$ suggested in the present article are motivated by the propositions of Sections 2.2 and 2.4. If a lot of uncertainty is present, the algorithm uses large $\beta$, otherwise small $\beta$. The aggressive choices are not necessarily the best, in our experiments below softer choices were more effective. Since overestimation is a priori not problematic for the learning process (if all estimates are equally overestimated the best action does not change, only skewed overestimation slows down the learning) we suggest a choice of $\beta$ that takes into account the local structure, normalizing variances over possible actions.

## 3.2. New algorithm: locally adaptive distributional double $Q$-learning

Following the ideas above we introduce ADDQ, integrating locally adaptive overestimation mitigation in DQL. The pseudo-code given in Algorithm 2 extends distributional RL pseudo-code to include the double algorithm and adaptive weights. For the tabular target update we follow the measure-mixture approach of (Rowland et al., 2018), Section 8. Changes to the code for other target updates (for

instance gradient steps to minimize KL loss) are straight forward. The algorithm is seen to be a combination of distri-

---

**Algorithm 2** ADDQ update step

---
**Require:** Proxies $\eta^A$, $\eta^B$ for $\eta^*$, pair $(s, a)$ to be updated
    Determine step-size $\alpha$
    Sample reward/next state $(r, s')$
    Randomly choose Update(A) or Update(B)
    **if** Update(A) **then**
        # Compute target with locally adapted weight
        $a^* \leftarrow \arg\max_a \mathbb{E}_{Z \sim \eta^A(s', a)}[Z]$
        Determine weight $\beta \in [0, 1]$ based on $\eta_{old}^A, \eta_{old}^B$
        $\nu \leftarrow (1-\beta)\eta^B(s', a^*) + \beta\eta^A(s', a^*)$
        $\hat{\eta}_*^A \leftarrow b_{r,\gamma}\#\nu$
        # Project target back onto support
        $\hat{\eta}^A \leftarrow \Pi_{\mathcal{F}}(\hat{\eta}_*^A)$
        # Move $\eta(s, a)$ towards the target, for tabular RL e.g.
        $\eta^A(s, a) \leftarrow (1-\alpha)\eta^A(s, a) + \alpha\hat{\eta}^A$
    **end if**
    **if** Update(B) **then**
        Proceed analogously with $A$ and $B$ exchanged
    **end if**

---

butional QL and distributional DQL. Keeping $\beta$ constant to 1 is distributional QL, constant 0 distributional DQL. The key is to chose $\beta$ dependent on the uncertainty that drives the skewed QL overestimation for different actions. Extending arguments from the literature, notably the convergence proof of (Rowland et al., 2018) for categorical $Q$-learning with stochastic approximation target upate and the SARSA trick of (Singh et al., 2000), we prove convergence of Algorithm 2 for categorical measure parametrizations:

**Theorem 3.1.** *Given some initial return distribution functions $\eta_0^A, \eta_0^B$ supported within $[\theta_1, \theta_m]$. If*

- *rewards are bounded in $[R_{min}, R_{max}]$ and it holds $[\frac{R_{min}}{1-\gamma}, \frac{R_{max}}{1-\gamma}] \subseteq [\theta_1, \theta_m]$,*

- *step-sizes fulfill the Robbins-Monro conditions and $\eta^A$ or $\eta^B$ are updated randomly,*

- *the sequences $(\beta_t^A)_{t \in \mathbb{N}}, (\beta_t^B)_{t \in \mathbb{N}}$ only depend on the past and fulfill $\lim_{t \to \infty} |\beta_t^A - \beta_t^B| = 0$ almost surely,*

*then the induced $Q$-values converge almost surely towards $Q^*$. If additionally the MDP has a unique optimal policy $\pi^*$, then $(\eta_t^A), (\eta_t^B)$ converge almost surely in $\bar{\ell}_2$ to some $\eta_C^* \in \mathcal{F}_{C,m}$ and the greedy policy with respect to $\eta_C^*$ is $\pi^*$.*

According to Theorem 3.1 symmetric sequences $\beta^A = \beta^B$ that can depend on past distributions $\eta^A, \eta^B$ yield convergence. A particularly simple choice compares the deviations in $\eta^{A/B}$ locally as a local measure for uncertainty.

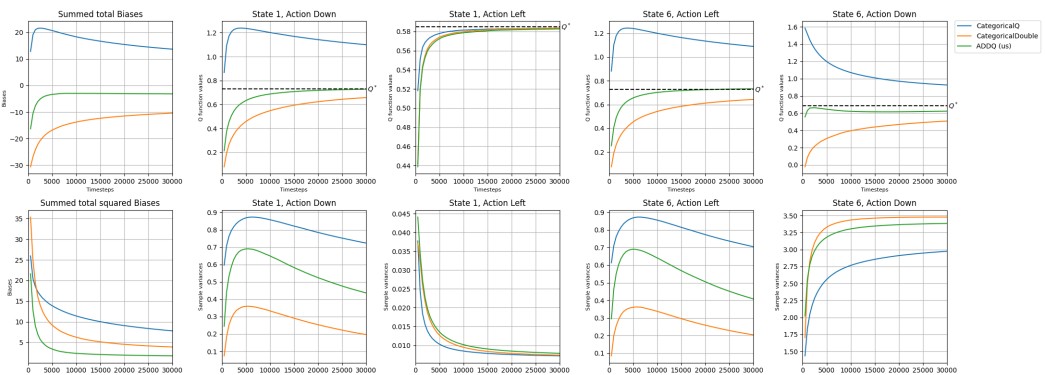

*Figure 2.* Grid world. First column: Biases summed over all state-action pairs. Other columns highlight the relation of sample variance (bottom) and the effect on $Q$-value (top) estimation. For more experiments and comparisons to Maxmin, EBQL, REDQ see Appendix C.

**An exemplary choice of adaptive $\beta$:** As motivated earlier QL suffers from overestimation in the presence of uncertainty (function approximation, aleatoric randomness, epistemic randomness) which is reflected in the sample variance (or other measures of the spread of the distribution) of the discrete (random) distributions $\eta_t^{A/B}$. As uniform overestimation is not particularly troubling (e.g. adding a constant to all $Q$-values does not harm at all) the learning process is slowed down by differences in sample variances for allowed actions. There are many other possibilities to choose $\beta$ but we decided to fix this example for all our experiments. For a finite atomic measures $\nu = \sum_{i=1}^{n} p_i \delta_{a_i}$ define the sample mean $M(\nu) = \frac{1}{n}\sum_{i=1}^{n} a_i p_i$ and the sample variance $S^2(\nu) = \frac{1}{n-1}\sum_{i=1}^{n} p_i(a_i - M(\nu))^2$. Now define

$$S_{s,a}^2 := \frac{1}{2}\big(S^2(\eta_t^A(s,a) + S^2(\eta_t^B(s,a))\big),$$

$$S_s^2 := \frac{1}{|\mathcal{A}|}\sum_{a \in \mathcal{A}} S_{s,a}^2, \quad \text{and} \quad S_{rel}^2(s,a) := \frac{S_{s,a}^2}{S_s^2}.$$

According to our computations in the two-sided bandit model, evenly distributed relative sample variances (i.e. all values around 1) correspond to balanced overestimation effects. Otherwise, overestimation is unbalanced. We define locally adaptive weights for the next update at $(s,a)$:

$$\beta := \begin{cases} 0.75 & : S_{rel}^2(s,a) < 0.75 \\ 0.5 & : S_{rel}^2(s,a) \in [0.75, 1.25] \\ 0.25 & : S_{rel}^2(s,a) > 1.25 \end{cases} \tag{1}$$

For algorithmic simplicity one could also use squared deviations to the median instead of the mean as the median can be red off directly for measures in the $\mathcal{F}_{C,m}$ and $\mathcal{F}_{Q,m}$. The choice combines Q and DQ rather softly, more aggressive choices reduce the length of the interval and increase (resp. decrease) towards $\beta = 1$ and $\beta = 0$. In practice, (tabular, all Atari, all MuJoCo) the choice worked well without any tuning. The thresholds in the definition of $\beta$ can be seen as hyperparameters to the algorithm. We leave the development of adaptive thresholds for future research.

### 3.3. A grid world example

To show the advantages of ADDQ we present a grid world that highlights, in a more complicated and more realistic way, the main effects of the bandit MDP. For more details, see Appendix C. In particular, in Appendix C.3 we compare ADDQ to other algorithms for overestimation reduction.

Deep RL experiments presented below use essentially deterministic environments with uncertainty mainly from function approximations. We thus provide a grid world with complicated stochasticity that, depending on parameters, is hard for QL and DQL. The high stochasticity region with low rewards confuses the QL agent. The agent strongly overestimates suboptimal $Q$-values that lead to the gray area, in the gray area $Q$-values are overestimated for actions that stay in the gray area. Hence, all $\varepsilon$-greedy exploration mechanisms spend much time in the gray area. On the other hand, DQL is motivated by estimators of iid random variables and underestimates strongly if action-values for different actions are unevenly distributed. Thus, the DQL agent gets confused by the local deviations caused by the fake goal and the stochastic region. What results in overestimation for QL, results in underestimation for DQL. In contrast, ADDQ locally combines update-rules of QL and DQL to reduce the estimation biases a lot. In Appendix C we show experimentally that the choice of thresholds in $\beta$ is relatively harmless, more or less aggressive updates still perform well. In contrast, constant $\beta$ and the non-distributional choice from (Zhang et al., 2017) with $c = 10$ have larger biases.

| F | 1 | 2 | S |
|---|---|---|---|
| 4 | 5 | 6 | 7 |
| 8 | 9 | 10 | 11 |
| 12 | G | 14 | 15 |

*Figure 3.* Start in S, goal in G, fake goal in F, high stochasticity and low reward area in gray

Plots in the figure above show a few key insights, more in Appendix C. First, the estimation bias of ADDQ is much smaller than those of QL and DQL. Most importantly in the complicated states 1, 4, 6, 7. The reason ADDQ better estimates the $Q$-values is the adaptive choice to prefer QL or DQL, depending on (relatively) large variances.

## 3.4. ADDQ adaptation for distributional DQN

Our DRL based local adaptive overestimation mitigation can be integrated into existing code for DRL with a few extra lines. In the following we describe the integration into C51 (Bellemare et al., 2017). We run experiments on Atari environments from the Arcade Learning Environment (Bellemare et al., 2013) using the Gymnasium API (Towers et al., 2023). Algorithms are few line modifications based on the RL Baselines3 Zoo (Raffin, 2020) training framework, without any further tuning of hyperparameters.

The C51 algorithm obtained its name from using a categorical representation of return distributions with $m = 51$ atoms. The weights of the parametrization are parametrized via feedforward neural networks following the DQN architecture (Mnih et al., 2015). The state $s$ serves as input and the last layer outputs $m = 51$ logits for each action followed by a softmax to return probability weights. We write $\eta_\omega(s,a) = \sum_{i=1}^m p_i(s,a;\omega)\delta_{\theta_i}$, where $\omega$ comprises the online networks weights. We add a bar $\bar{\eta}$ to denote a delayed target network which is kept constant and is overwritten from $\eta$ every e.g. 10000 steps with the parameters from the online network. The corresponding expectation is denoted by $Q_\omega(s,a) = \sum_{i=1}^m p_i(s,a;\omega)\theta_i$. Given a transition $(s,a,r,s')$ the projected target is

$$\hat{\eta} = \Pi_C(b_{r,\gamma}\#\eta_{\bar{\omega}}(s',a^*)) =: \sum_{i=1}^m \hat{p}_i\delta_{\theta_i},$$

where $a^* = \mathrm{argmax}_{a'} Q_{\bar{\omega}}(s',a')$. Gradient descent on the weights with respect to some loss (here: cross-entropy) is used to move the distribution $\eta_\omega(s,a)$ towards the target distribution $\hat{\eta}$. As for the tabular algorithm variants with overestimation reduction intervene in the target definition. We keep track of two independently initialized online networks denoted by $\omega^A, \omega^B$ and a pair of respective target networks $\bar{\omega}^A, \bar{\omega}^B$. For each gradient step we simulate a vector of random variables with the same size as the batch size with each element determining which of the two estimators is being updated based on the respective transition with the same position in the batch. Accordingly, we use twice the batch size for these methods, so that on average per gradient step, the same number of transitions is used for each estimator, compared to the single-estimator case. We can now describe how to modify the targets for a given transition $(s,a,r,s')$. With the placeholder $\Gamma$ in

$$\bar{\eta}^{A/B} = \Pi_C(b_{r,\gamma}\#\Gamma^{A/B})$$

only the place holder is modified for different algorithms:

**Double C51:** Set $\Gamma^{A/B} = \eta_{\bar{\omega}^{B/A}}(s',z^*)$ with $z^* = \mathrm{argmax}_{a'} Q_{\bar{\omega}^{A/B}}(s',a')$.

**Clipped C51**: Inspired by (Fujimoto et al., 2018). Set $\Gamma^{A/B} = \eta_{\bar{\omega}^X}(s',z^*)$ with $z^* = \mathrm{argmax}_{a'} Q_{\bar{\omega}^{A/B}}(s',a')$

and $X = \mathrm{argmin}_{c\in\{A,B\}} Q_{\bar{\omega}^c}(s',z^*)$.[1]:

**ADDQ (us):** The ADDQ target uses

$$\Gamma^{A/B} = \beta\eta_{\bar{\omega}^{A/B}}(s',z^*) + (1-\beta)\eta_{\bar{\omega}^{B/A}}(s',z^*),$$

where $z^* = \mathrm{argmax}_{a'} Q_{\bar{\omega}^{A/B}}(s',a')$. The locally adaptive weights $\beta$ may depend on entire state-action return distributions. For the experiments we used the choice from Equation (1) based on the online networks $\eta_{\omega^A}, \eta_{\omega^B}$.

Results for two Atari environments and a RLiable comparison (Agarwal et al., 2021) are presented in Figure 4 and more extensively in Appendix E. The experiments show that ADDQ is more stable than QL and DQL (it never fails completely) and achieves higher scores in different metrics.

## 3.5. ADDQ adaptation for QRDQN

Modifications and results for quantile DRL (Dabney et al., 2018) are similar to the categorical setting. The setup is explained in Appendix D; for a quick view two experiments and a RLiable plot for probability of improvements can be found in Figure 4. More experimental results are provided in Appendix E. ADDQ is more stable (it never fails completely) and achieves higher scores in different metrics.

## 3.6. ADDQ adaptation for quantile SAC

Compared to Atari environments it is known that overestimation in MuJoCo environments is more severe. Using the double estimator in critic estimation is not enough, the clipping trick of TD3 (Fujimoto et al., 2018) greatly improves performance. Recall that, in the formulation of biased $Q$-learning of Section 3.1, clipping uses bias $\min\{Q^{B/A} - Q^{A/B}, 0\}$ instead of $Q^{B/A} - Q^{A/B}$. Thus, using our distributional overestimation identification to combine QL and DQL does not hint towards an algorithm competitive with SAC/TD3 or algorithms with more refined overestimation control such as REDQ (Chen et al., 2021) or TQC (Kuznetsov et al., 2020). For completeness of the article we still consider the ADDQ effect with standard single and double estimator. We used the base implementations from Stable-Baselines3 to study SAC with quantile regression. With a single critic estimator (we call this QRSAC), with double estimator, clipped double estimator (QB-SAC in the terminology of (Kuznetsov et al., 2020)), and ADDQ estimator (ADQRSAC). Experiments are shown in Figure 5 and more extensively in Appendix E.

As expected ADDQ improves QRSAC and double QRSAC but is not competitive with clipping. We leave for future work to experiment distributional RL based overestimation identification for REDQ and TQC.

---

[1]TD3 [(Fujimoto et al., 2018)] introduced clipping in an actor-critic setting, where the action is given by the actor. In our C51 adaptation, we select the greedy action based on the target network.

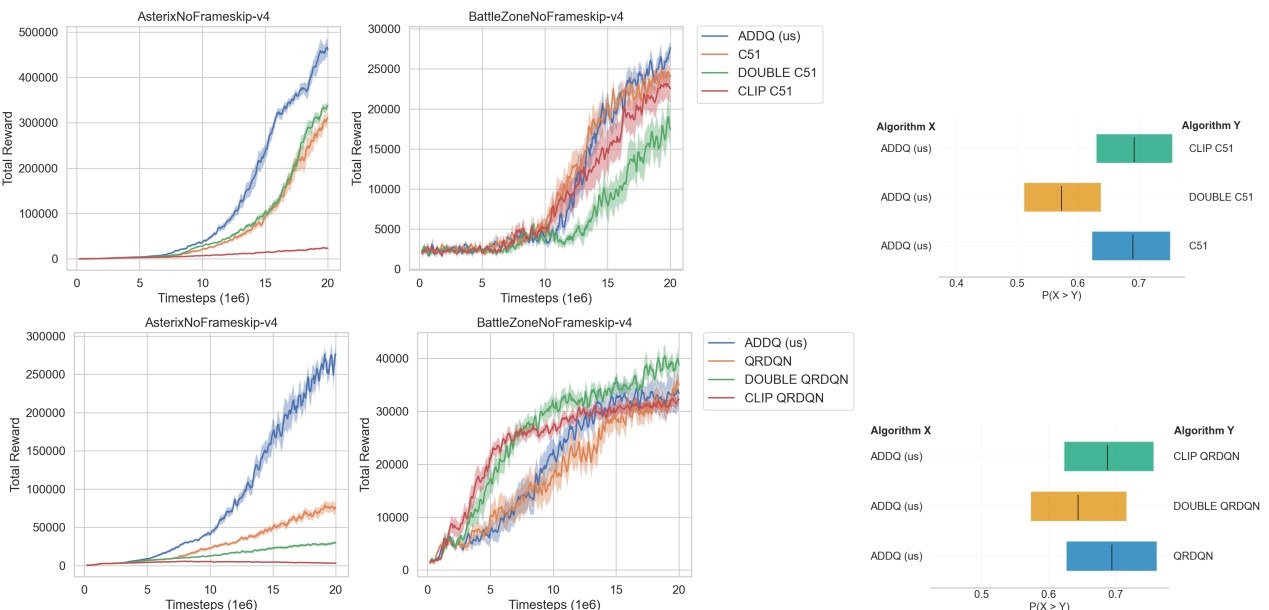

*Figure 4.* Our method ADDQ (blue) compared to distributional DQN (orange), with double estimator (green) and clipped double estimator (red). First row with categorical, second row quantile parametrization. Learning curves are given for two environments, 8 more in Appendix E. We used 10 seeds. RLiable probability of improvement plots are for all 10 environments, more metrics in Appendix E.

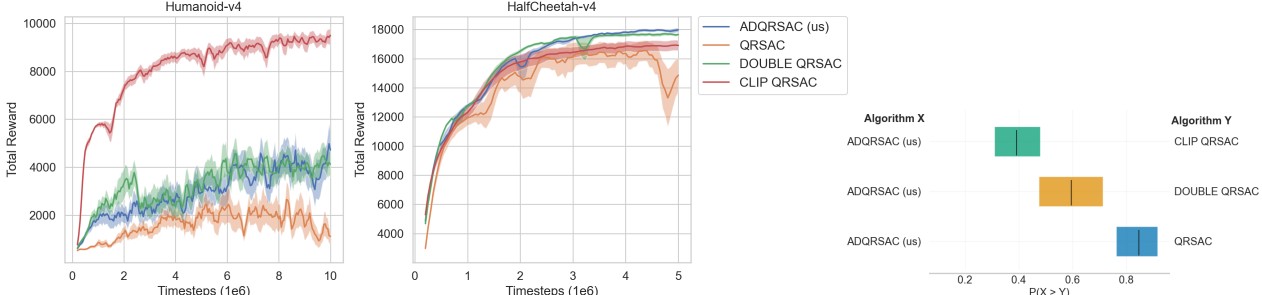

*Figure 5.* For completeness: Learning curves for two MuJoCo environments and RLiable probability of improvement on 5 environments. Adapting locally (blue, us) the double critic estimator (green) and direct estimator (orange) cannot (by far) reach performance of the clipping estimator (red). Nonetheless, ADDQ improves distributional direct and double critic estimators with almost no change to the code. Runs are averaged on 10 seeds, more details can be found in the Appendix E.

## 4. Summary, limitations, and future work

Built on theoretical insight for a bandit MDP we suggest to use sample variances in distributional RL to mitigate the overestimation of QL. Our approach does not use a novel estimation procedure for $Q$-values but instead combines known estimators and tries to use the better one. Using DRL, the agent in state-action pair $(s, a)$ has access to next-state information that predicts the overestimation of QL-updates and accordingly prevers QL- or an alternative. The approach can be incorporated in different estimation methods (e.g. (random) ensemble methods, truncation methods). For the present article we decided to improve DQL, leading to our algorithm ADDQ. The algorithm has the expected feature that in contrast to DQ/DQL it does not fail completely on some environments. Probability of improvement and normalized scores using the RLiable library show clear improvement of ADDQ to underlying deep DQ/DQL.

While ADDQ improves QL and DQL in different settings there is future work to be done. (i) It is clear that our probabilistic calculations can be extended. It is quite likely that results from Gaussian processes can be used for computations in more general settings. (ii) The exemplary choice of $\beta$ can be improved. It would be interesting to replace the hyperparameter thresholds by some adaptive learnable choice. (iii) The MuJoCo simulation study was only included for completeness, it would have been very surprising if a local adaptation of simple and double critic estimates could improve the clipped critic estimate (or even better algorithms such as REDQ or TQC). It is interesting future research to include local overestimation mitigation into REDQ (make the chosen ensemble number depend locally on sample variances) or TQC (make the number of truncated atoms depend locally on sample variances). (iv) Use sample variances to perform updates of target networks after non-constant steps.

## Code

The code used in our experiments can be found on GitHub: https://github.com/BommeHD/ADDQ.git.

## Acknowledgement

The authors acknowledge support by the state of Baden-Württemberg through bwHPC and the German Research Foundation (DFG) through grant INST 35/1597-1 FUGG.

## Impact statement

This paper presents work whose goal is to advance the field of machine learning, in particular reinforcement learning. There are many potential societal consequences of our work, none which we feel must be specifically highlighted here.

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

# A. Theoretical backup from probability theory

Theoretical backup for all known overestimation reduction algorithms is rather weak, the update rule of $Q$-learning makes precise computations very difficult. As we discuss below, the stochastic approximation rule requires computations with sums of random variables which is not well compatible with computations of maxima of random variables. Justifications for overestimation reduction algorithms are typically based on qualitative arguments for the estimation bias of expectations $\mathbb{E}[\max_i\{X_1, ..., X_K\}]$ of maxima of independent random variables even though random variables appearing in $Q$-learning maxima are clearly not independent. We will give explicit estimates using probabilistic insights for sums and maxima of independent random variables.

We should make very clear that there are different factors to the overestimation problem that are captured in the algorithmic approach of ADDQ. To provide some rigorous theoretical evidence, we focus in this section on the tabular setting without function approximation error. We refer the reader to the seminal article (Thrun & Schwartz, 1993) for some simplified computations on the overestimation effect caused by approximation errors.

## A.1. A lower bound computation for the overestimation bias in an episodic bandit MDP (Proof of Proposition 2.1)

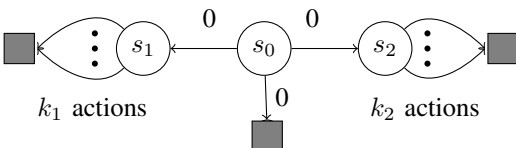

$$k_1 \text{ actions} \qquad k_2 \text{ actions}$$

The two-sided bandit MDP from the main text can be analyzed side by side. Without loss of generality we thus study the overestimation of the left-side which is essentially the bandit MDP that appeared quite a bit in the literature on overestimation of $Q$-learning (see (van Hasselt, 2011), Example 6.7 of (Sutton & Barto, 2018), or (Lan et al., 2020)).

Before stating our results on the overestimation bias we state a technical lemma that is based on the Sudakov-Fernique inequality, see (Sudakov, 1971; 1976; Fernique, 1975), which is a comparison inequality for Gaussian processes.

**Lemma A.1.** *Let* $k \in \mathbb{N}$ *and suppose* $(X_j^i)_{j \in \mathbb{N}, i \in \{1,...,k\}}$ *are iid* $\mathcal{N}(\mu, \sigma^2)$. *Further, let* $n_1, \ldots, n_k \in \mathbb{N}$ *such that* $n_1 + \cdots + n_k \leq kn$ *and set* $\gamma := \max_{1 \leq i \neq i' \leq k} |1/n_{n_i} + 1/n_{i'} - 2/n|$. *Then,*

$$\left| \mathbb{E}\left[ \max_{1 \leq i \leq k} \frac{1}{n_i} \sum_{j=1}^{n_i} X_j^i \right] - \mathbb{E}\left[ \max_{1 \leq i \leq k} \frac{1}{n} \sum_{j=1}^{n} X_j^i \right] \right| \leq \sqrt{2\gamma \ln k}. \tag{2}$$

*In particular, if, additionally,* $1/n_i + 1/n_{i'} \geq 2/n$ *for all* $i, i' \in \{1, \ldots, k\}$ *with* $i \neq i'$ *then*

$$\mathbb{E}\left[ \max_{1 \leq i \leq k} \frac{1}{n} \sum_{j=1}^{n} X_j^i \right] \leq \mathbb{E}\left[ \max_{1 \leq i \leq k} \frac{1}{n_i} \sum_{j=1}^{n_i} X_j^i \right]. \tag{3}$$

*Proof.* Given $n_1, \ldots, n_k \in \mathbb{N}$ with $n_1 + \cdots n_k \leq kn$, set $Y^i := n^{-1} \sum_{j=1}^{n} X_j^i$ and $Z^i := n_i^{-1} \sum_{j=1}^{n_i} X_j^i$ for any $i \in \{1, \ldots, k\}$. Clearly, $Y = (Y^1, \ldots, Y^k)$ and $Z = (Z^1, \ldots, Z^k)$ are Gaussian vectors with $\mathbb{E}[Y^i] = \mathbb{E}[Z^i]$ for all $i \in \{1, \ldots, k\}$ and

$$\gamma_{i,i'}^Y := \mathbb{E}\left[ \left( Y^i - Y^{i'} \right)^2 \right] = 2/n \qquad \text{and} \qquad \gamma_{i,i'}^Z := \mathbb{E}\left[ \left( Z^i - Z^{i'} \right)^2 \right] = 1/n_i + 1/n_{i'}$$

for all $i, i' \in \{1, \ldots, k\}$ with $i \neq i'$. Thus, (2) is an immediate consequence of Adler & Taylor, 2007, Theorem 2.2.5, Eq. (2.2.11). If, in addition, $1/n_i + 1/n_{i'} \geq 2$ for all $i \neq i'$ then $\gamma_{i,i'}^Y \leq \gamma_{i,i'}^Z$ and (3) follows from Adler & Taylor, 2007, Theorem 2.2.5, Eq. (2.2.12). $\square$

We are now in a position to give a lower bound on the overestimation bias for all exploration policies under the typical $\frac{1}{\#\text{visits}}$-step-size schedule. This extends the overestimation analysis of (van Hasselt, 2011) which was based on a simple synchronous exploration mechanism. The step-size schedule is crucial for our probabilistic analysis as it turns the analysis in a computation with sums and maxima of independent random variables. The sum structure is a consequence of the

simple fact that $z_t := \frac{1}{t} \sum_{i=1}^{t} a_{i-1}$ solves the stochastic approximation recursion $z_{t+1} = (1 - \alpha_t)z_t + \alpha_t a_t$, $z_0 = 0$, with $\alpha_t = \frac{1}{t+1}$. Here is a formal lower bound of the overestimation bias in the episodic bandit MDP. As mentioned above we only consider exploration of the left-side of the bandit MDP.

**Theorem A.2.** *Suppose $Q_0$ is initialized as the zero matrix and step-sizes are chosen as $\alpha_t(s, a) = \frac{1}{T_{s,a}(t)}$, with $T_{s,a}(t)$ the number of updates of $Q(s, a)$. If rewards are $\mathcal{N}(\mu, \sigma^2)$-distributed, then every sufficient exploratory exploration rule leads to overestimation bias at least*

$$\mathbb{E}[Q_{Nk}(s_0, "left")] - Q^*(s_0, "left") \geq \frac{\gamma}{\sqrt{\pi \log(2)}} \frac{\sigma\sqrt{\log(k)}}{\sqrt{N}}.$$

*By sufficient exploratory we mean that $\frac{1}{n_a(t)} + \frac{1}{n_{a'}(t)} \geq \frac{2}{N}$ for all actions $a, a'$ with $n_a(t) = T_{s,a}(t)$.*

In simple words, the expected overestimation in $n$ episodes of $Q$-learning on the left-side of the bandit MDP is at least of the order $\frac{\sigma\sqrt{k\log(k)}}{\sqrt{n}}$. Higher variance and more actions obviously lead to stronger overestimation.

*Proof.* In contrast to (van Hasselt, 2011) we do not assume synchronous exploration of all actions in each episode. Instead, we give a comparison argument that allows to reduce sufficiently exploratory exploration to simple cyclic exploration. The approach is motivated by the target network (here: target matrix) trick from DQN (Mnih et al., 2015). With target networks the update works as follows:

$$Q_{t+1}(s, a) \leftarrow (1 - \alpha_t)Q_t(s, a) + \alpha_t\big(r + \gamma \max_{a'} \bar{Q}_t(s', a')\big),$$

where $\bar{Q}$ is kept constant for a fixed number of steps and then updated from the current $Q$-matrix. Although the target network was introduced in DQN to reduce overfitting in function approximation, it also reduces the overestimation of $Q$-values. We use the latter effect locally in this proof to construct a lower bound. The target matrix is kept constant (0 matrix) at $s_0$ until the last update. We show that (i) this target matrix trick with cyclic exploration yields a lower bound for the overestimation bias of standard $Q$-learning with sufficient exploratory exploration strategy and (ii) allows for computations using elements of probability theory. In what follows we denote by $X_i^a$ the reward obtained when playing chosing $a$ for the $i$th time. By assumption all $(X_i^a)$ are iid.

**Step 1:** In the first step we show that using cyclic exploration (playing one action after the other) for $N$ rounds ($Nk$ steps in total) followed by one update at $(s_0, "left")$ minimizes the overestimation of $Q$-learning estimators $\hat{Q}^{cyc,tar}(s_0, "left")$ that explore each action at most $N$ times. For the claim we compare arbitrary $Q$-learning with the cyclic variant. Recalling the $Q$-learning update (with arbitrary exploration rule) and the step-size schedule shows that

$$Q_t(s_1, a) = \frac{1}{T_{s,a}(t)} \sum_{j=1}^{T_{s,a}(t)} X_j^a \sim \mathcal{N}(\mu, t\sigma^2).$$

Since $s_0$ is explored once per episode the step-size schedule of regular $Q$-learning yields

$$Q_{Nk}(s_0, "left") = \frac{1}{Nk} \sum_{t=1}^{Nk} \gamma \max_a Q_t(s_1, a)$$

for the $N$th episode. We next show that $\mathbb{E}[Q_{Nk}(s_0, "left")] \geq \mathbb{E}[\hat{Q}^{cyc,tar}(s_0, "left")]$. Denoting $n_k(t) = T_{s_1,a_k}(t)$ and

using Lemma A.1 we obtain

$$
\begin{aligned}
\mathbb{E}[Q_N(s_0,"left")] &= \mathbb{E}\Big[\frac{1}{Nk}\sum_{t=1}^{Nk}\gamma\max_a\big\{Q_t(s_1,a_1),...,Q_t(s_1,a_k)\big\}\Big] \\
&= \frac{1}{Nk}\sum_{t=1}^{Nk}\gamma\mathbb{E}\Big[\max\Big\{\frac{1}{n_1(t)}\sum_{j=1}^{n_1(t)}X_j^{a_1},...,\frac{1}{n_k(t)}\sum_{j=1}^{n_k(t)}X_j^{a_k}\Big\}\Big] \\
&\geq \frac{1}{Nk}\sum_{t=1}^{Nk}\gamma\mathbb{E}\Big[\max\Big\{\frac{1}{N}\sum_{j=1}^{N}X_j^{a_1},...,\frac{1}{N}\sum_{j=1}^{N}X_j^{a_k}\Big\}\Big] \\
&= \gamma\mathbb{E}\Big[\max\Big\{\frac{1}{N}\sum_{j=1}^{N}X_j^{a_1},...,\frac{1}{N}\sum_{j=1}^{N}X_j^{a_k}\Big\}\Big] \\
&= \mathbb{E}[\hat{Q}^{cyc,tar}(s_0,"left")].
\end{aligned}
$$

**Step 2:** We now analyze the overestimation bias for the $Q$-learing with target matrix trick after $Nk$ episodes. To deduce the claim we use a fact on the expectation of the maximum of independent $\mathcal{N}(\mu,\sigma^2)$-distributed random variables:

$$
\mu + \frac{1}{\sqrt{\pi\log(2)}}\sigma\sqrt{\log(k)} \leq \mathbb{E}[\max\{X_1,...,X_k\}] \leq \mu + \sqrt{2}\sigma\sqrt{\log(k)}
$$

The inequality can for instance be found in (van Handel, 2016). According to the step-size schedule the update is as follows. If $N = nk$, i.e. in cyclic exploration every action is played $n$ times, then $Q_{Nk}^{cyc,tar}(s_1,a) = \frac{1}{N}\sum_{i=1}^{N}X_i^a$ for independent $\mathcal{N}(\mu,\sigma^2)$-distributed reward samples. Thus, $Q_{Nk}(s_1,a_1),...,Q_{Nk}(s_1,a_k)$ are iid and $\mathcal{N}(\mu,\frac{\sigma^2}{N})$-distributed. The final update after $N$ episodes is to set

$$
\hat{Q}^{cyc,tar}(s_1,"left") = \gamma\max\{Q_{Nk}(s_1,a_1),...,Q_{Nk}(s_1,a_k)\} \stackrel{(d)}{=} \gamma\max\{Z_1,...,Z_k\}
$$

for $Z_1,...,Z_k$ iid $\mathcal{N}(\mu,\frac{\sigma^2}{N})$. The claim follows from the lower bound on the expectation of maxima of independent Gaussians. $\qquad\square$

The computations give a number of quantitative insights. First, it becomes very clear why explicit computations for $Q$-learning are complicated. The stochastic approximation update is closely related to sums (exactly equal to sums for $\frac{1}{\#\text{visits}}$ step-sizes, while the update target involves a maximum. Unfortunately, sums and maxima of random variables do not get along well. We thus performed computations for Gaussian reward distributions for which sums are Gaussian again. Maxima of Gaussian random variables (processes) are a well-studied field in Mathematics and estimates can be derived.

Finally, the computations show how variances influence the overestimation problem. Stochastic rewards with high variance are more problematic than rewards with small variance. In the next section we study distribution RL in the same simple problem.

### A.2. On the use of variance in local overestimation control via distributional RL (Proof of Proposition 2.2)

Our computations for the bandit MDP above suggest uncertainty at next states (aleatoric uncertainty of rewards, epistemic uncertainty through small $N$) lead to larger overestimation. Thus, algorithms mitigating the overestimation problem locally at $(s,a)$ must somehow use uncertainty at next states $s'$. In the following proposition we analyze the distributional $Q$-learning update scheme (compare Section 8 of (Rowland et al., 2018))

$$
\eta(s,a) \leftarrow (1-\alpha)\eta(s,a) + \alpha\big(b_{r,\gamma}\#\eta(s',a^*)\big),
$$

with $a^* = \text{argmax}_{a'}Q(s',a')$, where $Q(s',a')$ are the (random) sample means of the estimated return distributions $\eta(s',a')$. The bootstrap function is $b_{r,\gamma}(z) = r + \gamma z$ and $f\#\nu(B) := \nu(f^{-1}(B))$ denotes the push-forward of measures. Recall from the main text that distributional QL learns random measures, $\eta(s,a)$ itself is a probability measure that depends on the random samples used in the updates. As a probability measure $\eta(s,a)$ has an expectation and a variance that are both random in terms of the random samples. The situation becomes a bit tricky as we are going to take variances of the variances.

To avoid confusion we use an analogy to statistics and speak of (random) sample averages $M$ (resp. sample variances $S^2$) of $\eta(s, a)$ and expectation $\mathbb{E}$ (resp. variance $\mathbb{V}$) for the integrals against the true randomness induced by the probability space behind all appearing random variables.

Motivated by the previous section we only study cyclic exploration with "target measure", i.e. all actions in $s_1$ are explored $N$ times before bootstrapping the estimates to $s_0$. Using the standard $\frac{1}{\#\text{visits}}$-step-size schedule allows us to carry out fairly explicit computations using tools from probability theory. Most importantly, we can derive the exact distribution for the sample variance of $\eta(s_0, \text{"left"})$, the state-action pair at risk for overestimation. The insight obtained from the computation motivates our ADDQ algorithm.

**Proposition A.3.** *Consider the bandit MDP introduced above with $\mathcal{N}(\mu, \sigma^2)$ reward distributions and $k$ actions. Denote by $\hat{\eta}^{cyc,tar}(s_0, \text{"left"})$ the return distribution of distributional Q-learning with cyclic exploration, step-size schedule $\alpha_t(s, a) = \frac{1}{T_{s,a}(t)}$, and "target distribution". More precisely, all arms are $N$ times explored before the first and only update at $(s_1, \text{"left"})$. It then follows that*

$$S^2(\hat{\eta}^{cyc,tar}(s_0, \text{"left"})) \sim \frac{\sigma^2}{N-1}\chi^2_{N-1},$$

*where $\chi^2_n$ denotes the chi-squared distribution with $n$ degrees of freedom.*

It is interesting to note that while there is a simple distributional expression for the sample variance there is no simple expression for the distribution of the sample mean $\max\{\frac{1}{N}\sum_{i=1}^N X_i^{a_1}, ..., \frac{1}{N}\sum_{i=1}^N X_i^{a_k}\}$ which is the maximum of independent Gaussians.

*Proof.* For the proof we need to spell-out the distributional $Q$-learning update and then use basic results from statistic for sample-mean/sample-variances.

Let us first understand the distributional update at the last step. Similar to scalar stochastic approximation schemes, the $\frac{1}{n+1}$-step-size schedule also gives the distributional estimator

$$\hat{\eta}^{cyc,tar}(s_1, a) = \frac{1}{N}\sum_{i=1}^N \delta_{X_i^a} \tag{4}$$

for the pre-terminal state $s_1$ after $N$ explorations of action $a$. To see why write $\eta_n := \frac{1}{n}\sum_{i=1}^n \delta_{X_i^a}$ to get

$$\eta_{n+1} = \frac{1}{n+1}\sum_{i=1}^{n+1} \delta_{X_i^a} = \left(1 - \frac{1}{n+1}\right)\frac{1}{n}\sum_{i=1}^n \delta_{X_i^a} + \frac{1}{n+1}\delta_{X_{n+1}^a} = (1 - \alpha_n)\eta_n + \alpha_n\delta_{X_{n+1}^a}$$

and recall that there is no max-term in the update for pre-terminal states. Note that the equal weights $\frac{1}{N}$ are a consequence of the step-size schedule. To compute the return distribution at $s_0$ for action "left" we need to identify $a^*$. For that sake we must identify the sample means of $\hat{\eta}^{cyc,tar}(s_1, a)$:

$$\hat{Q}^{cyc,tar}(s_1, a) = \frac{1}{N}\sum_{i=1}^N X_i^a \sim \mathcal{N}(\mu, \sigma^2/N)$$

Now we chose $a^* = \arg\max_a \hat{Q}^{cyc,tar}(s_1, a)$ and set

$$\hat{\eta}^{cyc,tar}(s_0, \text{"left"}) = b_{0,\gamma}\#\hat{\eta}^{cyc,tar}(s_1, a^*) =: \gamma X.$$

While we do not know much about $X$ (it is the empirical distribution of the set of Gaussians $X_1^a, ..., X_N^a$ with maximal sum) we know the exact distribution of the sample variance for the following reason. The sample variance of every $\eta^{cyc,tar}(s_1, a)$ is nothing what is called sample variance of the iid observation $X_1^a, ..., X_N^a$ in statistics. If $M_a := \frac{1}{N}\sum_{i=1}^N X_i^a$ is the sample mean and $S_a^2 := \frac{1}{N-1}\sum_{i=1}^N (X_i^a - M^a)^2$ the sample variance then it is well-known that

- $\mathbb{E}[S_a^2] = \sigma^2,$

- $S_a^2$ is $\frac{\sigma^2}{N-1}\chi_{N-1}^2$-distributed,

- $M_a$ and $S_a^2$ are independent.

The third property implies that if $S_{a_1}^2, ..., S_{a_k}^2$ are independent sample variances and $a^*$ is chosen to maximize the sample means, then also $S_{a^*}^2 \sim \frac{\sigma^2}{N-1}\chi_{N-1}^2$ while $M_{a^*} \not\sim M_a$. $\qquad\qquad\qquad\qquad\qquad\qquad\qquad\square$

As a consequence, even though we cannot compute the distribution of the sample mean of $\hat{\eta}^{cyc,tar}(s_0, \text{"left"})$ we can compute the distribution of its sample variance as $\frac{\sigma^2}{N-1}\chi_{N-1}^2$. The expectation is $\mathbb{E}[S_a^2] = \sigma^2$ while the variance is $\mathbb{V}(S_a^2) = \frac{2\sigma^4}{N-1}$. Note that one might be tempted to believe the sample variance is independent of the number $k$ of actions. This is not the case, as the number of explorations $Nk$ needed for $N$ explorations depends on $k$. Alternatively, one might denote the number of total episodes by $n$ and replace $N$ by $\lfloor \frac{n}{k} \rfloor$.

We now come back to the motivation of our ADDQ algorithm, in particular the choice of $\beta$. Let us consider the two-sided bandit MDP with $\mathcal{N}(\mu_1, \sigma_1^2)$ (resp. $\mathcal{N}(\mu_2, \sigma_2^2)$) distributed rewards. Suppose $\mu_1 > \mu_2$ are small but $\sigma_1^2 \ll \sigma_2^2$ and/or $k_1 \ll k_2$. The MDP is delicate as the following can happen using QL (overestimation) and DQL (global overestimation reduction). In QL the agent will believe for a long time that "right" is the optimal action in $s_0$. In DQL the agent will believe for a long time that "down" is the optimal action as both non-trivial $Q$-values can be underestimated to be negative. It is thus more reasonable to mitigate overestimation locally instead of globally. This is what ADDQ achieves through the choice of $\beta$.

To use our explicit computations above, the agent explores both sides with cyclic exploration and target-matrix update. Now assume both sides have been explored with a total number of $n$ episodes each. The agent knows the estimates $\hat{\eta}(s_0, \text{"left"})$ (resp. $\hat{\eta}(s_0, \text{"right"})$) and the corresponding $Q$-values (sample means of return distributions) $\hat{Q}(s_0, \text{"left"})$ (resp. $\hat{Q}(s_0, \text{"right"})$). From the overestimation study of the previous section, the agent unwittingly overestimated the true $Q$-values in the order of $\frac{\sigma_1\sqrt{k_1 \log(k_1)}}{\sqrt{n}}$ (resp. $\frac{\sigma_2\sqrt{k_2 \log(k_2)}}{\sqrt{n}}$) and will believe "right" is the correct action. Now the smart agent also knows he/she should take into account the sample variances that is known to the agent and we know are of the order $\sigma_1^2$ (resp. $\sigma_2^2$) with concentration depending on $k_1$ (resp. $k_2$). The local overestimation control of $\beta$ (based on relative sample variances) from (1) thus compares $\sigma_1^2$ to $\sigma_2^2$ and suggests to mitigate overestimation of $\hat{Q}(s_0, \text{"right"})$. In our algorithm we use double $Q$-learning to mitigate, the same idea can of course be integrated into other algorithms (such as changing number of truncation atoms in truncated quantile critics algorithms (Kuznetsov et al., 2020)).

## B. Experimental confirmation of theoretical results for two-sided bandit MDP

We provide an experimental analysis of the two-sided bandit MDP for which we proved theoretical results on the overestimation. For reimplementation purposes we collect here all required information on the environment and the training for all plots.

Environment:

- $\gamma = 0.9$

- $\mu_1 = -0.1$, $\mu_2 = 0.1$, $k_2 = 5$, $\sigma_2 = 1$; the correct decision is thus moving to the right in the Start State.

Distributional properties for ADDQ:

- categorical parametrization, 51 atoms equally spaced on $[-3, 3]$

- initialization as $\delta_0$

Algorithmic choices:

- $\beta$ is chosen according to (1)

- step-size schedules $\alpha_t(s, a) = \frac{1}{T_{s,a}(t)}$, with $T_{s,a}(t)$ the number of visits in $(s, a)$ up to time $t$, i.e. $\frac{1}{n}$ state-action wise counted

- exploration: either $\epsilon$-greedy with $\epsilon$ linearly decreasing from 1 to 0.1 in 10000 steps, then constant (E) or uniform random (U)

Policy evaluation: evaluation of 3 steps with a frequency of every 500 steps using current greedy policy, correct action rates refer to if the exploration was greedy

In order to demonstrate the proven proportionality of the overestimation of $Q$-values by QL in the number of arms and the variance to the left, we keep one of the values fixed and compare different values in the other one.

- First experiment: $\sigma_1 = 5$ fixed, iterating over $k_1 = 5, 10, 15, 20$, denoted as $K$ in the legend

- Second experiment: $k_1 = 10$ fixed, iterating over $\sigma_1 = 2, 4, 6, 8$, denoted as $S$ in the legend

The plots also demonstrate that

- ADDQ is much better in terms of bias, leveraging local information given by the (relative) variances

- the variances and relative variances at state 0 capture the real variances given by the MDP

- although our theorems assumed a sufficiently exploratory policy, the results seem to generalize to the much more commonly used $\epsilon$-greedy setting

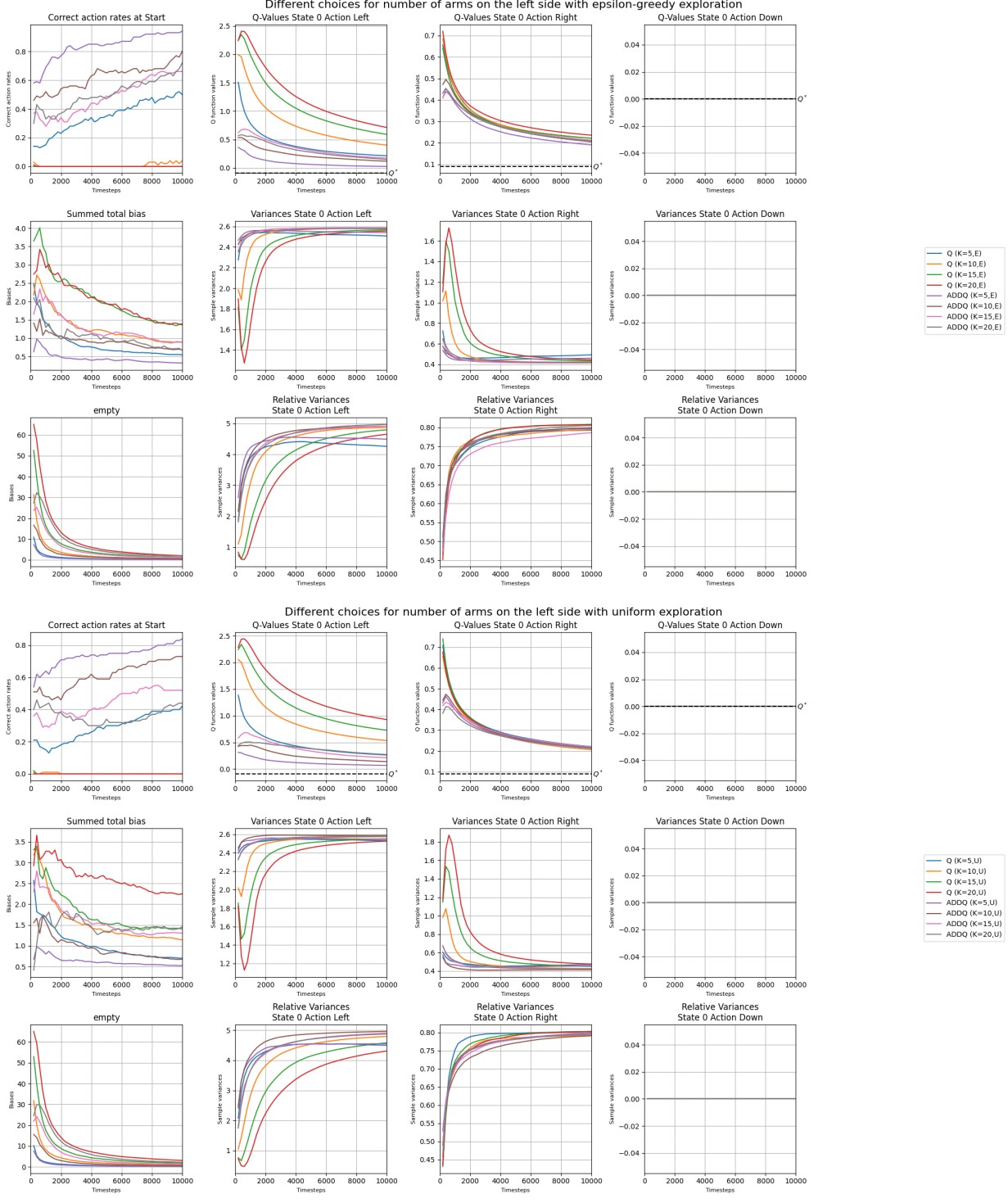

*Figure 6.* Comparing ADDQ and QL on two-sided bandit MDP with different number of arms on the left side and different exploration settings.

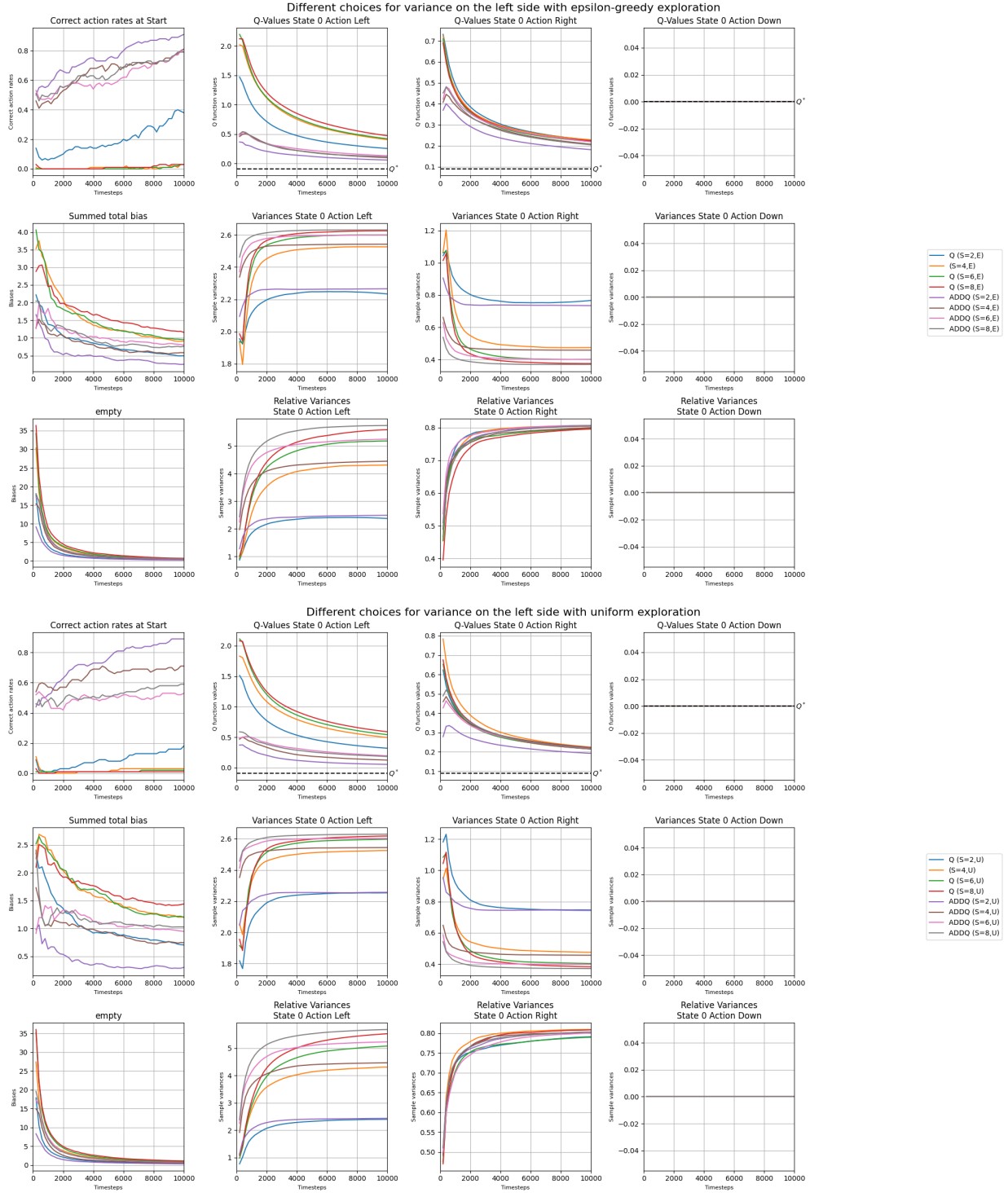

*Figure 7.* Comparing ADDQ and QL on two-sided bandit MDP with different variances on the left side and different exploration settings.

# C. Grid world details

## C.1. Experiment from the main text

For reimplementation purposes we collect here all required information on the environment and the training for all plots.

Environment:

- $\gamma = 0.9$

- white low stochastic high average reward region: rewards $-0.05$, $+0.05$ with equal probabilities

- gray high stochastic low average reward region: rewards $-2.1$, $+2$ with equal probabilities

- goal: deterministic reward of $1$, fake goal: deterministic reward of $0.65$

Distributional properties for CategoricalQ, CategoricalDouble, and ADDQ:

- categorical parametrization, 51 atoms equally spaced on $[-3, 3]$

- initialization as $\delta_0$

Algorithmic choices:

- $\beta$ is chosen according to (1)

- step-size schedule $\alpha_t(s, a) = \frac{1}{T_{s,a}(t)}$, with $T_{s,a}(t)$ the number of visits in $(s, a)$ up to time $t$, i.e. $\frac{1}{n}$ state-action wise counted,

- exploration: $\varepsilon$-greedy with $\varepsilon$ linearly decreasing from $1$ to $0.1$ in $10000$ steps, then constant

Policy evaluation: evaluation of 6 steps with a frequency of every 500 steps, correct action rates refer to if the exploration was greedy

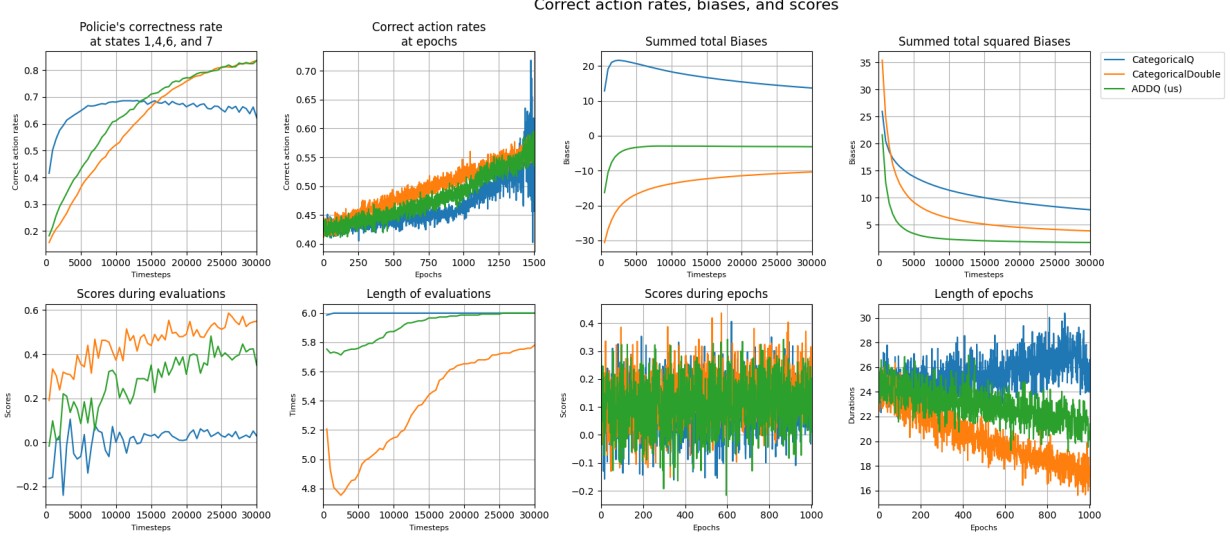

*Figure 8.* additional plots

For completeness we give plots pairing learning curves for $Q$-values with the corresponding sample variance. We give all state-action combinations for the interesting states 1 and 4 next to the fake goal, 6 and 7 next to the region with high stochasticity, and 10 and 14 before leaving the region with high stochasticity.

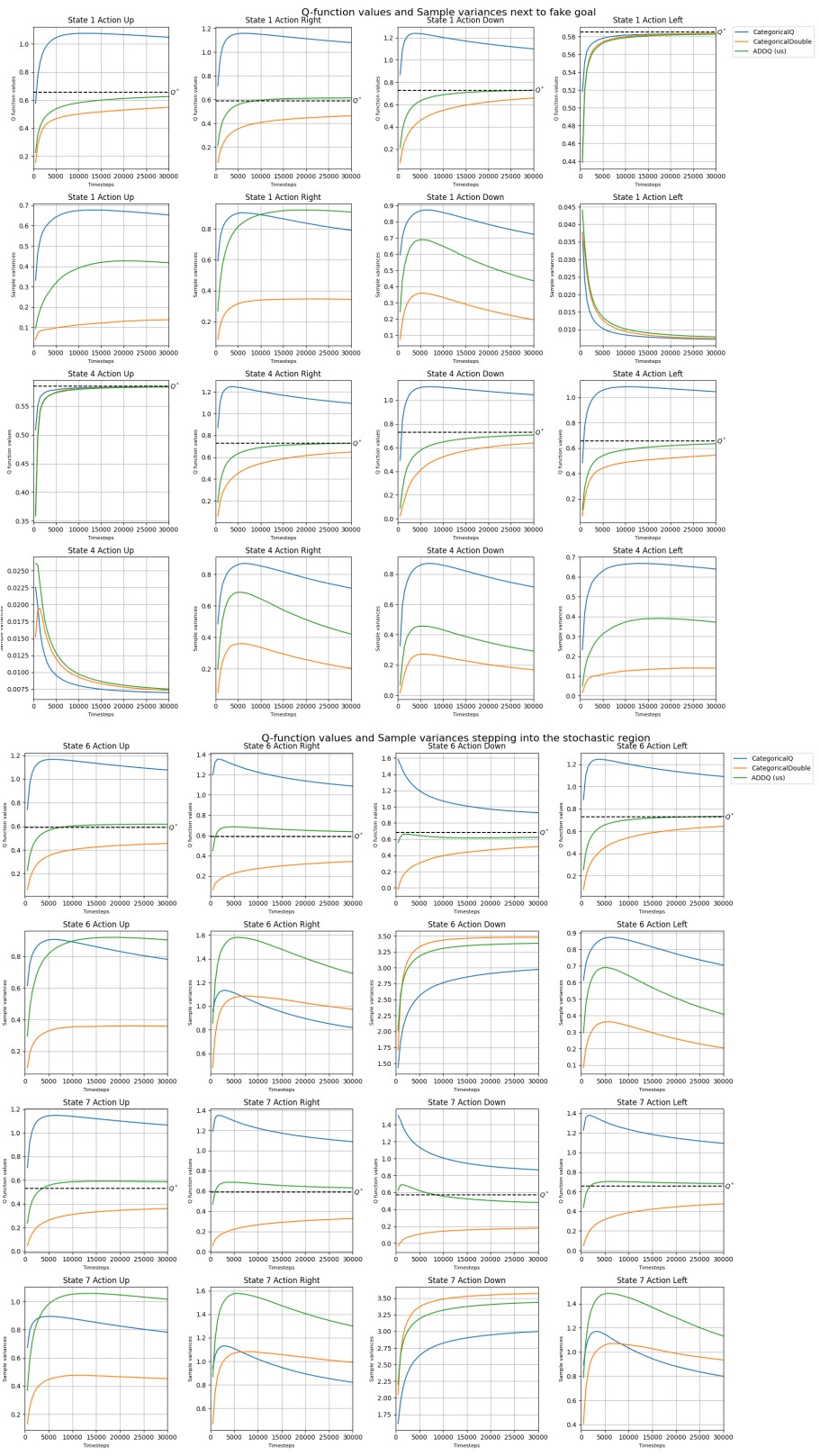

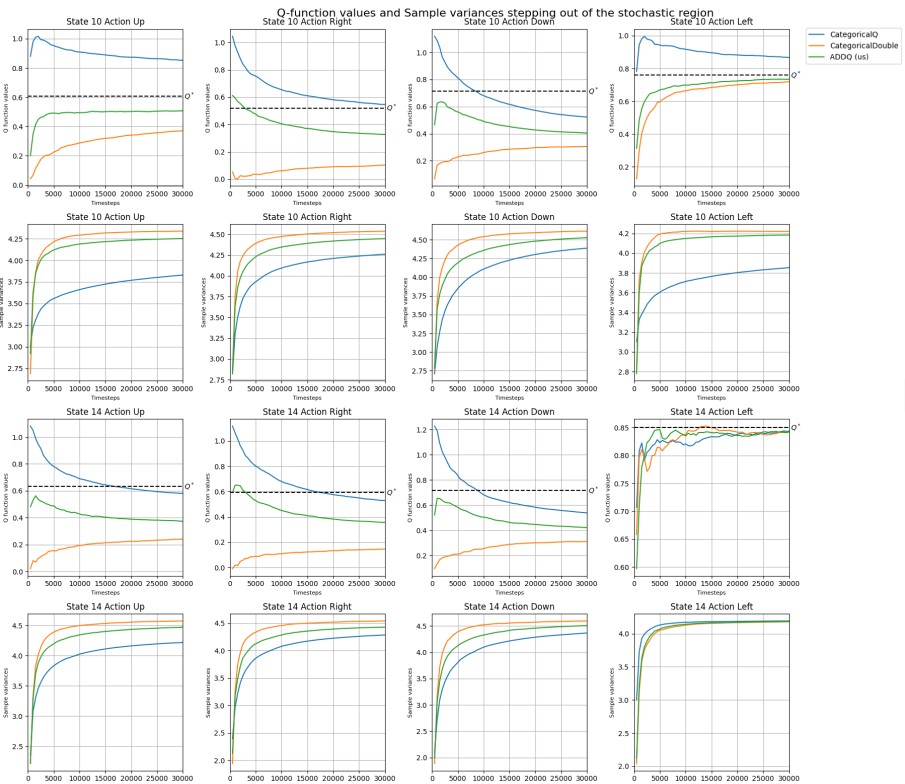

Finally, the following plot demonstrates that, in this GridWorld example, the relative variances are also strongly determined by the variances of the next state.

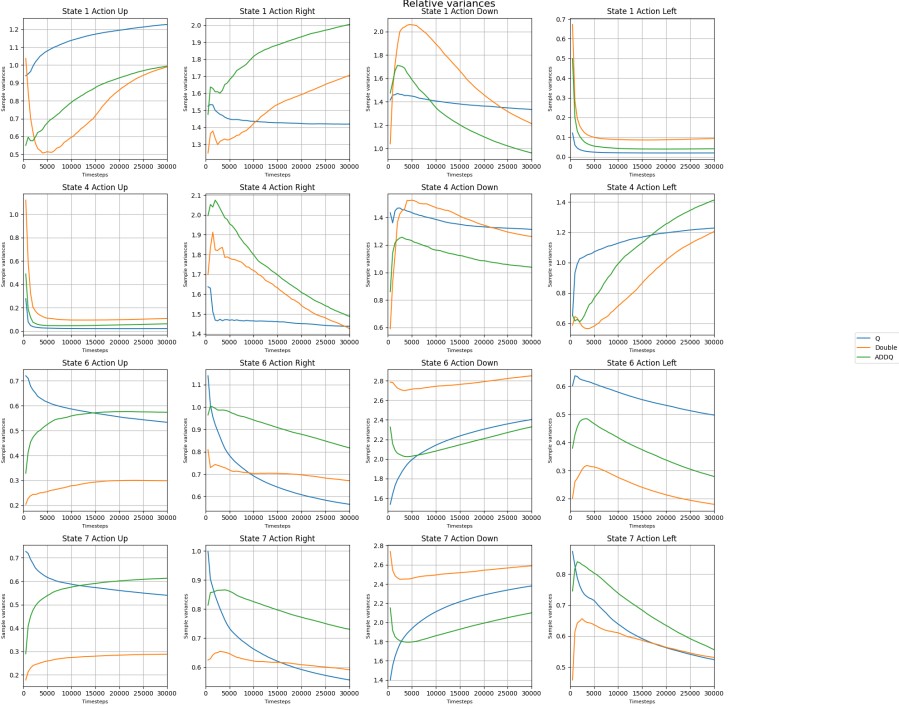

## C.2. Ablation study

In the following, on the same environment as before, we experiment with different choices for $\beta$ and compare with weighted DQL (Zhang et al., 2017) with $c = 10$ (WDQ), the standard choice from that paper. The choice of beta's names are comprised as follows:

- (Optional) First two letters: Left-tilted (lt), Right-tilted (rt)

- First/Third letter: Neutral (n), Aggressive (a), Conservative (c)

- Final digit: Refers to the number of intervals in the definition of Beta (3 or 5)

The choices of Aggressive, Conservative, and Neutral refer to the trade-off of no interpolation (just choosing which Algorithm's update to take) and softening the interpolation, with neutral being in between the two choices and corresponding to the choice presented in the main body. Left- and Right-tilted refers to shifting the intervals for the relative Variance to fall into while choosing the interpolation coefficient. Left-tilted favors the Q update, Right-tilted the DQ update.
The concrete choices are:

$$n3: \quad \beta := \begin{cases} 0.75 & : S^2_{rel}(s,a) < 0.75 \\ 0.5 & : S^2_{rel}(s,a) \in [0.75, 1.25] \\ 0.25 & : S^2_{rel}(s,a) > 1.25 \end{cases}, \qquad a3: \quad \beta := \begin{cases} 1 & : S^2_{rel}(s,a) < 0.99 \\ 0.5 & : S^2_{rel}(s,a) \in [0.99, 1.01] \\ 0 & : S^2_{rel}(s,a) > 1.01 \end{cases}$$

$$ltn3: \quad \beta := \begin{cases} 0.75 & : S^2_{rel}(s,a) < 1.25 \\ 0.5 & : S^2_{rel}(s,a) \in [1.25, 1.75] \\ 0.25 & : S^2_{rel}(s,a) > 1.75 \end{cases}, \qquad lta3: \quad \beta := \begin{cases} 1 & : S^2_{rel}(s,a) < 1.49 \\ 0.5 & : S^2_{rel}(s,a) \in [1.49, 1.51] \\ 0 & : S^2_{rel}(s,a) > 1.51 \end{cases}$$

$$rtn3: \quad \beta := \begin{cases} 0.75 & : S^2_{rel}(s,a) < 0.25 \\ 0.5 & : S^2_{rel}(s,a) \in [0.25, 0.75] \\ 0.25 & : S^2_{rel}(s,a) > 0.75 \end{cases}, \qquad rta3: \quad \beta := \begin{cases} 1 & : S^2_{rel}(s,a) < 0.49 \\ 0.5 & : S^2_{rel}(s,a) \in [0.49, 0.51] \\ 0 & : S^2_{rel}(s,a) > 0.51 \end{cases}$$

$$c3: \quad \beta := \begin{cases} 0.6 & : S^2_{rel}(s,a) < 0.6 \\ 0.5 & : S^2_{rel}(s,a) \in [0.6, 1.4] \\ 0.4 & : S^2_{rel}(s,a) > 1.4 \end{cases}, \qquad n5: \quad \beta := \begin{cases} 1 & : S^2_{rel}(s,a) \leq 0.25 \\ 0.75 & : S^2_{rel}(s,a) \in (0.25, 0.75) \\ 0.5 & : S^2_{rel}(s,a) \in [0.75, 1.25] \\ 0.25 & : S^2_{rel}(s,a) \in (1.25, 1.75) \\ 0 & : S^2_{rel}(s,a) \geq 1.75 \end{cases}$$

$$ltc3: \quad \beta := \begin{cases} 0.6 & : S^2_{rel}(s,a) < 1.1 \\ 0.5 & : S^2_{rel}(s,a) \in [1.1, 1.9] \\ 0.4 & : S^2_{rel}(s,a) > 1.9 \end{cases}, \qquad ltn5: \quad \beta := \begin{cases} 1 & : S^2_{rel}(s,a) \leq 0.75 \\ 0.75 & : S^2_{rel}(s,a) \in (0.75, 1.25) \\ 0.5 & : S^2_{rel}(s,a) \in [1.25, 1.75] \\ 0.25 & : S^2_{rel}(s,a) \in (1.75, 2.25) \\ 0 & : S^2_{rel}(s,a) \geq 2.25 \end{cases}$$

$$rtc3: \quad \beta := \begin{cases} 0.6 & : S^2_{rel}(s,a) < 0.1 \\ 0.5 & : S^2_{rel}(s,a) \in [0.1, 0.9] \\ 0.4 & : S^2_{rel}(s,a) > 0.9 \end{cases}, \qquad rtn5: \quad \beta := \begin{cases} 1 & : S^2_{rel}(s,a) \leq -0.25 \\ 0.75 & : S^2_{rel}(s,a) \in (-0.25, 0.25) \\ 0.5 & : S^2_{rel}(s,a) \in [0.25, 0.75] \\ 0.25 & : S^2_{rel}(s,a) \in (0.75, 1.25) \\ 0 & : S^2_{rel}(s,a) \geq 1.25 \end{cases}$$

a5: $\beta := \begin{cases} 1 & : S^2_{rel}(s,a) \leq 0.99 \\ 0.75 & : S^2_{rel}(s,a) \in (0.99, 0.995) \\ 0.5 & : S^2_{rel}(s,a) \in [0.995, 1.005] \\ 0.25 & : S^2_{rel}(s,a) \in (1.005, 1.01) \\ 0 & : S^2_{rel}(s,a) \geq 1.01 \end{cases}$,

c5: $\beta := \begin{cases} 0.7 & : S^2_{rel}(s,a) \leq 0.1 \\ 0.6 & : S^2_{rel}(s,a) \in (0.1, 0.7) \\ 0.5 & : S^2_{rel}(s,a) \in [0.7, 1.3] \\ 0.4 & : S^2_{rel}(s,a) \in (1.3, 1.9) \\ 0.3 & : S^2_{rel}(s,a) \geq 1.9 \end{cases}$

lta5: $\beta := \begin{cases} 1 & : S^2_{rel}(s,a) \leq 1.49 \\ 0.75 & : S^2_{rel}(s,a) \in (1.49, 1.495) \\ 0.5 & : S^2_{rel}(s,a) \in [1.495, 1.505] \\ 0.25 & : S^2_{rel}(s,a) \in (1.505, 1.51) \\ 0 & : S^2_{rel}(s,a) \geq 1.51 \end{cases}$,

ltc5: $\beta := \begin{cases} 0.7 & : S^2_{rel}(s,a) \leq 0.6 \\ 0.6 & : S^2_{rel}(s,a) \in (0.6, 1.2) \\ 0.5 & : S^2_{rel}(s,a) \in [1.2, 1.8] \\ 0.4 & : S^2_{rel}(s,a) \in (1.8, 2.4) \\ 0.3 & : S^2_{rel}(s,a) \geq 2.4 \end{cases}$

rta5: $\beta := \begin{cases} 1 & : S^2_{rel}(s,a) \leq 0.49 \\ 0.75 & : S^2_{rel}(s,a) \in (0.49, 0.495) \\ 0.5 & : S^2_{rel}(s,a) \in [0.495, 0.505] \\ 0.25 & : S^2_{rel}(s,a) \in (0.505, 0.51) \\ 0 & : S^2_{rel}(s,a) \geq 0.51 \end{cases}$,

rtc5: $\beta := \begin{cases} 0.7 & : S^2_{rel}(s,a) \leq -0.4 \\ 0.6 & : S^2_{rel}(s,a) \in (-0.4, 0.2) \\ 0.5 & : S^2_{rel}(s,a) \in [0.2, 0.8] \\ 0.4 & : S^2_{rel}(s,a) \in (0.8, 1.4) \\ 0.3 & : S^2_{rel}(s,a) \geq 1.4 \end{cases}$

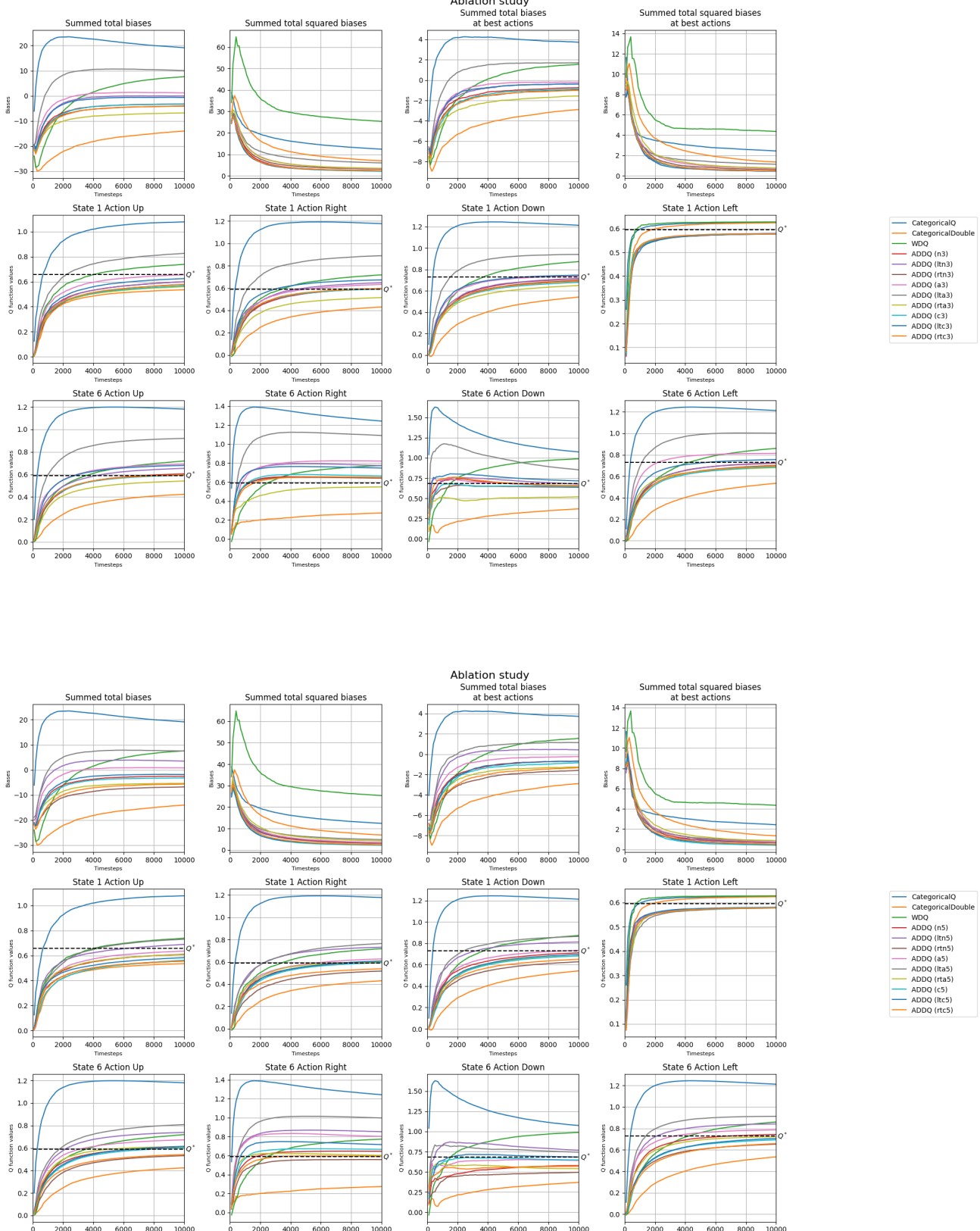

*Figure 9.* Ablation study plots regarding bias improvement. State 1 is adjacent to the fake goal, state 6 adjacent to the stochastic region.

It turns out that

- the choice of thresholds in $\beta$ (hyperparameter to the algorithm) is harmless, results do not vary a lot,

- conservative choices seem to work especially well.

- weighted double Q-learning, an algorithm that is similar to ADDQ with non-distributional choice of $\beta$ is improved by our locally adaptive distributional RL based choice of $\beta$.

### C.3. Comparison to other algorithms

We compared ADDQ with other bias reduction methods using different hyperparameters used in the respective papers on the same GridWorld environment.

- Maxmin with 2, 4, 6, and 8 ensembles (Lan et al., 2020)

- Ensemble Bootstrapped QL (EBQL) with 3, 7, 10, and 15 ensembles (Peer et al., 2021)

- Randomized Ensemble DQL (REDQ) with 3 and 5 ensembles and size 1 and 2 of random update subset sizes (Chen et al., 2021)

It turns out that ADDQ decreases the estimation bias stronger than those algorithms while only using two ensembles.

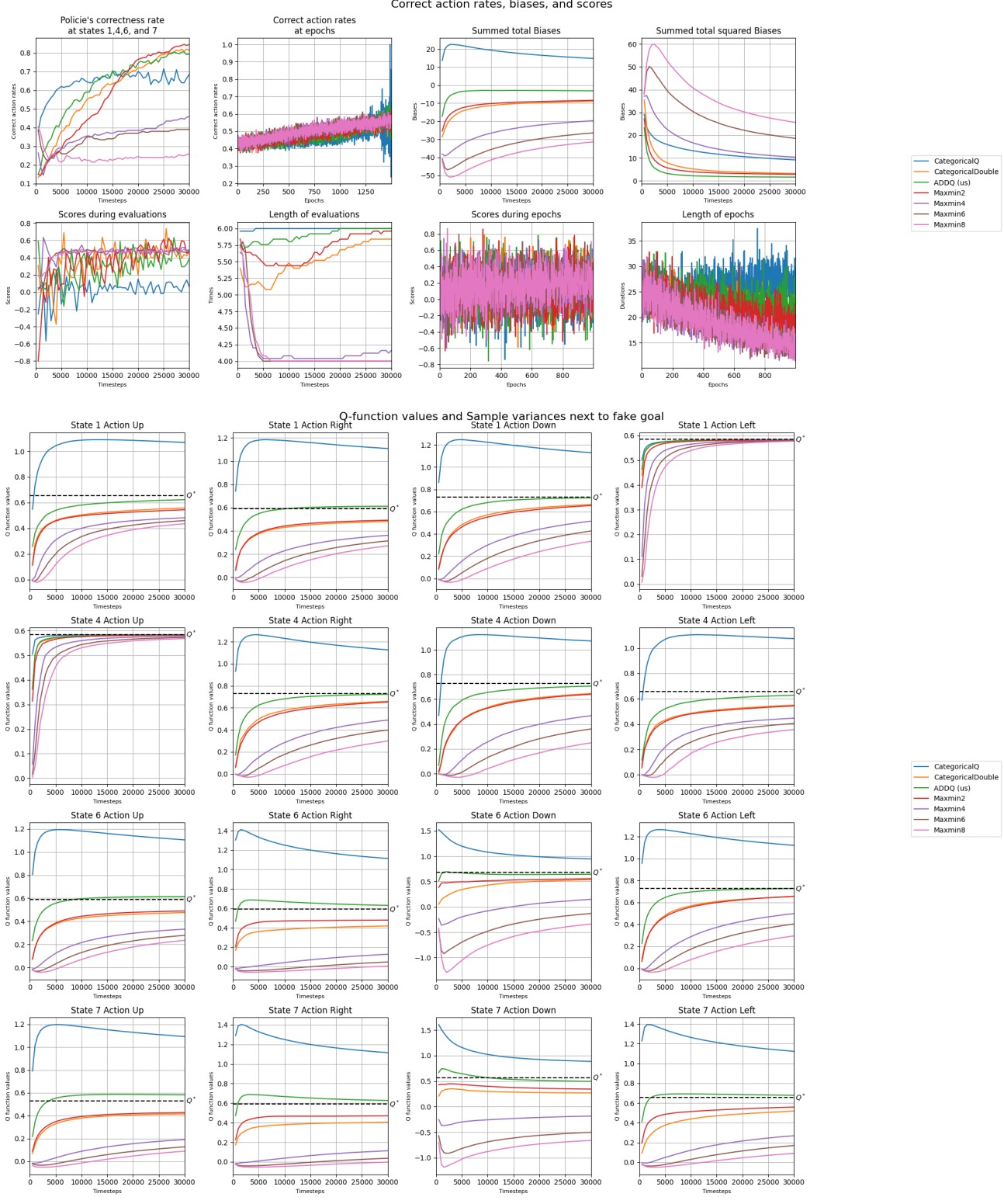

*Figure 10.* Comparison to MaxMin.

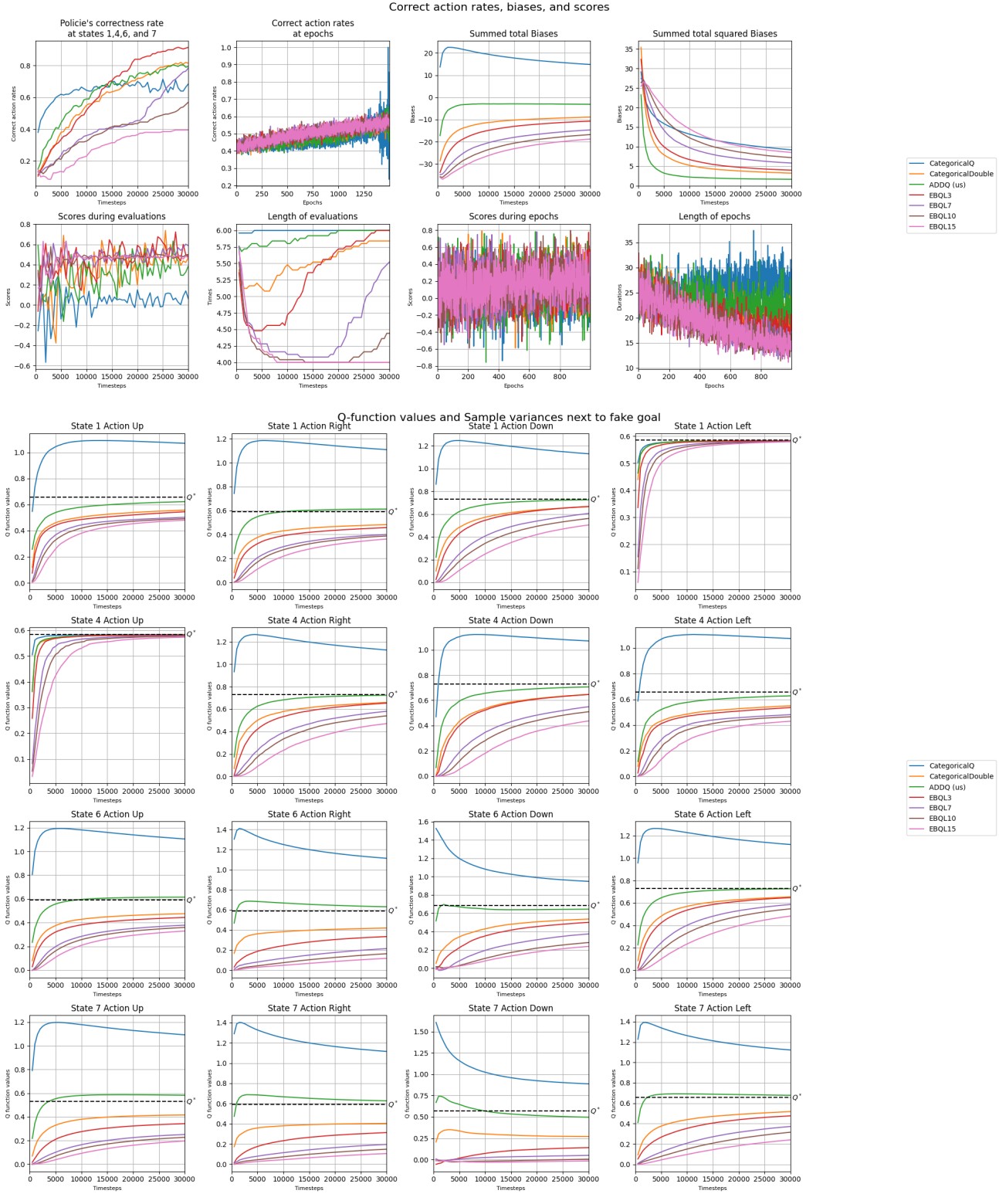

*Figure 11.* Comparison to EBQL

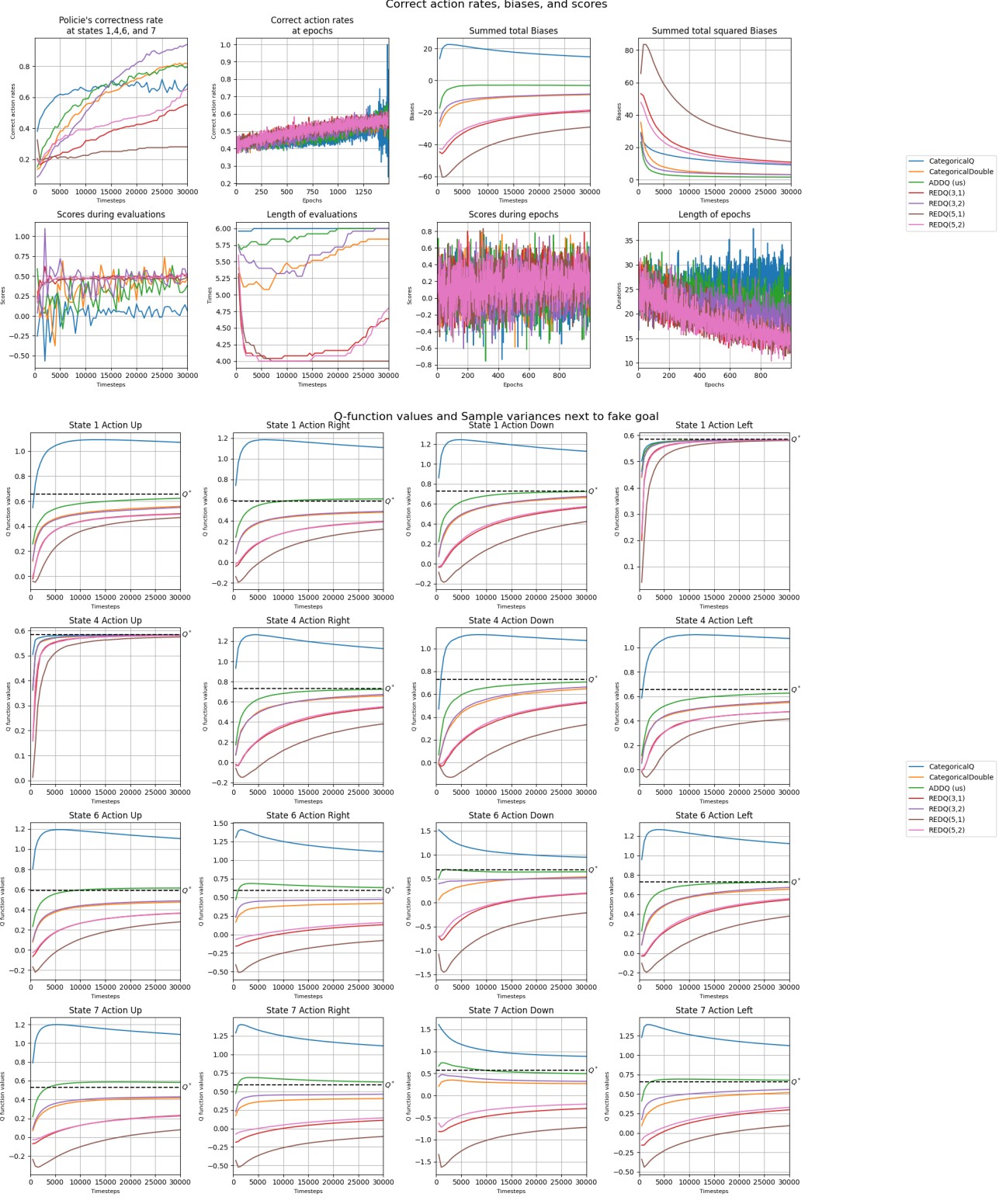

*Figure 12.* Comparison to REDQ.

# D. ADDQ adaptation for QRDQN - setup

To not double too much in the main text we moved the adaptation of ADDQ to the quantile setup (Dabney et al., 2018) to the appendix. In this section we explain how to adapt the ADDQ idea into QRDQN, experimental results are provided in the next section.

The categorical approach has multiple disadvantages, most notably rewards and the fixed atom positions must be compatible. The categorical algorithm was included in the main text to keep notation simple. For quantile distributional RL the return distributions are parametrized by

$$\mathcal{F}_{Q,m} = \Big\{ \sum_{i=1}^{m} \frac{1}{m} \delta_{\theta_i} : \theta_i \in \mathbb{R} \Big\}.$$

In contrast to the categorical setting the positions of the atoms are not fixed, but the weights of all atoms are set equal. The computation of the target is equal to the categorical version (except using the projection on $\mathcal{F}_{Q,m}$ instead of $\mathcal{F}_{C,m}$). The update step is a gradient step in computing the Wasserstein-projection on $\mathcal{F}_{Q,m}$ of the target distribution $\hat{\eta}$, that is a gradient step in the quantile Huber-loss minimization:

$$\min_{\hat{\theta}_1^A(s,a),\dots,\hat{\theta}_m^A(s,a)} \sum_{i=1}^{m} \mathbb{E}_{Z \sim \hat{\eta}}[\rho_{\tau_i}^{\kappa}(Z - \hat{\theta}_i^A(s,a))],$$

with quantile mid-points $\tau_i = \frac{2i-1}{2m}$ and

$$\rho_{\tau}^{\kappa}(u) = \begin{cases} |\tau - \mathbf{1}_{u<0}| \frac{1}{2} u^2 & : |u| \leq \kappa \\ |\tau - \mathbf{1}_{u<0}| \kappa(|u| - \frac{1}{2}\kappa) & : |u| > \kappa \end{cases}.$$

The quantile $Q$-learning modification of classical DQN (Mnih et al., 2015) is called QRDQN. Using the same network architecture as C51, QRDQN approximates return distributions using the quantile representation. Therefore the last layer outputs the $m$ quantile locations for each action. In the quantile setup we write $\eta_{\omega}(s,a) = \frac{1}{m} \sum_{i=1}^{m} \delta_{\theta_i(s,a;\omega)}$ with induced mean values $Q_{\omega}(s,a) = \frac{1}{m} \sum_{i=1}^{m} \theta_i(s,a;\omega)$. Given a sample transition $(s,a,r,s')$, the network parameters are updated via gradient descent with respect to the loss function

$$\mathcal{L}(\omega) = \frac{1}{m} \sum_{i,j=1}^{m} \rho_{\tau_i}^1(r + \gamma\theta_j(s',z^*;\bar{\omega}) - \theta_i(s,a;\omega)), \quad z^* = \operatorname{argmax}_{a'} Q_{\bar{\omega}}(s',a'), \tag{5}$$

and the quantile mid-points $\tau_i = \frac{2-1}{2m}$.

In what follows we turn three known double variants of DQN with overestimation reduction into QRDQN variants and compare them on several Arcade environments to our algorithms. We use double DQN (van Hasselt et al., 2015), a $Q$-learning adaptation of the clipping trick included in TD3 (Fujimoto et al., 2018), a quantile version of our ADDQ algorithm, and an additional variant of ADDQ. Using the Stable-Baselines3 framework [(Raffin et al., 2021)] there is very little that must be modified. Return distributions $\eta^A$ and $\eta^B$ are parametrized with two independently initialized neural networks denoted by $\omega^A$ and $\omega^B$. As in DQN we use delayed target networks, one for $A$, one for $B$, that are indicated with an additional bar. For each gradient step we simulate a vector of random variables with the same size as the batch size with each element determining which of the two estimators is being updated based on the respective transition with the same position in the batch. Accordingly, we use twice the batch size for these methods, so that on average per gradient step, the same number of transitions is used for each estimator, compared to the single-estimator case. The only difference in different algorithms is the target used to update the neural networks.

Similar to modifying C51 implementations we only modify the target return distributions $b_{r,\gamma} \# \eta_{\bar{\omega}}(s',z^*)$ for an appropriate action $z^*$. In the quantile setup those are given by the locations of their atoms:

$$\Gamma = \{r + \gamma\theta_j(s',z^*;\omega) : j = 1,\dots,m\}$$

We again use the compact $A/B$ notation to indicate how the update applies for $\Gamma^A$ and $\Gamma^B$.

**Double QRDQN:** $\Gamma^{A/B} = \eta_{\bar{\omega}^{B/A}}(s',z^*)$, where $z^* = \operatorname{argmax}_{a'} Q_{\bar{\omega}^{A/B}}(s',a')$.

**Clipped QRDQN:** $\Gamma^{A/B} = \eta_{\bar{\omega}^X}(s', z^*)$, where $z^* = \text{argmax}_{a'} Q_{\bar{\omega}^{A/B}}(s', a')$ and $X = \text{argmin}_{c \in \{A,B\}} Q_{\bar{\omega}^c}(s', z^*)$.

**ADDQ (us):** $\Gamma^{A/B} = \beta\eta_{\bar{\omega}^{A/B}}(s', z^*) + (1-\beta)\eta_{\bar{\omega}^{B/A}}(s', z^*)$, where $z^* = \text{argmax}_{a'} Q_{\bar{\omega}^{A/B}}(s', a')$. The weights $\beta = \beta(s, a; \omega)$ are essentially arbitrary and can depend on the current estimated return distributions $\eta^A$ and $\eta^B$. For the experiments we take the same choice from 1 as used for the tabular and the categorical settings.

Experimental results are presented in the next section.

# E. Deep reinforcement learning experiments

To ensure fair comparison we modified the algorithms C51 [(Bellemare et al., 2017)] and QRDQN [(Dabney et al., 2018)] within the Stable-Baselines3 framework [(Raffin et al., 2021)]. The C51 implementation has been added to this framework by adapting from the Dopamine framework [(Castro et al., 2018)] and the DQN Zoo [(Quan & Ostrovski, 2020)]. We run Atari environments from the Arcade Learning Environment [(Bellemare et al., 2013)] and MuJoCo [(Todorov et al., 2012)] environments both using the Gymnasium API [(Towers et al., 2023)]. We run the experiments via the RL Baselines3 Zoo [(Raffin, 2020)] training framework.

The experiments were executed on a HPC cluster with NVIDIA Tesla V100 and NVIDIA A100 GPUs. The replay buffer on Atari environments takes around 57GB of memory and less than 7 GB of memory for MuJoCo environments.

For the experiments the training has been interrupted every 50000 steps and 10 evaluation episodes on 10 evaluation environments without exploration have been performed. The plots below show the mean total reward (sum of all rewards) averaged over **10 seeds** with standard errors of seeds as the shaded regions. To improve visibility a rolling window of size 4 is applied. Atari runs took less than 48 hours for 20 million train steps and periodic evaluations and MuJoCo runs less than 36 for 10 million train steps (Humanoid) and periodic evaluations. Note that one timestep in the Atari environments corresponds to 4 frames, which are stacked together. This corresponds to repeating every action 4 times in the actual game. Therefore 20 million timesteps correspond to 80 million frames. Additionally, a small ablation study comprising of 10 seeds on one evaluation environment with some of the choices of beta detailed in Appendix C.2 has been conducted.

## E.1. Full experimental results for the categorical parametrization

As in (Bellemare et al., 2017) we use 51 atoms for all C51 variants.

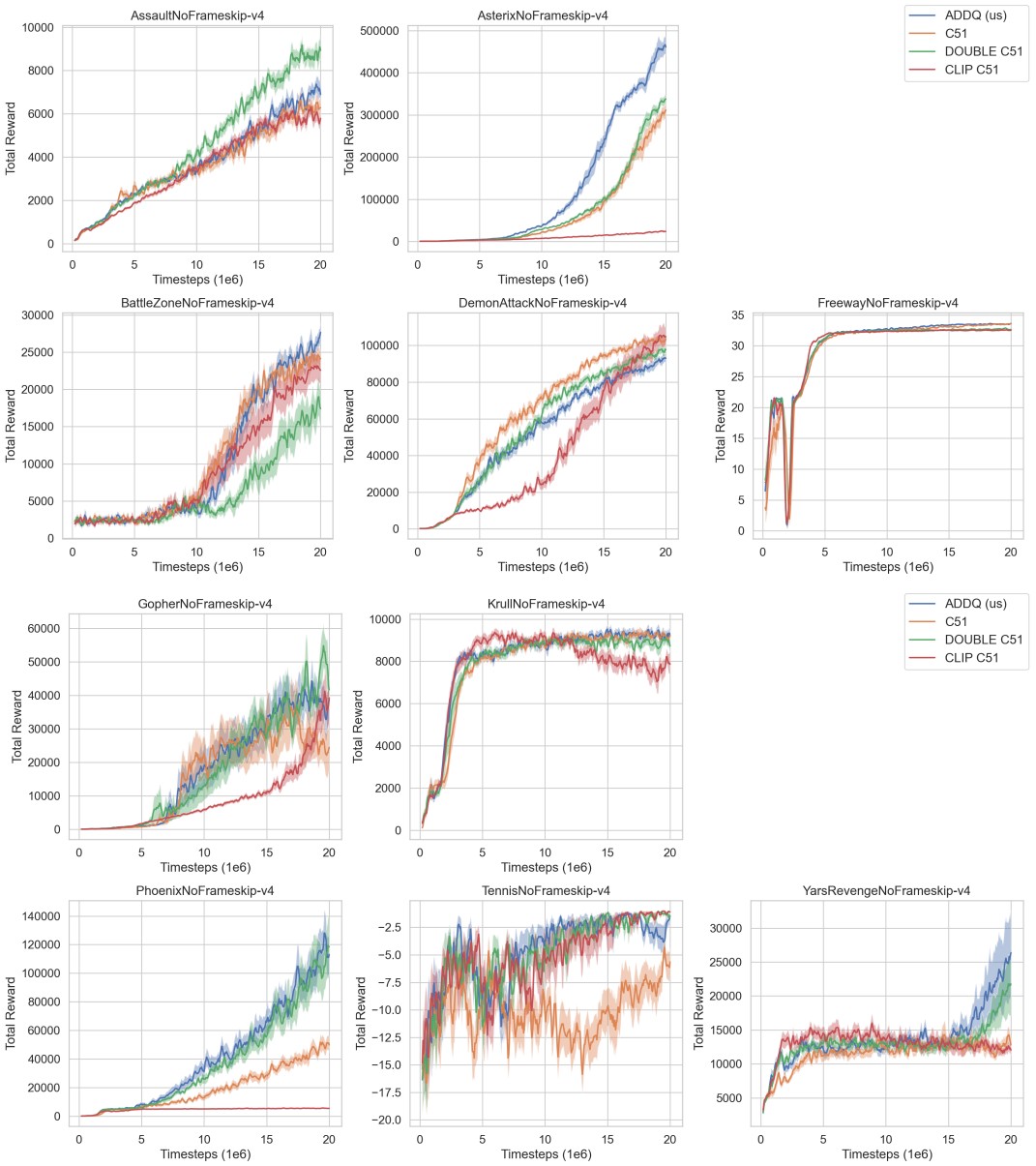

*Figure 13.* Learning curves on 10 Atari environments, averaged over 10 seeds

We additionally provide plots using the RLiable library ([Agarwal et al., 2021](#)).

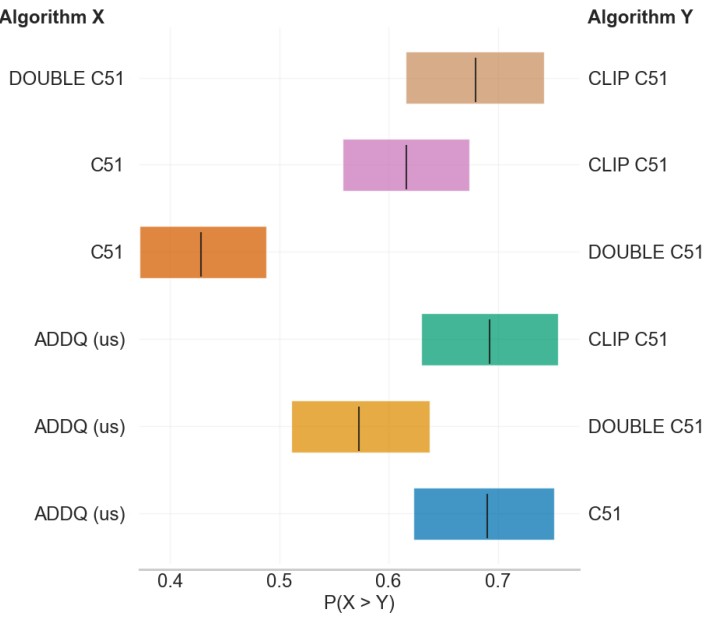

*Figure 14.* RLiable probability of improvement plot

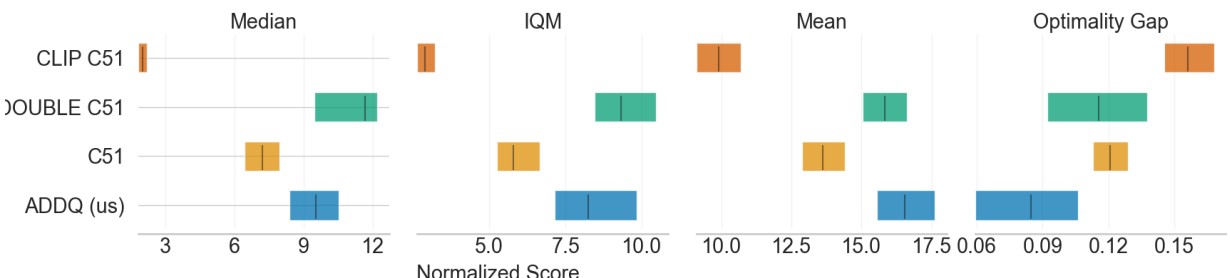

*Figure 15.* RLiable human normalized scores plot (based on ([Badia et al., 2020](#)))

### E.2. Full experimental results for the quantile parametrization

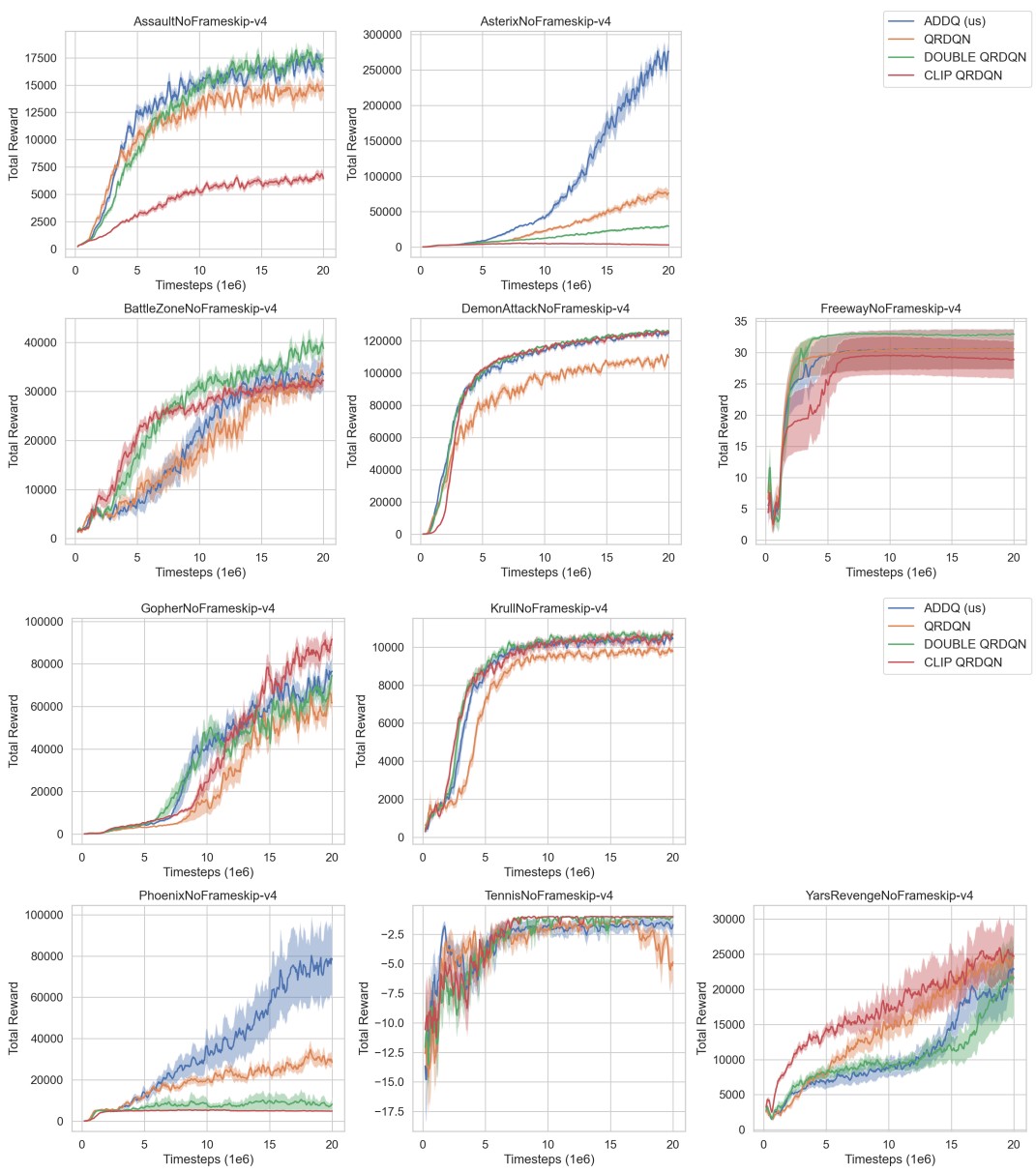

*Figure 16.* Learning curves on 10 Atari environments, averaged over 10 seeds

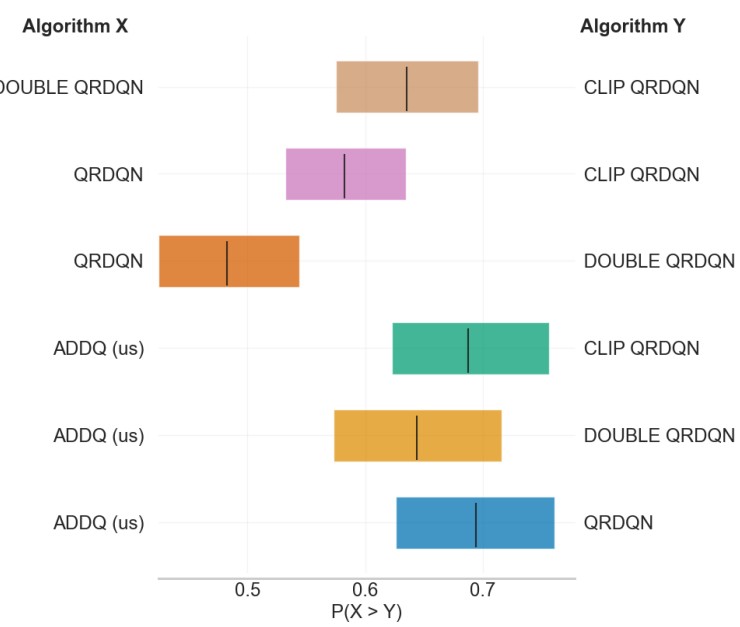

Figure 17. RLiable probability of improvement plot

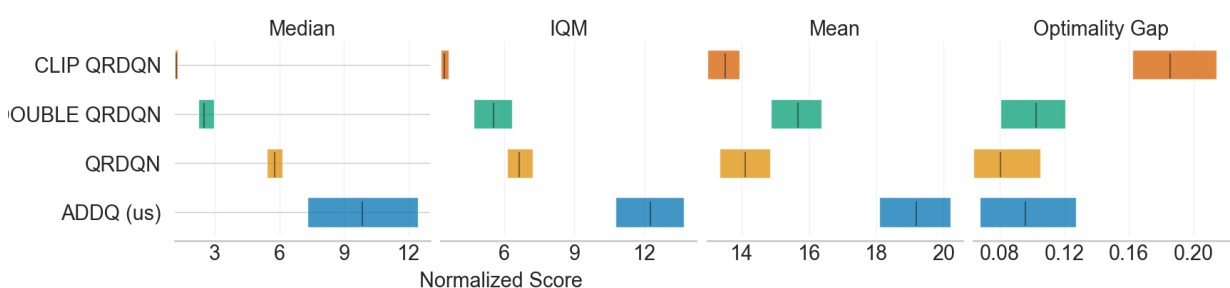

Figure 18. RLiable human normalized scores plot (based on (Badia et al., 2020))

## E.3. Full experimental results for the actor critic setting

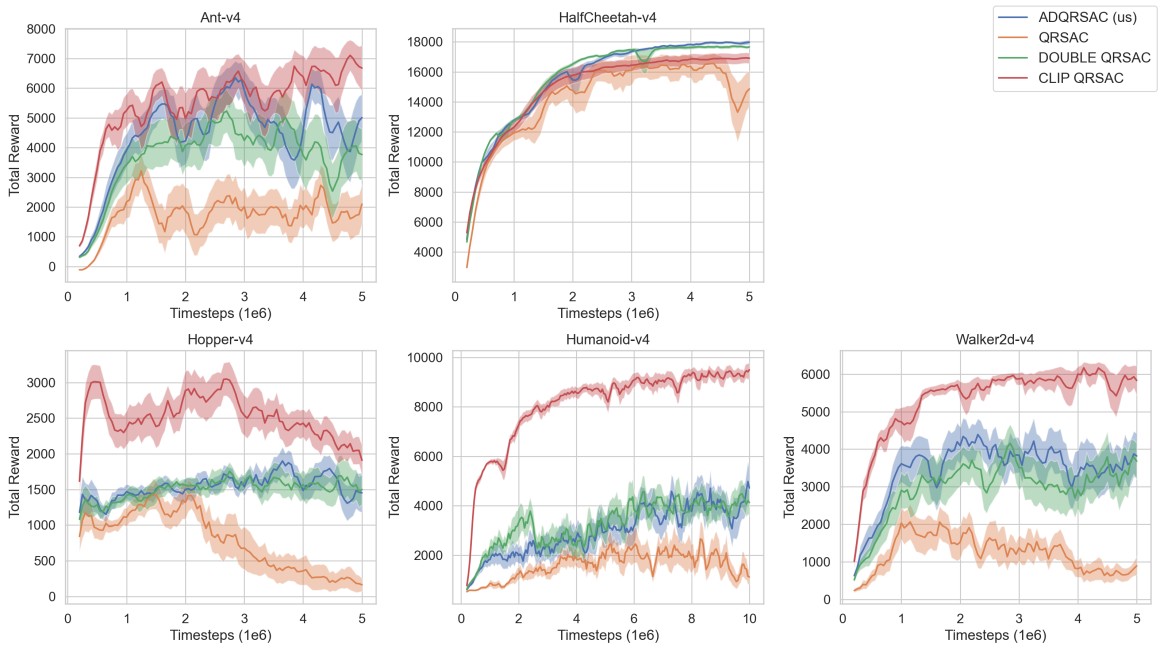

*Figure 19.* Learning curves on 5 MuJoCo environments, averaged over 10 seeds

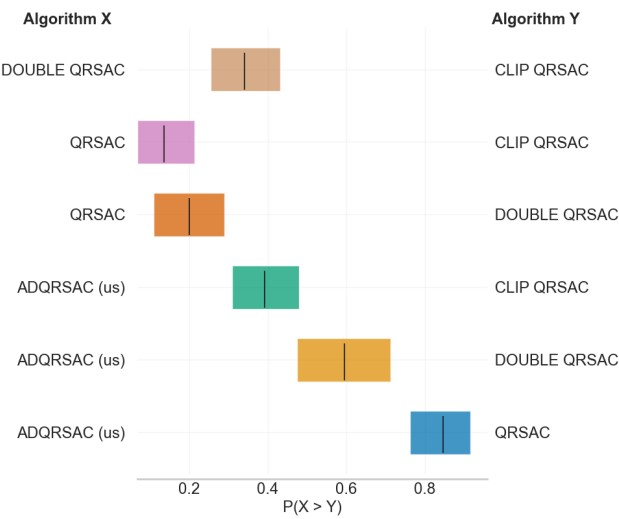

*Figure 20.* RLiable probability of improvement plot

*Figure 21.* RLiable normalized scores plot, based on highest and lowest performance on each environment

## F. Convergence proof in the tabular categorical setup

In this section we give a convergence proof for the adaptive distributional double-Q algorithm in the simplest setting, the categorical setting. The proof is based on known arguments from the literature and requires some modifications to work in our generality. Since many papers only sketched proofs we decided to spell out all details.

*Remark* F.1 (Notation and short recap). The Cramer distance $\ell_2$ for probability distributions $\nu, \nu' \in \mathbb{P}(\mathbb{R})$ is given by

$$\ell_2(\nu, \nu') = \left( \int_{\mathbb{R}} |F_\nu(z) - F_{\nu'}(z)|^2 \, dz \right)^{1/2}.$$

Following (Rowland et al., 2018; Bellemare et al., 2023) the supremum extension of a probability metric $d$ between two return distribution functions $\eta, \eta' \in \mathbb{P}^{\mathcal{S} \times \mathcal{A}}$ is denoted as

$$\bar{d}(\eta, \eta') = \sup_{s,a \in \mathcal{S} \times \mathcal{A}} d(\eta(s,a), \eta'(s,a)).$$

The iterates $\eta_{k+1} = \Pi_C \mathcal{T}^\pi \eta_k$ converge to the unique fixed point in $\mathcal{F}_{C,m}^{\mathcal{S} \times \mathcal{A}}$ with respect to $\bar{\ell}_2$ based on Banach's fixed point Theorem. This follows from the contraction property

$$\bar{\ell}_2(\Pi_C \mathcal{T}^\pi \eta, \Pi_C \mathcal{T}^\pi \eta') \leq \sqrt{\gamma} \bar{\ell}_2(\eta, \eta'), \tag{6}$$

[compare (Rowland et al., 2018; Bellemare et al., 2023)].

**Theorem F.2** (Convergence of adaptive distributional $Q$-learning in the categorical setting). *Given some initial return distribution functions $\eta_0^A, \eta_0^B$ supported within $[\theta_1, \theta_m]$, the induced Q-values, i.e. the expected values of the return distributions $(\eta_t^A), (\eta_t^B)$, recursively defined by Algorithm 1 converge almost surely towards $Q^*$ if the following conditions are satisfied:*

1. *the step sizes $\alpha_t(s,a)$ almost surely fulfill the Robbins-Monro conditions $\sum_{t=0}^{\infty} \alpha_t(s,a) = \infty$ and $\sum_{t=0}^{\infty} \alpha_t^2(s,a) < \infty$.*

2. *rewards are bounded in $[R_{min}, R_{max}]$ and $[\frac{R_{min}}{1-\gamma}, \frac{R_{max}}{1-\gamma}] \subseteq [\theta_1, \theta_m]$,*

3. *the choice of updating $\eta^A$ or $\eta^B$ is random and independent of all previous random variables*

4. *the sequences $(\beta_t^A)_{t \in \mathbb{N}}, (\beta_t^B)_{t \in \mathbb{N}}$ only depend on the past and fulfill $\lim_{t \to \infty} |\beta_t^A - \beta_t^B| = 0$ almost surely.*

*If additionally the MDP has a unique optimal policy $\pi^*$, then $(\eta_t^A), (\eta_t^B)$ converge almost surely in $\bar{\ell}_2$ to some limit $\eta_C^* \in \mathcal{F}_{C,m}$ and the greedy policy with respect to $\eta_C^*$ is the optimal policy.*

Note that the algorithm and proof uses $\beta_{t+1}^{A/B}(s,a)$ with index $t+1$ when updating $\eta_t^{A/B}$. This is to show that in general the parameter is allowed to depend on $S_{t+1}$ and the respective greedy action, i.e. it must only be $\mathcal{F}_{t+1}$ measurable. To portray this generality in the following we will only write $\beta_{t+1}^{A/B}$ without referencing a state-action pair.

The simplest way to guarantee the assumptions on the adaptive parameters $\beta^A$, $\beta^B$ to be satisfied is to chose them equal.

As in (van Hasselt, 2010), the proof is based on the following stochastic approximation result, see also (Bertsekas & Tsitsiklis, 1996), Proposition 4.5.

**Lemma F.3** ((Singh et al., 2000), Lemma 1). *Suppose $(\Omega, \mathcal{A}, \mathbb{P}, (\mathcal{F}_n))$ is a filtered probability space on which all appearing random variables are defined. Suppose that*

1. *a stochastic process $(F_n)_{n \in \mathbb{N}} \subset \mathbb{R}^d$ with the coordinates $F_{i,n}$ for $i = 1, \ldots, d$ such that $F_n$ is $\mathcal{F}_{n+1}$-measurable and for all $i = 1, \ldots, d$*

$$\|\mathbb{E}[F_n | \mathcal{F}_n]\|_\infty \leq \kappa \|X_n\|_\infty + c_n \quad \text{and} \quad \mathbb{V}[F_{i,n} | \mathcal{F}_n] \leq K(1 + \kappa \|X_n\|_\infty)^2 \quad n \geq 1,$$

*where $\kappa \in [0,1)$, an adapted, stochastic process $(c_n)_{n \in \mathbb{N}} \subset \mathbb{R}^+$ that converges to 0 almost surely and some constant $K > 0$.*

2. *the non-negative stochastic process* $(\alpha_n)_{n\in\mathbb{N}} \subset \mathbb{R}^d$, *with the coordinates* $\alpha_{i,n} \in [0,1]$ *for* $i = 1, \ldots, d$ *is adapted with*

$$\sum_{n=1}^{\infty} \alpha_{i,n} = \infty \quad \text{and} \quad \sum_{n=1}^{\infty} \alpha_{i,n}^2 < \infty \quad a.s..$$

*Then, for any* $\mathcal{F}_0$-*measurable initial condition* $X_0$ *the stochastic process* $(X_n)_{n\in\mathbb{N}} \subset \mathbb{R}^d$ *with coordinates* $X_{i,n}$ *for* $i = 1, \ldots, d$ *that is recursively defined by*

$$X_{i,n+1} = (1 - \alpha_{i,n})X_{i,n} + \alpha_{i,n}F_{i,n}, \quad n \in \mathbb{N},$$

*converges almost surely to zero.*

Furthermore, we follow (Rowland et al., 2018) by first showing the convergence of the mean-values to $Q^*$ and afterwards showing convergence of the return distribution functions, under the assumption of a unique optimal policy, by coupling it with policy evaluation. The convergence of the latter is easier to prove and we will do so at the end.

**Lemma F.4** (Adaptive Double Categorical Temporal Difference for Policy Evaluation). *Given some initial return distribution functions* $\eta_0^A, \eta_0^B$ *supported within* $[\theta_1, \theta_m]$ *and a stationary policy* $\pi \in \Pi_S$, *the return distribution functions* $(\eta_t^A), (\eta_t^B)$ *recursively defined by Algorithm 1, but with* $a^* \sim \pi(\cdot; S_{t+1})$ *instead, converge almost surely towards the unique fixed point* $\eta_C \in \mathbb{P}(\mathbb{R})^{\mathcal{S}\times\mathcal{A}}$ *of the operator* $\Pi_C \mathcal{T}^\pi$ *with respect to* $\bar{\ell}_2$, *if the following conditions are satisfied:*

1. *the step sizes* $\alpha_t(s,a)$ *fulfill the Robbins-Monro conditions:*
   - $\sum_{t=0}^{\infty} \alpha_t(s,a) = \infty$
   - $\sum_{t=0}^{\infty} \alpha_t^2(s,a) < \infty$,

2. *rewards are bounded in* $[R_{min}, R_{max}]$ *and* $[\frac{R_{min}}{1-\gamma}, \frac{R_{max}}{1-\gamma}] \subseteq [\theta_1, \theta_m]$,

3. *the choice of updating* $\eta^A$ *or* $\eta^B$ *is random and independent of all other previous random variables*

The above result is only relevant for the proof of Theorem F.2, as policy evaluation with a double estimator is not of interest. Note that convergence of categorical temporal difference for policy evaluation (in the single estimator case) has been proven in [(Rowland et al., 2018) Theorem 2 mimicking (Tsitsiklis, 1994) Theorem 2] and [(Bellemare et al., 2023) Theorem 6.12 applying (Tsitsiklis, 1994) Theorem 3 or (Bertsekas & Tsitsiklis, 1996) Proposition 4.5].

**Lemma F.5.** *Let* $(\alpha_t)_{t\in\mathbb{N}_0}$ *be a sequence fulfilling the Robbins-Monro conditions and* $(Y_t)_{t\in\mathbb{N}}$ *an iid sequence of Bernoulli(0.5) random variables, i.e.* $\mathbb{P}(Y_t = 1) = \mathbb{P}(Y_t = 0) = 0.5$ *for all* $t \in \mathbb{N}_0$. *Then* $(\alpha_t Y_t)_{t\in\mathbb{N}_0}$ *also fulfills the Robbins-Monro condition.*

*Proof.* The almost sure convergence of the summed squares is obviously fulfilled due to

$$\sum_{t=0}^{\infty} (\alpha_t Y_t)^2 \leq \sum_{t=0}^{\infty} \alpha_t^2 < \infty \quad \text{almost surely.}$$

Due to independence of each $Y_t$ with $\{Y_n | n \in \mathbb{N}_0, \ n \neq t\}$ as well as with $\alpha = (\alpha_t)_{t=0}^{\infty}$ we will consider a two stage experiment, where we first draw the sequence $\alpha = (\alpha_t)_{t=0}^{\infty}$ and then independently of this realization sample the *iid* sequence $Y = (Y_t)_{t=0}^{\infty}$. Due to the independence the joint measure of $\alpha$ and $Y$ is the product measure. Consider the product space $(\Omega, \mathcal{F}, \mathbb{P}) = (\Omega_\alpha \times \Omega_Y, \mathcal{F}_\alpha \otimes \mathcal{F}_Y, \mathbb{P}_\alpha^{\otimes\mathbb{N}} \otimes \mathbb{P}_Y^{\otimes\mathbb{N}})$ where $\Omega_\alpha, \Omega_Y = [0,1]^{\mathbb{N}}$, $\mathcal{F}_\alpha, \mathcal{F}_Y = \mathcal{B}([0,1])^{\otimes\mathbb{N}}$. Then, using that $\sum_{t=0}^{\infty} \alpha_t = \infty$ $\mathbb{P}_\alpha$-almost surely, we have

$$\mathbb{P}\Big(\sum_{t=0}^{\infty} \alpha_t Y_t = \infty\Big) = \int_{\Omega_\alpha} \mathbb{P}_Y\Big(\sum_{t=0}^{\infty} \alpha_t Y_t = \infty\Big) d\mathbb{P}_\alpha(\alpha)$$

$$= \int_{\{(\alpha_t)_{t=0}^{\infty} \in \Omega_\alpha : \sum_{t=0}^{\infty} \alpha_t = \infty\}} \mathbb{P}_Y\Big(\sum_{t=0}^{\infty} \alpha_t Y_t = \infty\Big) d\mathbb{P}_\alpha(\alpha)$$

$$\overset{(a)}{=} \int_{\{(\alpha_t)_{t=0}^{\infty} \in \Omega_\alpha : \sum_{t=0}^{\infty} \alpha_t = \infty\}} 1 \, d\mathbb{P}_\alpha(\alpha)$$

$$= 1,$$

where $(a)$ can be seen as follows. Consider any deterministic sequence $(b_t) \subseteq [0,1]$ fulfilling $\sum_{t=0}^{\infty} b_t = \infty$. Then

$$\infty = \sum_{t=0}^{\infty} b_t = \sum_{t=0}^{\infty} b_t Y_t + \sum_{t=0}^{\infty} b_t \mathbf{1}_{Y_t=0}.$$

Now notice that $A = \sum_{t=0}^{\infty} b_t Y_t$ and $B = \sum_{t=0}^{\infty} b_t \mathbf{1}_{Y_t=0}$ are identically distributed and since the sum of $A$ and $B$ is always infinity, almost surely either one of them is infinite. Given the identical distribution, we infer

$$\mathbb{P}_Y(\sum_{t=0}^{\infty} b_t Y_t = \infty) > 0.$$

But since $(b_t Y_t)$ is an independent sequence of random variables and the event that the infinite sum diverges is in the tail sigma algebra, the Kolmogorov 0-1 law yields:

$$\mathbb{P}_Y(\sum_{t=0}^{\infty} b_t Y_t = \infty) = 1.$$

$\square$

*Remark* F.6. As outlined in (Rowland et al., 2018), proof of Proposition 1, denoting by $\mathcal{M}(\mathbb{R})$ the space of all finite signed measures on $(\mathbb{R}, \mathcal{B}(\mathbb{R}))$, the subspace

$$\mathcal{M}_0(\mathbb{R}) := \{\nu \in \mathcal{M}(\mathbb{R}) | \nu(\mathbb{R}) = 0, \int_{\mathbb{R}} F_\nu(x)^2 dx < \infty\},$$

"where $F_\nu(x) = \nu([-\infty, x))$ for $x \in \mathbb{R}$, is isometrically isomorphic to a subspace of the Hilbert space $L^2(\mathbb{R})$ with inner product given by

$$\langle \nu_1, \nu_2 \rangle_{\ell_2} = \int_{\mathbb{R}} F_{\nu_1}(x) F_{\nu_2}(x) dx.\text{"}$$

Then the affine translation $\delta_0 + \mathcal{M}_0$ is also Hilbert space endowed with the same inner product. It contains probability measures $\nu \in \mathbb{P}(\mathbb{R})$ satisfying

$$\int_{-\infty}^{0} F_\nu(x)^2 \, dx < \infty \quad \text{and} \quad \int_{0}^{\infty} (1 - F_\nu(x))^2 \, dx < \infty.$$

To see this, consider $\mu = \nu - \delta_0$ fulfills $F_\mu(x) = F_\nu(x)$ for $x < 0$ and $F_\mu(x) = F_\nu(x) - 1$ for $x \geq 1$. Hence, $\mu \in \mathcal{M}_0$. The two conditions assure that the tails decay fast enough.

Note that the inner product induces a norm through $\|\nu\|_{\ell_2}^2 = \langle \nu, \nu \rangle$. And we have $\ell_2(\nu_1, \nu_2) = \|\nu_1 - \nu_2\|_{\ell_2}$. In the following proof, we will make use of the relationship

$$\ell_2^2(\nu_1 + \nu_2, \nu_1' + \nu_2') = \|\nu_1 - \nu_1'\|_{\ell_2}^2 + \|\nu_2 - \nu_2'\|_{\ell_2}^2 + 2\langle \nu_1 - \nu_1', \nu_2 - \nu_2' \rangle$$

holding by bilinearity of the inner product.

*Proof of Thoerem F.2.* **Step 1: Convergence of mean values to $Q^*$**

The proof mainly follows (Rowland et al., 2018) and (van Hasselt, 2010). Let the filtration be given by $\mathcal{F}_t = \sigma(\eta_0^A, \eta_0^B, s_0, a_0, \alpha_0, R_0, S_1, Y_1, \beta_1^A, \beta_1^B \dots, s_t, a_t, \alpha_t)$, where $(Y_n)_{n \in \mathbb{N}}$ is an iid sequence of *Bernoulli(0.5)* random variables, independent of all other appearing random variables, such that A is updated when $Y_{n+1} = 1$. Denote the expected values of the return-distributions by $Q_t^A(s,a) = \mathbb{E}_{R \sim \eta_t^A(s,a)}[R]$ and overloading notation, let us further write $\mathbb{E}[\nu]$ for the expected value $\mathbb{E}_{R \sim \nu}[R]$ of a probability distribution $\nu \in \mathbb{P}(\mathbb{R})$. We will first consider how the expected values evolve. Due to the symmetry of the updates it is sufficient to show convergence of $Q_t^A$ to $Q^*$. It is implied that $\alpha(s,a) = 0$ for $(s,a) \neq (s_t, a_t)$. Further, define

$$X_t(s_t, a_t) := Q_t^A(s_t, a_t) - Q^*(s_t, a_t)$$

$$F_t(s_t, a_t) := \mathbf{1}_{Y_{t+1}=1}\Big(R_t + \gamma(\beta_{t+1}^A Q_t^A(S_{t+1}, a^*) + (1 - \beta_{t+1}^A)Q_t^B(S_{t+1}, a^*)) - Q^*(s_t, a_t)\Big)$$

$$+ \mathbf{1}_{Y_{t+1}=0} X_t(s_t, a_t)$$

$$F_t(s, a) := 0 \text{ whenever } (s, a) \neq (s_t, a_t)$$

with $a^* = \arg\max_{a' \in \mathcal{A}_{S_{t+1}}} Q^A(S_{t+1}, a')$. According to [(Lyle et al., 2019) Proposition 1] projection $\Pi_C$ is mean-preserving, i.e $\mathbb{E}[\Pi_C \nu] = \mathbb{E}[\nu]$ for when $\nu$ is a distribution supported within $[\theta_1, \theta_m]$. This is the case for every $\hat{\eta}_*$ as in Algorithm 1, which can be seen as following. Assume $\eta_t^A(s_t, a_t), \eta_t^B(s_t, a_t) \in \mathcal{F}_{C,m}$. Then also

$$\nu = \beta_{t+1}^A \eta_t^A(S_{t+1}, a^*) + (1 - \beta_{t+1}^A)\eta_t^B(S_{t+1}, a^*)) \in \mathcal{F}_{C,m},$$

and suppose $\nu = \sum_{i=1}^m p_i \delta_{\theta_i}$ for some $p_i$. Then

$$\hat{\eta}_* := b_{R_t, \gamma} \# \nu = \sum_{i=1}^m p_i \delta_{R_t + \gamma \theta_i}.$$

But now

$$\theta_1 \leq \frac{R_{min}}{1 - \gamma} \leq \frac{R_{min}}{1 - \gamma} \leq \theta_m$$

(Assumption $(ii)$) guarantees that

$$\theta_1 \leq R_t + \gamma \theta_i \leq \theta_m \quad \forall\, i \in \{1, \ldots, m\}$$

and $\hat{\eta}_*$ is supported within $[\theta_1, \theta_m]$. Similarly for a realized transition with $(R_t, S_{t+1}) = (r_t, s_{t+1})$, we have for the expected value of the distribution

$$\mathbb{E}\left[b_{r_t, \gamma} \#\left(\beta_{t+1}^A \eta_t^A(s_{t+1}, a^*) + (1 - \beta_{t+1}^A)\eta_t^B(s_{t+1}, a^*))\right)\right]$$
$$= r_t + \gamma(\beta_{t+1}^A Q_t^A(s_{t+1}, a^*) + (1 - \beta_{t+1}^A)Q_t^B(s_{t+1}, a^*)).$$

Hence, the expected values of the return distributions $\eta_t^A$ subtracted by $Q^*$ indeed evolve as

$$X_{t+1}(s, a) = (1 - \alpha_t(s, a))X_t(s, a) + \alpha_t(s, a)F_t(s, a).$$

We now proceed similarly as in (van Hasselt, 2010) to show that the conditions of Lemma F.3 are satisfied.
We first show that $\mathbb{V}[F_t(s, a)|\mathcal{F}_t]$ is bounded for all $(s, a) \in \mathcal{S} \times \mathcal{A}$ and therefore satisfies $\mathbb{V}[F_t(s, a)|\mathcal{F}_t] \leq K(1 + \kappa\|X_t\|_\infty)$ as required. Since the rewards were assumed to be bounded there is an $\bar{R} > 0$ such that $|r|, |\theta_1|, |\theta_m| \leq \bar{R} \,\forall r \in \mathcal{R}$. Hence, we have

$$|F_t(s_t, a_t)| \leq |R_t + \gamma(\beta_{t+1}^A Q_t^A(S_{t+1}, a^*) + (1 - \beta_{t+1}^A)Q_t^B(S_{t+1}, a^*)) - Q^*(s_t, a_t)|$$
$$+ |X_t(s_t, a_t)|$$
$$\leq \bar{R} + 3\bar{R} + 2\frac{\bar{R}}{1 - \gamma}.$$

Next, we need to show that $\|\mathbb{E}[F_t \mid \mathcal{F}_t]\|_\infty \leq \kappa\|X_t\|_\infty + c_n$. Let us therefore decompose

$$F_t(s_t, a_t) = \mathbf{1}_{Y_{t+1}=1}\left(F_t^Q(s_t, a_t) + \gamma(1 - \beta_{t+1})(Q_t^B(S_{t+1}, a^*) - Q_t^A(S_{t+1}, a^*))\right)$$
$$+ \mathbf{1}_{Y_{t+1}=0}\alpha_t(s_t, a_t)X_t(s_t, a_t)$$

with $F_t^Q(s_t, a_t) := R_t + \gamma Q_t^A(S_{t+1}, a^*) - Q^*(s_t, a_t)$. This yields

$$|\mathbb{E}[\mathbf{1}_{Y_{t+1}=1}F_t^Q(s_t, a_t) + \mathbf{1}_{Y_{t+1}=0}\alpha_t(s_t, a_t)X_t(s_t, a_t)|\mathcal{F}_t]|$$
$$= |\frac{1}{2}\mathbb{E}[R_t + \gamma Q_t^A(S_{t+1}, a^*)] - Q^*(s_t, a_t) + \frac{1}{2}X_t(s_t, a_t)|$$
$$\leq |T^*Q^A(s_t, a_t) - T^*Q^*(s_t, a_t)| + |\frac{1}{2}X_t(s_t, a_t)|$$
$$\leq \gamma\|Q_t^A - Q^*\|_\infty + \frac{1}{2}\|X_t\|_\infty$$
$$= \underbrace{(\frac{1}{2}\gamma + \frac{1}{2})}_{<1}\|X_t\|_\infty,$$

since the Bellman optimality operator is a $\gamma$-contraction. Subsequently, it only remains to show that

$$c_t := |\mathbb{E}[\mathbf{1}_{Y_{t+1}=1}\gamma(1 - \beta_{t+1}^A)(Q_t^B(S_{t+1}, a^*) - Q_t^A(S_{t+1}, a^*)|\mathcal{F}_t]|$$

goes to zero almost surely. This is immediate if we verify that

$$X_t^{BA}(s, a) := Q_t^B(s, a) - Q_t^A(s, a)$$

goes to zero almost surely for all $(s, a) \in \mathcal{S} \times \mathcal{A}$ which will be achieved by another application of Lemma F.3. We infer that

$$
\begin{aligned}
&X_{n+1}^{BA}(s_n, a_n) \\
=&X_n^{BA}(s_n, a_n) + \alpha_n(s_n, a_n)\Big( \\
&\mathbf{1}_{Y_{n+1}=0}\Big(R_n + \gamma\big(\beta_{n+1}^B Q_n^B(S_{n+1}, b^*) + (1 - \beta_{n+1}^B)Q_n^A(S_{n+1}, b^*)\big) - Q_n^B(s_n, a_n)\Big) \\
&-\mathbf{1}_{Y_{n+1}=1}\Big(R_n + \gamma\big(\beta_{n+1}^A Q_n^A(S_{n+1}, a^*) + (1 - \beta_{n+1}^A)Q_n^B(S_{n+1}, a^*)\big) - Q_n^A(s_n, a_n)\Big) \\
&\Big) \\
=&(1 - \alpha_n(s_n, a_n))X_n^{BA}(s_n, a_n) + \alpha_n(s_n, a_n)\Big( \\
&\mathbf{1}_{Y_{n+1}=0}\Big(R_n + \gamma\big(\beta_{n+1}^B Q_n^B(S_{n+1}, b^*) + (1 - \beta_{n+1}^B)Q_n^A(S_{n+1}, b^*)\big)\Big) \\
&-\mathbf{1}_{Y_{n+1}=1}\Big(R_n + \gamma\big(\beta_{n+1}^A Q_n^A(S_{n+1}, a^*) + (1 - \beta_{n+1}^A)Q_n^B(S_{n+1}, a^*)\big)\Big) \\
&+\mathbf{1}_{Y_{n+1}=1}Q_n^B(s_n, a_n) - \mathbf{1}_{Y_{n+1}=0}Q_n^A(s_n, a_n)\Big) \\
=&(1 - \alpha_n(s_n, a_n))X_n^{BA}(s_n, a_n) + \alpha_n(s_n, a_n)\tilde{F}_n(s_n, a_n),
\end{aligned}
$$

with

$$
\begin{aligned}
\tilde{F}_n(s_n, a_n) =&\Big(\mathbf{1}_{Y_{n+1}=0}\Big(R_n + \gamma\big(\beta_{n+1}^B Q_n^B(S_{n+1}, b^*) + (1 - \beta_{n+1}^B)Q_n^A(S_{n+1}, b^*)\big)\Big) \\
&-\mathbf{1}_{Y_{n+1}=1}\Big(R_n + \gamma\big(\beta_{n+1}^A Q_n^A(S_{n+1}, a^*) + (1 - \beta_{n+1}^A)Q_n^B(S_{n+1}, a^*)\big)\Big) \\
&+\mathbf{1}_{Y_{n+1}=1}Q_n^B(s_n, a_n) - \mathbf{1}_{Y_{n+1}=0}Q_n^A(s_n, a_n)\Big).
\end{aligned}
$$

Now, using that $Q_n^B(s_n, a_n), Q_n^A(s_n, a_n), X_n^{BA}(s_n, a_n), \alpha_n(s_n, a_n)$ are $\mathcal{F}_n$-measurable and $Y_{n+1}$ is independent of $\mathcal{F}_n$, the conditional expectation satisfies

$$
\begin{aligned}
|\mathbb{E}[\tilde{F}_n(s_n, a_n) \mid \mathcal{F}_n]| =&\frac{1}{2}\gamma|\mathbb{E}[\beta_{n+1}^B Q_n^B(S_{n+1}, b^*) + (1 - \beta_{n+1}^B)Q_n^A(S_{n+1}, b^*) \\
&-\beta_{n+1}^A Q_n^A(S_{n+1}, a^*) - (1 - \beta_{n+1}^A)Q_n^B(S_{n+1}, a^*)|\mathcal{F}_n]| \\
&+\frac{1}{2}|Q_n^B(s_n, a_n) - Q_n^A(s_n, a_n)| \\
\leq&\frac{1}{2}\gamma\Big(\big|\mathbb{E}[\beta_{n+1}^B(Q_n^B(S_{n+1}, b^*) - Q_n^A(S_{n+1}, a^*))|\mathcal{F}_n]\big| \\
&+\big|\mathbb{E}[(1 - \beta_{n+1}^B)(Q_n^A(S_{n+1}, b^*) - Q_n^B(S_{n+1}, a^*))|\mathcal{F}_n]\big| \\
&+\big|\mathbb{E}[(\beta_{n+1}^B - \beta_{n+1}^A)Q_n^A(S_{n+1}, a^*)|\mathcal{F}_n]\big| \\
&+\big|\mathbb{E}[((1 - \beta_{n+1}^B) - (1 - \beta_{n+1}^A))Q_n^B(S_{n+1}, a^*)|\mathcal{F}_n]\big|\Big) \\
&+\frac{1}{2}\|X_n\|_\infty
\end{aligned}
$$

Now if it holds $\mathbb{E}[Q_n^B(S_{n+1}, b^*)|\mathcal{F}_n] \geq \mathbb{E}[Q_n^A(S_{n+1}, a^*)|\mathcal{F}_n]$, by definition of $a^*$ we have $Q_n^A(S_{n+1}, a^*) = \max_{a \in \mathcal{A}_{S_{n+1}}} Q_n^A(S_{n+1}, a) \geq Q_n^A(S_{n+1}, b^*)$ and therefore

$$\left|\mathbb{E}[Q_n^B(S_{n+1}, b^*) - Q_n^A(S_{n+1}, a^*)|\mathcal{F}_n]\right| = \mathbb{E}[Q_n^B(S_{n+1}, b^*) - Q_n^A(S_{n+1}, a^*)|\mathcal{F}_n]$$
$$\leq \mathbb{E}[Q_n^B(S_{n+1}, b^*) - Q_n^A(S_{n+1}, b^*)|\mathcal{F}_n] \leq \|X_n^{BA}\|_\infty.$$

Analogously, if $\mathbb{E}[Q_n^B(S_{n+1}, b^*)|\mathcal{F}_n] < \mathbb{E}[Q_n^A(S_{n+1}, a^*)|\mathcal{F}_n]$, then we have by definition of $b^*$

$$\left|\mathbb{E}[Q_n^B(S_{n+1}, b^*) - Q_n^A(S_{n+1}, a^*)|\mathcal{F}_n]\right| = \mathbb{E}[Q_n^A(S_{n+1}, a^*) - Q_n^B(S_{n+1}, b^*)|\mathcal{F}_n]$$
$$\leq \mathbb{E}[Q_n^A(S_{n+1}, a^*) - Q_n^B(S_{n+1}, a^*)|\mathcal{F}_n] \leq \|X_n^{BA}\|_\infty.$$

Similarly, by distinguishing cases, one shows that

$$\left|\mathbb{E}[Q_n^A(S_{n+1}, b^*) - Q_n^B(S_{n+1}, a^*)|\mathcal{F}_n]\right| \leq \|X_n^{BA}\|_\infty + \frac{1}{2}\|X_n^{BA}\|.$$

Combining the above yields

$$\left|\mathbb{E}[\tilde{F}_n(s_n, a_n) \mid \mathcal{F}_n]\right| \leq \frac{1}{2}\gamma(\beta_{n+1}^B + (1 - \beta_{n+1}^B))\|X_n^{BA}\|_\infty$$
$$+ \left|\gamma\mathbb{E}[(\beta_{n+1}^B - \beta_{n+1}^A)\underbrace{Q_n^A(S_{n+1}, a^*)}_{<\bar{R}<\infty}|\mathcal{F}_n]\right| + \left|\gamma\mathbb{E}[((1 - \beta_{n+1}^B) - (1 - \beta_{n+1}^A))\underbrace{Q_n^B(S_{n+1}, a^*)}_{<\bar{R}<\infty}|\mathcal{F}_n]\right|.$$

$$\underbrace{\qquad\qquad\qquad\qquad\qquad\qquad\qquad\qquad\qquad\qquad\qquad\qquad\qquad\qquad}_{:=\tilde{c}_n \to 0, \text{ since } |\beta_n^A - \beta_n^B| \text{ converges to 0 for } n \to \infty \text{ due to } (iv)}$$

Hence, we invoke Lemma F.3 to obtain convergence of $X_t^{BA}$ and thus with another application of Lemma F.3, $X_t(s, a)$ converges to zero which finally implies $Q_t^A(s, a)$ (and also $Q^B(s, a)$) converges to $Q^*(s, a)$ almost surely for every $(s, a) \in \mathcal{S} \times \mathcal{A}$.

Since $\mathcal{S}, \mathcal{A}$ are finite, for every $\varepsilon > 0$, there exists a random variable $N > 0$ such that for all $t > N$, we have

$$\max_{z \in \{A, B\}} \|Q_t^z - Q^*\|_\infty < \varepsilon \quad \text{almost surely.}$$

**Step 2: Convergence of return distributions**
Suppose the MDP has a unique optimal policy $\pi^*$. Now following (Rowland et al., 2018), we take $\varepsilon$ to be half the minimum action gap for the optimal action-value function $Q^* = Q^{\pi^*}$, i.e.

$$\varepsilon = \frac{1}{2} \min_{s \in \mathcal{S}}(Q^{\pi^*}(s, \pi^*(s) - \max_{a \neq} Q^{\pi^*}(s, a))$$

which is greater than zero by assumption $(v)$. Hence, denoting the action of the deterministic optimal policy in a certain state $s$ by $\pi^*(s)$, we get

$$\max_a Q_t^A(s, a) = \max_a Q_t^B(s, a) = \pi^*(s)$$

for all $t > N$. For some initial condition $\tilde{\eta}_0 \in \mathcal{F}_{C,m}^\mathcal{S}$, let now $\tilde{\eta}_k$ be the iterates created by a double categorical policy evaluation algorithm for the optimal policy $\pi^*$, i.e.

$$\tilde{\eta}_{k+1}^A(s_k, a_k) = (1 - \mathbf{1}_{Y_{k+1}=1}\alpha_k(s_k, a_k))\tilde{\eta}_k(s_k, a_k)$$
$$+ \mathbf{1}_{Y_{k+1}=1}\alpha_k(s_k, a_k)\Pi_C\left(b_{R_k,\gamma}\#\left(\beta_{k+1}^A\tilde{\eta}_k^A(S_{k+1}, \pi^*(S_{k+1}))\right.\right.$$
$$\left.\left. + (1 - \beta_{k+1}^A)\tilde{\eta}_k^B(S_{k+1}, \pi^*(S_{k+1}))\right)\right)$$
$$\tilde{\eta}_{k+1}^A(s, a) = \tilde{\eta}_k^A(s, a) \text{ for } (s, a) \neq (s_k, a_k).$$

and analogously for $\tilde{\eta}^B$. Note that the appearing $\mathbb{Y}_k, \alpha_k, \beta_k^A, \beta_k^B$ are chosen to be the same as in the control case above. Then $\tilde{\eta}^A, \tilde{\eta}^B$ converges almost surely to the unique fixed point $\eta_C^*$ of the projected operator $\Pi_C \mathcal{T}^{\pi^*}$ with respect to $\bar{\ell}_2$ by

**Lemma F.4.** Similarly to (Rowland et al., 2018), we now proceed by a coupling argument. Denote by $\pi_k^A, \pi_k^B$ any greedy selection rule with respect to $\eta_k^A$ and $\eta_k^B$ and $A_k = \{\pi_k^A = \pi_k^B = \pi^* \text{ for all } n \geq k\}$. Then $A_k \subseteq A_{k+1}$ and by the above explanation we have $\mathbb{P}(A_k) \nearrow 1$. Additionally, let $B$ be the event of probability 1 for which the (double) policy evaluation algorithm converges. Now on the event $B \cup A_k$, we have

$$\bar{\ell}_2^2(\tilde{\eta}_n^A, \eta_C^*) \to 0 \quad \text{and} \quad \bar{\ell}_2^2(\tilde{\eta}_n^B, \eta_C^*) \to 0.$$

Then by the triangle inequality it suffices to show $\bar{\ell}_2(\eta_n^A, \tilde{\eta}_n^A) \to 0$ and $\bar{\ell}_2(\eta_n^B, \tilde{\eta}_n^B) \to 0$ on this event too, since then the Theorem follows by $\mathbb{P}(B \cup A_k) \nearrow 1$.

To prove this we will again apply Lemma F.3. This time with $d = 2 \cdot |\mathcal{S}||\mathcal{A}|$, where we identify

$$X_n := \begin{bmatrix} \ell_2^2(\eta_n^A, \tilde{\eta}_n^A) \\ \ell_2^2(\eta_n^B, \tilde{\eta}_n^B) \end{bmatrix} \in \mathbb{R}^{2|\mathcal{S}||\mathcal{A}|}.$$

Additionally, we expand the filtration by $\tilde{\mathcal{F}}_n = \sigma(\mathcal{F}_n, Y_{n+1})$ and define $\tilde{\alpha}_n^A(s, a) = \alpha_n(s, a)\mathbf{1}_{Y_{n+1}=1}$ and $\tilde{\alpha}_n^B(s, a) = \alpha_n(s, a)\mathbf{1}_{Y_{n+1}=0}$. By Lemma F.5 these steps-size sequences still fulfill the Robbins-Monro conditions. Then, writing

$$\nu^A = \beta_{n+1}^A \eta_n^A(S_{n+1}, \pi^*(S_{n+1})) + (1 - \beta_{n+1}^A)\eta_n^B(S_{n+1}, \pi^*(S_{n+1}))$$
$$\tilde{\nu}^A = \beta_{n+1}^A \tilde{\eta}_n^A(S_{n+1}, \pi^*(S_{n+1})) + (1 - \beta_{n+1}^A)\tilde{\eta}_n^B(S_{n+1}, \pi^*(S_{n+1}))$$

for short, for $n \geq k$, on $A_k$ we have

$$\ell_2^2(\eta_{n+1}^A(s_n, a_n), \tilde{\eta}_{n+1}^A(s_n, a_n))$$
$$= (1 - \tilde{\alpha}_n^A(s_n, a_n))^2 \|\eta_n^A(s_n, a_n) - \tilde{\eta}_n^A(s_n, a_n)\|_{\ell_2}^2$$
$$+ \tilde{\alpha}_n^A(s_n, a_n)^2 \|\Pi_C(b_{R_n,\gamma}\#\nu^A) - \Pi_C(b_{R_n,\gamma}\#\tilde{\nu}^A)\|_{\ell_2}^2$$
$$+ (1 - \tilde{\alpha}_n^A(s_n, a_n))\tilde{\alpha}_n^A(s_n, a_n)2\langle\eta_n^A(s_n, a_n) - \tilde{\eta}_n^A(s_n, a_n), \Pi_C(b_{R_n,\gamma}\#\nu^A) - \Pi_C(b_{R_n,\gamma}\#\tilde{\nu}^A)\rangle_{\ell_2}.$$

This can be rewritten in terms of Lemma F.3 as

$$X_{n+1}^A(s_n, a_n) = (1 - \zeta_n^A(s_n, a_n))X_n^A(s_n, a_n) + \zeta_n^A(s_n, a_n)F_n^A(s_n, a_n)$$

with $\zeta_n^A(s_n, a_n) = 2\tilde{\alpha}_n^A(s_n, a_n) - \tilde{\alpha}_n^A(s_n, a_n)^2$ and

$$F_n^A(s_n, a_n) = \frac{1}{\zeta_n^A(s_n, a_n)}(\tilde{\alpha}_n^A(s_n, a_n)^2\|\Pi_C(b_{R_n,\gamma}\#\nu^A) - \Pi_C(b_{R_n,\gamma}\#\tilde{\nu}^A)\|_{\ell_2}^2$$
$$+ (1 - \tilde{\alpha}_n^A(s_n, a_n))\tilde{\alpha}_n^A(s_n, a_n)2\langle\eta_n^A(s_n, a_n) - \tilde{\eta}_n^A(s_n, a_n),$$
$$\Pi_C(b_{R_n,\gamma}\#\nu^A) - \Pi_C(b_{R_n,\gamma}\#\tilde{\nu}^A)\rangle_{\ell_2})$$

and $F_n^A(s, a) = 0$ if $(s, a) \neq (s_n, a_n)$. It is mentioned that $\zeta_n^A(s_n, a_n) > 0$. Notice that,

$$\sum_{n=1}^{\infty}\zeta_n^A(s_n, a_n) = \sum_{n=1}^{\infty}(2\tilde{\alpha}_n^A(s_n, a_n) - \tilde{\alpha}_n^A(s_n, a_n)^2) = \infty \quad a.s.$$
$$\sum_{n=1}^{\infty}\zeta_n^A(s_n, a_n)^2 = \sum_{n=1}^{\infty}4\tilde{\alpha}_n^A(s_n, a_n)^2 - 4\tilde{\alpha}_n^A(s_n, a_n)^3 + \tilde{\alpha}_n^A(s_n, a_n)^2 < \infty \quad a.s.$$

(7)

Finally we have

$$
\begin{aligned}
|F_n^A(s_n, a_n)| \leq & \frac{1}{\zeta_n^A(s_n, a_n)} (\tilde{\alpha}_n^A(s_n, a_n)^2 \gamma \bar{\ell}_2^2(\beta_{n+1}^A \eta_n^A + (1 - \beta_{n+1}^A)\eta_n^B, \beta_{n+1}^A \tilde{\eta}_n^A + (1 - \beta_{n+1}^A)\tilde{\eta}_n^B) \\
& + (1 - \tilde{\alpha}_n^A(s_n, a_n))\tilde{\alpha}_n^A(s_n, a_n)2\sqrt{\gamma}|\langle \eta_n^A - \tilde{\eta}_n^A, \\
& \quad \beta_n^A \eta_n^A + (1 - \beta_n^A)\eta_n^B - \beta_n^A \tilde{\eta}_n^A - (1 - \beta_n^A)\tilde{\eta}_n^B \rangle_{\bar{\ell}_2}|) \\
\leq & \frac{1}{\zeta_n^A(s_n, a_n)} (\tilde{\alpha}_n^A(s_n, a_n)^2 \gamma \max_{z \in \{A,B\}} \bar{\ell}_2^2(\eta_n^z, \tilde{\eta}_n^z) \\
& + (1 - \tilde{\alpha}_n^A(s_n, a_n))\tilde{\alpha}_n^A(s_n, a_n)2\sqrt{\gamma} \max_{z \in \{A,B\}} \bar{\ell}_2^2(\eta_n^z, \tilde{\eta}_n^z)) \\
= & \frac{\tilde{\alpha}_n^A(s_n, a_n)^2 \gamma + (1 - \tilde{\alpha}_n^A(s_n, a_n))\tilde{\alpha}_n^A(s_n, a_n)2\sqrt{\gamma}}{2\tilde{\alpha}_n^A(s_n, a_n) - \tilde{\alpha}_n^A(s_n, a_n)^2} \max_{z \in \{A,B\}} \bar{\ell}_2^2(\eta_n^z, \tilde{\eta}_n^z) \\
\leq & \frac{(2\tilde{\alpha}_n^A(s_n, a_n) - \tilde{\alpha}_n^A(s_n, a_n)^2)\sqrt{\gamma}}{2\tilde{\alpha}_n^A(s_n, a_n) - \tilde{\alpha}_n^A(s_n, a_n)^2} \max_{z \in \{A,B\}} \bar{\ell}_2^2(\eta_n^z, \tilde{\eta}_n^z) \\
\leq & \sqrt{\gamma} \max_{z \in \{A,B\}} \bar{\ell}_2^2(\eta_n^z, \tilde{\eta}_n^z) = \sqrt{\gamma}\|X_n\|_\infty
\end{aligned}
$$

where we used regularity and 1/2-homogeneity of the $\ell_2$ metric as described in [(Bellemare et al., 2023) Section 4.6] as well as that $\Pi_C$ is a non-expansion in $\ell_2$ and

$$
\begin{aligned}
|\langle u, \beta u + (1 - \beta)v \rangle| = \beta\langle u, u \rangle + (1 - \beta)|\langle u, v \rangle| \leq \beta \max(\|u\|^2, \|v\|^2) + (1 - \beta)\|u\|\|v\| \\
\leq \max(\|u\|^2, \|v\|^2)
\end{aligned}
$$

by the Cauchy-Schwarz inequality. Further, by the above the Variance also fulfills

$$
\begin{aligned}
\mathbb{V}[F_n^A(s_n, a_n)|\tilde{\mathcal{F}}_n] = \mathbb{E}[F_n^A(s_n, a_n)^2|\mathcal{F}_n] - \mathbb{E}[F_n^A(s_n, a_n)|\tilde{\mathcal{F}}_n]^2 \\
\leq 2(\sqrt{\gamma} \max_{z \in \{A,B\}} \bar{\ell}_2^2(\eta_n^z, \tilde{\eta}_n^z))^2 \\
\leq 2\gamma \sup_{\eta, \eta \in \mathcal{F}_{C,m}^S} \bar{\ell}_2^4(\eta, \eta') < \infty.
\end{aligned}
$$

Therefore, by Lemma F.3 we obtain convergence $\bar{\ell}_2(\eta_n^A, \tilde{\eta}_n^A) \to 0$ and $\bar{\ell}_2(\eta_n^B, \tilde{\eta}_n^B) \to 0$ on $A_k$. As already described above, this results in

$$
\bar{\ell}_2(\eta_n^A, \eta_C^*) \to 0 \quad \text{and} \quad \bar{\ell}_2(\eta_n^B, \eta_C^*) \to 0 \quad \text{almost surely.}
$$

$\square$

*Proof of Lemma F.4.* Let the filtration be given by $\mathcal{F}_t = \sigma(\eta_0^A, \eta_0^B, s_0, a_0, \alpha_0, R_0, S_1, Y_1, \beta_1^A, \beta_1^B \ldots, s_t, a_t, \alpha_t, Y_{t+1})$, where $(Y_n)_{n \in \mathbb{N}}$ is an iid sequence of *Bernoulli(0.5)* random variables, independent of all other appearing random variables, such that A is updated when $Y_{n+1} = 1$. To clarify, abbreviating

$$
\begin{aligned}
\nu^A &= \beta_{t+1}^A \eta_t^A(S_{t+1}, A_{t+1}) + (1 - \beta_{t+1}^A)\eta_t^B(S_{t+1}, A_{t+1}) \\
\nu^B &= \beta_{t+1}^B \eta_t^B(S_{t+1}, A_{t+1}) + (1 - \beta_{t+1}^B)\eta_t^A(S_{t+1}, A_{t+1}) \quad \text{where} \\
A_{t+1} &\sim \pi(\cdot; S_{t+1}),
\end{aligned}
$$

we are confronted with the updates

$$
\begin{aligned}
\eta_{t+1}^A(s, a) &= \eta_t^A(s, a) + \alpha_t(s, a)\mathbf{1}_{Y_{t+1}=1}(\Pi_C(b_{R_t, \gamma}\#\nu^A) - \eta_t^A(s, a)) \\
\eta_{t+1}^B(s, a) &= \eta_t^B(s, a) + \alpha_t(s, a)\mathbf{1}_{Y_{t+1}=0}(\Pi_C(b_{R_t, \gamma}\#\nu^B) - \eta_t^B(s, a)).
\end{aligned}
$$

As in the proof above, define $\tilde{\alpha}_n^A(s, a) = \alpha_n(s, a)\mathbf{1}_{Y_{n+1}=1}$ and $\tilde{\alpha}_n^B(s, a) = \alpha_n(s, a)\mathbf{1}_{Y_{n+1}=0}$. By Lemma F.5 these steps-size sequences still fulfill the Robbins-Monro conditions. Also note that as in step 2 of the proof of Theorem F.2, $Y_{t+1}$ is

$\mathcal{F}_t$-measurable and hence so is $\tilde{\alpha}_t^{A/B}$. In order to align this with Lemma F.3, we rewrite

$$
\begin{aligned}
X_{t+1}^A(s,a) &= \ell_2^2(\eta_{t+1}^A(s,a), \eta_C(s,a)) \\
&= (1 - \tilde{\alpha}_t^A(s,a))^2 \|\eta_t^A(s,a) - \eta_C(s,a)\|_{\ell_2}^2 \\
&\quad + \tilde{\alpha}_t^A(s,a)^2 \|\Pi_C(b_{R_t,\gamma}\#\nu^A) - \eta_C(s,a)\|_{\ell_2}^2 \\
&\quad + (1 - \tilde{\alpha}_t^A(s,a))\tilde{\alpha}_t^A(s,a)2\langle\eta_t^A(s,a) - \eta_C(s,a), \Pi_C(b_{R_t,\gamma}\#\nu^A) - \eta_C(s,a)\rangle_{\ell_2} \\
&= (1 - \zeta_t^A(s,a))X_t^A(s,a) + \zeta_t^A(s,a)F_t^A(s,a)
\end{aligned}
$$

with $\zeta_t^A(s,a) = 2\tilde{\alpha}_t^A(s,a) - \tilde{\alpha}_t^A(s,a)^2$,

$$
X_t := \begin{bmatrix} \ell_2^2(\eta_t^A, \eta_C) \\ \ell_2^2(\eta_t^B, \eta_C) \end{bmatrix} \in \mathbb{R}^{2|\mathcal{S}||\mathcal{A}|}
$$

and

$$
\begin{aligned}
F_t^A(s,a) &= \frac{1}{\zeta_t^A(s,a)}\mathbf{1}_{\tilde{\alpha}_t^A(s,a)>0}(\tilde{\alpha}_t^A(s,a)^2\ell_2^2(\Pi_C(b_{R_t,\gamma}\#\nu^A), \eta_C(s,a)) \\
&\quad + (1 - \tilde{\alpha}_t^A(s,a))\tilde{\alpha}_t^A(s,a)2\langle\eta_t^A(s,a) - \eta_C(s,a), \Pi_C(b_{R_t,\gamma}\#\nu^A) - \eta_C(s,a)\rangle_{\ell_2}).
\end{aligned}
$$

As in Equation (7), the sequence $\zeta_t^A(s,a)$ fulfills the Robbins-Monro condition. Additionally, note that there exists $K > 0$, such that $\ell_2^2(\Pi_C(b_{R_t,\gamma}\#\nu^A), \eta_C(s,a)) < K$ independent of $s,a,t$. Further, observe that

$$
c_t := \max_{z\in\{A,B\}} \frac{1}{\zeta_t^z(s,a)}\mathbf{1}_{\tilde{\alpha}_t^z(s,a)>0}\tilde{\alpha}_t^z(s,a)^2 K \to 0 \text{ for } t \to \infty \text{ almost surely.}
$$

We use that $\Pi_C$ is mean-preserving [(Lyle et al., 2019) Proposition 1] for discrete distributions supported within $[\theta_1, \theta_m]$, which is satisfied by $b_{R_t,\gamma}\#\nu^A$, due to Assumption $(ii)$ and $\nu^A \in \mathcal{F}_m$. Together with the fact that $\Pi_C\mathcal{T}^\pi$ is a $\sqrt{\gamma}$-contraction with respect to $\bar{\ell}_2$ and the Cauchy-Schwarz inequality, we have

$$
\begin{aligned}
&|\mathbb{E}[\langle\eta_t^A(s,a) - \eta_C(s,a), \Pi_C(b_{R_t,\gamma}\#\nu^A) - \eta_C(s,a)\rangle_{\ell_2}|\mathcal{F}_t]| \\
&= |\langle\eta_t^A(s,a) - \eta_C(s,a), \mathbb{E}[\Pi_C(b_{R_t,\gamma}\#\nu^A)|\mathcal{F}_t] - \eta_C(s,a)\rangle_{\ell_2}| \\
&= |\langle\eta_t^A(s,a) - \eta_C(s,a), \mathbb{E}[b_{R_t,\gamma}\#\nu^A|\mathcal{F}_t] - \eta_C(s,a)\rangle_{\ell_2}| \\
&= |\langle\eta_t^A(s,a) - \eta_C(s,a), \Pi_C\mathcal{T}^\pi(\beta_{t+1}^A\eta_t^A + (1 - \beta_{t+1}^A)\eta_t^B)(s,a) - (\Pi_C\mathcal{T}^\pi\eta_C)(s,a)\rangle_{\ell_2}| \\
&\leq \sqrt{\gamma}|\langle\eta_t^A - \eta_C, (\beta_{t+1}^A\eta_t^A + (1 - \beta_{t+1}^A)\eta_t^B) - \eta_C\rangle_{\bar{\ell}_2}| \\
&\leq \sqrt{\gamma}(\beta_{t+1}^A\bar{\ell}_2^2(\eta_t^A, \eta_C) + (1 - \beta_{t+1}^A)|\langle\eta_t^A - \eta_C, \eta_t^B - \eta_C\rangle_{\bar{\ell}_2}|) \\
&\leq \sqrt{\gamma}(\beta_{t+1}^A \max_{z\in\{A,B\}}\bar{\ell}_2^2(\eta_t^z, \eta_C) + (1 - \beta_{t+1}^A)\|\eta_t^A - \eta_C\|_{\bar{\ell}_2}\|\eta_t^B - \eta_C\|_{\bar{\ell}_2}|) \\
&\leq \sqrt{\gamma} \max_{z\in\{A,B\}}\bar{\ell}_2^2(\eta_t^z, \eta_C) \\
&= \sqrt{\gamma}\|X_t\|_\infty.
\end{aligned}
$$

In total, this yields

$$
\begin{aligned}
&|\mathbb{E}[F_t^A(s,a)|\mathcal{F}_t]| \\
&\leq \frac{1}{\zeta_t^A(s,a)}\mathbf{1}_{\tilde{\alpha}_t^A(s,a)>0}\tilde{\alpha}_t^A(s,a)^2 K + \frac{1}{\zeta_t^A(s,a)}\mathbf{1}_{\tilde{\alpha}_t^A(s,a)>0}(1 - \tilde{\alpha}_t^A(s,a))\tilde{\alpha}_t^A(s,a)2\sqrt{\gamma}\|X_t\|_\infty. \\
&\leq c_t + \sqrt{\gamma}\|X_t\|_\infty.
\end{aligned}
$$

Since $\bar{\ell}_2(\eta, \eta') < K$ for every $\eta, \eta' \in \mathcal{F}_{C,m}^{\mathcal{S}\times\mathcal{A}}$ some $K > 0$, the conditional variance $\mathbb{V}[F_t^A|\mathcal{F}_t]$ can be bounded uniformly in $t$.

Therefore, the requirements of Lemma F.3 are fulfilled, and its application yields $X_t^A(s,a) = \ell_2^2(\eta_t^A(s,a), \eta_C(s,a)) \to 0$ and $X_t^B(s,a) = \ell_2^2(\eta_t^B(s,a), \eta_C(s,a)) \to 0$. Hence, also $\eta_t^A, \eta_t^B$ converge to $\eta_C$ with respect to $\bar{\ell}_2$. $\qquad\square$

