# OpenReview forum: "ADDQ: Adaptive distributional double Q-learning"
_ICML.cc/2025/Conference — ICML 2025 poster_

### Official Review · Reviewer_SxzX · 2025-03-05

**Overall Recommendation:** 3

**Summary:**

Based on double Q-learning, this paper proposes a set of theoretical and practical solutions to reduce the bias of Q value estimation. The paper conducts some experiments in Atari, mujoco and table environments. Some results have demonstrate their effectiveness.

## Update after rebuttal
During the rebuttal, the additional experiments greatly increase the persuasiveness of the paper and address my main concern, so I am very happy to update my evaluation/rate to weak accept.

**Claims And Evidence:**

The paper mentions that it combines Q-learning and Double Q-learning through direct linear weighting, proves its convergence, and conducts experiments in environments such as tables and Atari to prove its effectiveness.
In the experiment, the paper demonstrates the effectiveness of the proposed method with DDQN and  SAC. It is worth mentioning that it only considers a small number of selected tasks on the Atari task, which may not be convincing enough. It is also difficult to distinguish the performance from the baseline method on the Mujoco task, which may greatly reduce the persuasiveness of the paper.

**Essential References Not Discussed:**

I think the paper would at least benefit from comparison with some recent SOTA methods on Atari tasks, such as MEME, EfficientZero, etc.

**Experimental Designs Or Analyses:**

The overall design of the experimental part is reasonable, but at least on the Atari task, it lacks sufficient comprehensive analysis metrics, such as IQM, HNS, etc. In addition, the paper also lacks discussion and comparison on the core contribution (i.e., reducing the evaluation variance of Q) in the experiment.

**Methods And Evaluation Criteria:**

The metrics used in the experiment part of the paper are final score, final reward, etc. However, the metrics commonly used in Atari tasks such as IQM and human normliazed score are missing, making it difficult to evaluate the overall performance of the algorithm.

**Other Comments Or Suggestions:**

The paper may benefit from discussing the more  experimental and theoretical proof of the proposed method on the convergent behavior and convergence speed of the algorithm. In addition, more Atari tasks and more comprehensive evaluation metrics may enhance the persuasiveness of the paper.

**Other Strengths And Weaknesses:**

The core contribution of the paper is to propose a simple and easy-to-use solution to reduce Q value over-estimation.

The main disadvantage of the paper is that the experimental part does not prove the  effectiveness of the proposed method. First, the number of Atari tasks is small, and there are no comprehensive metrics such as IQM and Mean HNS. In addition, the proposed method on the Mujoco task is even far surpassed by the baseline method, which makes it difficult to prove the effectiveness of the algorithm.

**Questions For Authors:**

1. Have the authors considered showing more performance on Atari tasks, such as more tasks, more comprehensive metrics such as IQM[1], etc.
2. Could the author give a detailed introduction on how the proposed method improves the efficiency of the algorithm and why the performance on the mujoco task is not good?

[1] Deep Reinforcement Learning at the Edge of the Statistical Precipice

**Relation To Broader Scientific Literature:**

Bias control in Q-learning is an issue worth discussing. The paper also applies its improvements to existing methods such as SAC and DDQN, but due to the lack of sufficiently convincing experiments and the fact that in some experiments (such as mujoco)  the proposed method is even far below the baseline method,  the effectiveness of proposed method is thus less convincing.

**Theoretical Claims:**

The paper seems to have some theoretical motivations that are not well validated in the experiment. In addition, the paper may benefit from more specific discussion and proof of convergence.

---

> ### Author Rebuttal · Authors · 2025-03-31
>
> Dear reviewer, thank you very much for your careful reading and thougths on our article!
>
> **Please check the anonymous repository https://anonymous.4open.science/r/ADDQ-B776 for figures addressing some of your thoughts.**
>
> **Methods and evaluation criteria:**
> * The metrics that you claimed to be missing are given in the Appendix, please have another look at Appendix D.
>
> **Theoretical claims:**
> * The experiments are not designed to validate theoretical claims. In opposite, the theoretical results motivate the design of the ADDQ algorithm. Nonetheless, since it is an interesting remark, to answer your question we now provide simulations for the bandit MDP as well. Please check the linked repository above for the plots.
> * A full convergence proof is indeed provided in the tabular setup, in the RL literature there seems generally little hope to provide convergence proofs for function approximation with neural networks in non-trivial settings.
>
> **Relation to literature:**
> * We now added comparisons (see the link above for the plots) with several algorithms: ensemble bootstrapped Q-learning, maxmin Q-learning, and random ensemble double Q learing. Note that we only provide tabular comparisons as there is no benchmark implementation available for distributional EBQL/maxmin/REDQ.
> * Our presentation of MuJoCo was not the smartest move from an advertisement point of view, we tried to keep the highest scientific standards and compare the same algorithmic idea over all experiments. For MuJoCo both Q and DQ estimators are clearly inferior to the clipped estimator (in contrast to Atari). The reason is that DQ does not substract enough positive bias, the main finding of the TD3 paper. The purpose of the experiment was to show that cleverly combining Q and DQ estimators (ADDQ) beates both ingredients Q and DQ. Of course, the combination of both won't beat clipped. Here is the but: Our main idea (using sample variances to locally adjust overestimation) is not restricted to combine Q and DQ, but this is the most natural setting for a presentation. In the same way one could use a locally adaptive mixing of Q (or double Q) and clipped Q to avoid too extreme underestimation of the clipped estimator.
> * This paper contributes to a fundamental method in RL, Q-learning. The goal is not specifically to solve Atari or MuJoCo problems, these examples serve for illustration purposes as we cannot prove much in deep RL. Comparing to different methods can be interesting for general curiosity, but does not serve the main purpose of this article which is to improve QL/DQL with very little extra effort.
>
> **Weaknesses:**
> * Again, please have a quick look at Appendix D for the metrics and for MuJoCo to our comment above.
>
> **Comments:**
> * We agree, more theory on convergence rates, etc. would be very desirable. Our paper (just as the entire RL literature away from unrealistic simple situations) does not provide insight.
> * Environmental rules of our research team do not allow to run all Atari examples. In fact, there is not much to learn from more Atari examples and the computational/environmental effort is quite big. That's why we used the RLiable metrics to get as much information as possible from the runs. We know that QL and DQL work reasonably well, so there is no reason to expect ADDQ to perform weaker. It would be much more interesting to attack complicated environments on which (distributional) deep Q-learning fails. But here we run in the usual problem of our community to have too few very different benchmark examples.
>
> **Questions:**
> * Question 1: See Appendix D
> * Question 2: The performance on MuJoCo is not bad. As expected, the algorithm improves the two base algorithms using Q or double Q estimators. It cannot be expected to beat the clipped estimator and we included the example for completeness.

---

> > ### Comment · Reviewer_SxzX · 2025-04-05
> >
> > Thank you for your reply. The additional experiments to greatly increase the persuasiveness of the article and address my main concern, so I am very happy to update my evaluation.

---

### Official Review · Reviewer_idYs · 2025-03-06

**Overall Recommendation:** 3

**Summary:**

The authors propose ADDQ, a distributional reinforcement learning (DRL) algorithm which adaptively combines two RL algorithms to combat overestimation. Specifically, sample variances from DRL are used to adaptively balance the updating scheme of an algorithm with a tendency to overestimation (e.g. Q-Learning) with one with a tendency to under-estimation (e.g. Double Q-Learning). The authors provide theoretical analysis on a simple bandit MDP as well as an tabular illustrative practical analysis in a gridworld setting in comparison to distributional QL and distributional DQN, as well a comparison to C51 and QRSAC on Atari and MuJoCo benchmarks.

**Claims And Evidence:**

The authors claim that they “show theoretically how distributional RL helps the agent identify the need for overestimation control”. This claim, or rather the connection between the utilized sample variance (from distributional RL) and the actual overestimation is crucial for the method, as “overestimation regularization” is applied whenever sample variance is high for a given action wrt. the other actions in a state. Thus this connection should, in my opinion, be shown to the reader clearly and optimally illustrative.
The main body of text does contain Proposition 2.1 and 2.2 related to this claim.
Proposition 2.1 states that the overestimation (lower bound of bias) is connected, among other things, to σ and N.
The Proposition 2.2. states that, among other things, the sample variance is decreasing with number of updates N to a given action and is proportional to σ. Thus the sample variance and the overestimation are both connected to the variance σ, and via that to each other.

However, corresponding proof for Proposition 2.2., contained in A.2, makes the assumptions that  “all actions in s1 are explored N times before bootstrapping the estimates to s0” and that “the standard 1 /#visits -step-size schedule” is used, which appears rather unrealistic. Further, in practice, bootstrapping and corresponding overestimation is typically propagated over multiple states.

I suggest making this relation more apparent and clear, and perhaps include a practical analysis on the bandit task which shows the overestimation bias in relation to the sample variance and how ADDQ, by adapting β, alleviates the problem at hand.

**Essential References Not Discussed:**

In [1] an adaptive balance (ACC), using a balance controlling parameter β, between over- and under-estimation in TQC is proposed. Instead of deriving β from the sample variance as in this work, Monte-Carlo Rollouts area used. The strong similarity between ACC and ADDQ in my opinion requires discussing said similarity. Further, ACC is evaluated on MuJoCo which indicates that it could be also used for comparison in this work's experiments. Especially since a combination with TQC is listed as promising future work, it appears especially important to discuss and compare to [1].

[1] Dorka, N., Welschehold, T., Bödecker, J., & Burgard, W. (2022). Adaptively calibrated critic estimates for deep reinforcement learning. IEEE Robotics and Automation Letters, 8(2), 624-631.

Further, many related works in “Relation to Broader Scientific Literature” would be suitable for discussion, however I would not refer to those as essential.

**Experimental Designs Or Analyses:**

The experimental design and analysis is sound and I found no issues. However, I’d suggest adding more related algorithms for comparison.
There has been a large body of literature regarding uncertainty estimation in (offline) Reinforcement Learning, often using ensemble methods.
The authors state that “Ensemble methods are promising in theory (assuming independent ensembles) but more problematic for deep RL as storage problems force ensembles to be parametrized by the same neural network.”. However, there have been ensemble methods with hundreds of ensembles with distinct networks [1].
As it appears quite related, integrating an uncertainty-based method into the comparison could help giving an insight into the performance of the proposed method.
For example [2] tries to find a balance between over and underestimation using uncertainty estimates.
Further, Maxmin Q-learning [3], which the authors also mention in the introduction, combines the update rule of QL with DQL an should be a simple baseline to add.

[1] An, G., Moon, S., Kim, J. H., & Song, H. O. (2021). Uncertainty-based offline reinforcement learning with diversified q-ensemble. Advances in neural information processing systems, 34, 7436-7447.

[2] Li, S., Tang, Q., Pang, Y., Ma, X., & Wang, G. (2021). Balancing value underestimation and overestimation with realistic actor-critic. arXiv preprint arXiv:2110.09712.

[3] Lan, Q., Pan, Y., Fyshe, A., & White, M. Maxmin Q-learning: Controlling the Estimation Bias of Q-learning. In International Conference on Learning Representations.

**Methods And Evaluation Criteria:**

The proposed method and evaluation of said method appears reasonable and convincing, assuming that the sample variance can be used to identify the need for overestimation control.
However, I feel that a broader comparison, especially to uncertainty-driven RL, would paint a clearer picture with respect of to the algorithms performance. More on that in “Experimental Designs or Analyses”.

**Other Comments Or Suggestions:**

- Line 159, “[(“ brackets can be removed.
- Function class projection, introduced in line 128 (right) and Algorithm 1 could benefit from a proper introduction to be more self-contained.
- Line 187 left, could use an inline citation.
- Formatting issues with page 18, 24,26 in Appendix, Figures too large.
- In Figure 4 and 5, the color coding of the RLiable plots does not match the color coding of the evaluation progress plot. Setting the colors to match could ease readability.
- Perhaps figure 2. could benefit from a graph which shows what fraction of updates used which beta values over time, such that the adaptivity of the proposed method can be observed.

**Other Strengths And Weaknesses:**

Weaknesses as discussed above.

Strengths:
- Mostly self-contained
- General writing style
- Extensive experiments
- Use of RLiable library for probability of improvement

**Questions For Authors:**

Could you illustrate your intuition for the connection between sample variance and the overestimation bias, also for more realistic settings than the one used for the proofs? As is, it remains unclear whether sample variance is a good proxy for measuring overestimation, which is the main contribution of this paper. If this can be justified by more extensive analysis and experiments, I am willing to increase my score.

--
The rebuttal addressed most of my concerns and I increased my score accordingly.

**Relation To Broader Scientific Literature:**

For me, this paper is related to many works in the field of uncertainty-based Reinforcement Learning. While it is also mentioned in the introduction that “overestimation should be addressed particularly in state-action regions with high uncertainty”, the authors disregard the field of uncertainty-based RL and instead only put a focus on ensemble methods.
In the field of uncertainty-based RL many works try to reduce Q-values for uncertain states [1, 2, 3, 4], which is conceptually similar to “putting more weight on DQN” for high variance samples as proposed here.

[1] Wu, Yue, et al. "Uncertainty Weighted Actor-Critic for Offline Reinforcement Learning." International Conference on Machine Learning. PMLR, 2021.

[2] Ghasemipour, K., Gu, S. S., & Nachum, O. (2022). Why so pessimistic? estimating uncertainties for offline rl through ensembles, and why their independence matters. Advances in Neural Information Processing Systems, 35, 18267-18281.

[3] Bai, C., Wang, L., Yang, Z., Deng, Z., Garg, A., Liu, P., & Wang, Z. (2022). Pessimistic bootstrapping for uncertainty-driven offline reinforcement learning. arXiv preprint arXiv:2202.11566.

[4] Li, S., Tang, Q., Pang, Y., Ma, X., & Wang, G. (2021). Balancing value underestimation and overestimation with realistic actor-critic. arXiv preprint arXiv:2110.09712.

**Theoretical Claims:**

I had a look at the proofs provided in the appendix but did not check all of them.

---

> ### Author Rebuttal · Authors · 2025-03-31
>
> Dear reviewer, thank you very much for your careful reading and thougths on our article!
>
> **Please check the anonymous repository https://anonymous.4open.science/r/ADDQ-B776 for figures addressing some of your thoughts.**
>
> **Bandit example:**
> * We agree with your summary about the main contribution of the article and the discussion of the bandit example. We would like to stress that this simplified theoretical contribution is  more reasonable than most of what appeared in the past DQL literature. In past theoretical contributions the max estimator is typically studied for IID random variables or for "chain MDPs" without any actions. Also the step-sizes are not very unrealistic even though practitioners like to use other schedules. The Robbins-Monroe conditions require asynchronous (that's why number of visits) step-sizes of the order $1/n^p$, for $p\in (1/2,1]$ so choosing $p=1$ is not particularly unrealistic. Our Lemma A.1 sheds light on the general situation - the simplified Q-learning analysed is a lower bound mechanism, the general situation is even worse. What is behind the Gaussian estimate is the following thought: The overestimation error of the max is governed by the worst single overestimation, thus, distributing the exploration evenly makes equally good estimators or, equivalently, minimises the worst single overestimation. We actually believe, and we are currently working on it, that the bandit MDP computation combined with Lemma A.1 can also shed light on the overestimation of general  MDPs. A backwards induction (dynamic programming) from terminal states for stochastic shortest paths problems (or the random geometric time-horizon for discounted MDPs) should allow to derive lower bounds for the overestimation also for general Q-learning exploration on general MDPs. It might be interesting to highlight that our simplified computation shows clearly where the difficulties of computations in distributional Q-learning come from. The update mechanism naturally combines sums and maxima of random variables, so it lives in the intersection of extremal value theory and the central limit theorem. Since there are no distributions that are sum-stable and max-stable at the same time, there is no shortcut in exact probabilistic computations.
> * We now performed a simulation of the bandit example to make our theoretical point clearer.
>
> **Experimental design:**
> * We now compare ADDQ to ensemble bootstrapped Q-learning, maxmin Q-learning, and random ensemble double Q learning. The difficulty of the example (very different local randomness) shows clearly the local overestimation control of ADDQ over all other methods. Note that we only provide tabular comparisons on the delicate grid world example as there is no benchmark implementation available for distributional EBQL/maxmin/REDQ.
>
>
> We also plotted the relative sample variances (and, thus, the choice of $\beta$) for the non-trivial grid world example in order to make the point clear: our algorithm automatically spots the problematic state-action pairs to mitigate the overestimation.
>
> **References:**
> * Thank you very much for providing additional literature, which we will certainly include in a revised version! While we believe that our approach is not directly related to approaches using uncertainty Bellmann, we agree that ACC is much closer in spirit. This is a very nice paper, thanks for sharing. As you point out, the approach resembles of what we sketched under future research for TQC. The way ACC is formulated (using the replay buffer), ACC does not control overestimation locally on state-action level but globally (averaging over state-action pairs from the replay buffer). This way the algorithm would struggle for instance on our delicate grid world example just as much as maxmin and ensemble methods. In the tabular situation this might be adjusted, with function approximation probably not. For ADDQ we use that the distributions (thus variances) are learned on state-action level.
>
> **Other comments:**
> * We will improve your minor comments, thanks!
> * We included plots to visualize the choices made by $\beta$, thanks!
>
> **Questions:**
> Thanks for reading carefull, yes, this can be seen as the core contribution. From statistical theory the observation is somewhat obvious (large variance means large sample variance, means large overestimation bias for max estimator) and this was essentially worked out for the bandit MDP and then translated in $\beta$ for ADDQ. Our intuition then uses dynamic programming (backwards induction) to bootstrap the idea up the decision tree. We added plots for our delicate grid world example for a state-action pair facing towards the stochastic region. The plots show that relative sample variance and overestimation are both large and decrease together (very slowly for Q-learning, much faster for ADDQ). In contrast, for less vulnerable state-action pairs both overestimation and relative sample variances are much smaller.

---

> > ### Comment · Reviewer_idYs · 2025-04-02
> >
> > The rebuttal addressed most of my concerns and I increased my score accordingly.

---

### Official Review · Reviewer_C9ok · 2025-03-13

**Overall Recommendation:** 2

**Summary:**

This paper proposes ADDQ, an adaptive distributional double Q-learning method that mitigates Q-value overestimation bias by locally adjusting update weights based on distributional uncertainty estimates. Built upon distributional RL frameworks (e.g., C51, QRDQN), ADDQ dynamically combines Q-learning and double Q-learning updates using sample variance information from return distributions. Theoretical analysis in a tabular bandit MDP quantifies overestimation bounds, and experiments across tabular, Atari, and MuJoCo environments demonstrate improved stability and performance compared to baseline methods.

**Claims And Evidence:**

Claims:
- Overestimation bias in Q-learning can be mitigated by locally adapting updates using distributional variance.
- ADDQ integrates seamlessly into existing distributional RL algorithms with minimal code changes.
- The method converges theoretically and outperforms QL/DQL in stochastic and high-uncertainty environments.

Evidence:
- Proposition 2.1 derives a lower bound for QL overestimation in a bandit MDP, linking bias to reward variance and sample size.
- Theorem 3.1 proves ADDQ’s convergence under Robbins-Monro conditions.
- Experiments on grid worlds, Atari, and MuJoCo show ADDQ reduces bias and achieves higher scores than QL, DQL, and clipped variants (Figures 2, 4-5).

**Essential References Not Discussed:**

NA

**Experimental Designs Or Analyses:**

Strength:
- Comprehensive evaluation across tabular, Atari, and MuJoCo benchmarks.
- Inclusion of RLiable metrics (e.g., interquartile mean, probability of improvement) enhances statistical rigor.

Weakness:
- MuJoCo experiments show limited gains (even worse) compared to clipped QRSAC , but this is not deeply analyzed.
- Ablation studies for $\beta$ thresholds (Section B.2) are preliminary; sensitivity to hyperparameters is unclear.

**Methods And Evaluation Criteria:**

Methods:
- Distributional RL: Uses return distribution variances to identify uncertain state-action pairs.
- Adaptive Weighting: Adjusts interpolation weights ($\beta$) between QL and DQL updates based on relative sample variances.
- Algorithm Integration: Modifies C51 and QRDQN with dual networks and adaptive targets.

Evaluation Criteria:
- Bias Reduction: Measured via Q-value deviations in tabular settings (Figure 2).
- Performance: Normalized scores and probability of improvement (RLiable plots) across 10 Atari and 5 MuJoCo environments.
- Stability: Comparison of failure rates and learning curve variances.

**Other Comments Or Suggestions:**

NA

**Other Strengths And Weaknesses:**

NA

**Questions For Authors:**

- I suggest to add detailed  discussion on the choice of $\beta$ considering its essential role in this paper , the threshold to determine $\beta$ seems careless and randomly. You must tell readers how to set the threshold by some criterion instead of using some fixed value.
- I believe the $b^{A/B}(s,a)$ in left panel of line 247 is aligned with that in left panel of line 234, but you also  have claimed $b^{A/B}(s,a)$ in left panel line 239, which would cause a confusion for readers.
- I wonder the $\nu \leftarrow \beta\eta^{B}(s^{\prime,a^*})+(1-\beta)\eta^{A}(s^{\prime},a^*)$ in Algorithm 2 is correct? It seems not akin to $\beta$ defined in equation (1).
- It is not clear how $(\beta_t^A)_{t\in\mathbb{N}}$ and its counterpart
 in Therorem 3.1 are defined and what is the relationship between them and  $\beta$ in Algorithm 2. I also wonder how to guarantee  the two $\beta$ would meet in limit behavior in your algorithm.
- Compile error occurs in line 1537.

**Relation To Broader Scientific Literature:**

The main results of this paper could contribute to applications and methodologies in robust RL learning and improve the performance of learning.

**Theoretical Claims:**

- Proposition 2.1 provides a tight lower bound for QL overestimation in Gaussian bandits, highlighting the role of variance and action count. While insightful, the analysis assumes cyclic exploration and ignores function approximation.
- Theorem 3.1 guarantees convergence under symmetric $\beta$ schedules. The proof leverages stochastic approximation theory but does not address deep RL settings with neural networks.

---

> ### Author Rebuttal · Authors · 2025-03-31
>
> Dear reviewer, thank you very much for your careful reading and thoughts on our article!
>
> **Please check the anonymous repository https://anonymous.4open.science/r/ADDQ-B776 for figures addressing some of your thoughts.**
>
> **Weaknesses:**
>
> * **Ablation study**: We included a more comprehensive ablation study for $\beta$, plotting many choices of $\beta$ showing that the choice is relatively irrelevant. We also compare ADDQ to ensemble bootstrapped Q-learning, maxmin Q-learning, random ensemble double Q learning. We hope that is more convincing. Note that we only provide tabular comparisons as there is no benchmark implementation available for distributional EBQL/maxmin/REDQ.
>
> * **MuJoCo**: Our presentation of MuJoCo was not the smartest move from an advertisement point of view, we tried to keep the highest scientific standards and compare the same algorithmic idea over all experiments. For MuJoCo both Q and DQ estimators are clearly inferior to the clipped estimator (in contrast to Atari). The reason is that DQ does not substract enough positive bias, the main finding of the TD3 paper. The purpose of the experiment was to show that cleverly combining Q and DQ estimators (ADDQ) beates both ingredients Q and DQ. Of course, the combination of both won't beat clipped. Here is the but: Our main idea (using sample variances to locally adjust overestimation) is not restricted to combine Q and DQ, but this is the most natural setting for a presentation. In the same way one could use a locally adaptive mixing of Q (or double Q) and clipped Q to avoid too extreme underestimation of the clipped estimator.
>
> **Questions:**
>
> * You are totally right that the suggested **choice of $\beta$** is somewhat arbitrary. There are plenty of stochastic approximation algorithms that converge to the optimal Q-matrix. In some sense one could argue that all of them (double Q, clipped, maxmin, averaged, weighted, ADDQ, etc.) are somewhat arbitrary choices from the set of all converging algorithms, none is the result of a rigorous derivation. Only Q-learning itself might be seen natural as it is the direct (but inefficient) stochastic counterpart of value iteration. There are no convincing theoretical arguments in the literature, and also no variants that are equally convincing over different environments. Clipped is perhaps a good example, it performs well in combination with SAC on MuJoCo but not on Atari - Q/DQL the opposite. For a new environment, the RL researcher has no other choice but to compare different variants plus some educated guessing. We have no better answer than saying that we add a new family of animals to the zoo of algorithms which is more flexible and combines the advantages of two other algorithms. The form of $\beta$ we propose is the natural choice based on our theoretical observation, only the choice of constants (hyper parameters) is somewhat artificial. We added a much more extensive ablation study (see the repository) to show that the choice off hyperparameters in $\beta$ is actually pretty harmless. Interestingly enough, the same choice of $\beta$ improves QL/DQL on diverse settings as tabular, Atari, MuJoCo.
>
> * **$b^{A/B}(s,a)$ notation**: Thanks for your comment, we will improve the notation to make readability easier.
>
> * **$\nu$ from Algorithm 2**: Thanks for your careful reading! This is a typo, and was realised only one hour after submission. The implementation is correct.
>
> * **Theorem 3.1**: The theorem holds for **arbitrary** such sequences, we will add the word "arbitrary". You are right about your question, it is unclear how to check equality in the limit. The easiest way (and this is what we do) is to chose $\beta^A_t=\beta^B_t$ for all $t$ so the condition is trivially satisfied.
>
> * **Compilation error**: Thanks for spotting!

---

### Official Review · Reviewer_amjr · 2025-03-14

**Overall Recommendation:** 3

**Summary:**

The paper introduces ADDQ (Adaptive Distributional Double Q-learning), a novel reinforcement learning (RL) algorithm that addresses the overestimation bias in Q-learning by leveraging distributional reinforcement learning (DRL). The key claim is that ADDQ provides a flexible and computationally efficient way to mitigate overestimation, improving learning stability and efficiency.

**Claims And Evidence:**

The paper makes several strong claims regarding ADDQ's advantages:
- Reduction of Overestimation Bias – Supported by a theoretical analysis using probability bounds.
- Better Stability than QL and DQL – Demonstrated through experiments in various environments.
- Improved Sample Efficiency – Shown via empirical comparisons in Atari and MuJoCo environments.

While the theoretical analysis is compelling, some claims (e.g., optimality of the chosen weighting function) could be more rigorously justified with additional ablation studies.

**Essential References Not Discussed:**

While the paper covers the key references, it could benefit from discussing:
- More recent advances in Bayesian RL, which also address uncertainty estimation.
- Work on uncertainty-aware RL methods beyond variance-based heuristics, such as Bootstrapped DQN (Osband et al., 2016).

**Ethical Review Concerns:**

Lack of *Impact Statements* section.

**Ethical Review Flag:**

Flag this paper for an ethics review.

**Ethics Expertise Needed:**

["Other expertise"]

**Experimental Designs Or Analyses:**

Strengths:
- Baseline comparisons include standard QL, DQL, and clipped QL, ensuring a fair evaluation.
- RLiable evaluation metrics (probability of improvement) provide a robust statistical comparison.

Weaknesses:
- No ablation studies on the choice of β and its sensitivity to different environments.
- No analysis of computational overhead (Does ADDQ introduce additional costs?).

**Methods And Evaluation Criteria:**

The methodology is well-designed and aligns with the problem at hand:
- The use of sample variance as an indicator of uncertainty is well-motivated.
- The local weighting mechanism between QL and DQL is clearly explained.
- The evaluation benchmarks (Atari, MuJoCo) are appropriate for demonstrating generalization.

However, the choice of hyperparameters for β (the weighting factor) is somewhat heuristic. It would be valuable to analyze different settings of β to ensure robustness across environments.

**Other Comments Or Suggestions:**

- Provide more discussion on practical implementation (e.g., how easy is it to integrate ADDQ into existing RL libraries?).
- Include an analysis of computational efficiency.

**Other Strengths And Weaknesses:**

Strengths:
- Theoretical grounding: Provides mathematical insight into overestimation bias.
- Practical implementation: Easily adaptable to existing DRL frameworks.
- Empirical validation: Strong benchmark comparisons with standard methods.

Weaknesses:
- Limited ablation studies: The choice of β is not fully justified.
- No computational cost analysis: Would ADDQ increase training time or memory usage?
- Limited discussion on failure cases: When does ADDQ not work well?

**Questions For Authors:**

- How does ADDQ compare in terms of computational cost? Since it requires computing sample variances, does it introduce a significant overhead?
- Why was β chosen heuristically rather than learned adaptively? Would an adaptive β (e.g., learned via meta-learning) improve performance?
- How does ADDQ perform with function approximation errors? The theoretical results focus on tabular settings—how well do they generalize to deep RL?
- Could ADDQ be combined with other overestimation reduction methods? For instance, can it be integrated with ensemble-based RL (e.g., Averaged-DQN)?

**Relation To Broader Scientific Literature:**

The paper is well-positioned in the reinforcement learning literature. The paper clearly distinguishes ADDQ from previous approaches.

**Theoretical Claims:**

The paper provides proofs of convergence for the ADDQ algorithm in tabular settings. The theoretical analysis is rigorous, but the following concerns arise:
- The impact of function approximation (i.e., deep RL settings) is not fully addressed in the theoretical framework.
- Some assumptions (e.g., independence of updates in DQL) may not hold in practical scenarios with deep learning.

A discussion on these limitations and potential extensions would strengthen the theoretical contribution.

---

> ### Author Rebuttal · Authors · 2025-03-31
>
> Dear reviewer, thank you very much for your careful reading and thougths on our article!
>
> **Please check the anonymous repository https://anonymous.4open.science/r/ADDQ-B776 for figures addressing some of your thoughts.**
>
> * You are totally right that the suggested **choice of $\beta$** is somewhat arbitrary. There are plenty of stochastic approximation algorithms that converge to the optimal Q-matrix. In some sense one could argue that all of them (double Q, clipped, maxmin, averaged, weighted, ADDQ, etc.) are somewhat arbitrary choices from the set of all converging algorithms, none is the result of a rigorous derivation. Only Q-learning itself might be seen natural as it is the direct (but inefficient) stochastic counterpart of value iteration. There are no convincing theoretical arguments in the literature, and also no variants that are equally convincing over different environments. Clipped is perhaps a good example, it performs well in combination with SAC on MuJoCo but not on Atari - Q/DQL the opposite. For a new environment, the RL researcher has no other choice but to compare different variants plus some educated guessing. We have no better answer than saying that we add a new family of animals to the zoo of algorithms which is more flexible and combines the advantages of two other algorithms. The form of $\beta$ we propose is the natural choice based on our theoretical observation, only the choice of constants (hyper parameters) is somewhat artificial. We added a much more extensive ablation study (see the repository) to show that the choice of hyperparameters in $\beta$ is actually pretty harmless. Interestingly enough, the same choice of $\beta$ improves QL/DQL on diverse settings as tabular, Atari, MuJoCo.
>
> * **Hyperparameter tuning**: To ensure fair comparison, we did not tune any hyperparameter and stick to the choices from stable baselines 3. We did not even try to optimise the choices of $\beta$ and used our first choice for all experiments. Usually researchers tune their methods before publication, but our entire point is that the method is very robust. It is a harmless trick to improve distributional algorithms without effort.
>
> * **Deep Theory**: We would love to provide theory in the deep setting. But to be honest, that seems pretty hope less. Not only for our article, but we are not aware of any convincing result in the RL literature.
>
> * **Computational overhead**: ADDQ has no computational overhead compared to double distributional QL (the runtime is almost identical). The only difference is to compute $\beta$ (a small finite sum), which compared to evaluating large NN is almost nothing. The computational effort mainly comes from distributional deep Q-learning, which is significantly more expansive (but more sample efficient) than ordinarly deep Q-learning.
>
> * **Failure cases**: We did not encounter failure cases (even though failure might be hard to judge). With the choice of $\beta$, ADDQ works stably if either Q or double Q works reasonably well. In contrast, in situations with diverse randomness as our grid world example, ADDQ works much better than Q and double Q. One could interprete the MuJoCo examples as a failure. In this case clipping is much more effective than both Q and double Q (the negative bias of double Q is not enough) so a combination of Q and double Q has no chance to beat clipping. MuJoCo is more a failure of Q and double Q (ADDQ still improves both!) compared to clipping. From an advertisement perspective it might have been smarter to locally combine for MuJoCo the Q and the clipped estimator (algorithmically this is almost identical) but the current presentation is scientifically more honest. After all, in this paper we wanted to show how to improve Q and DQ by a combination of both.
>
> * **Implementation**: The implementation overhead is adding one line and changing two lines of code to existing (here: stable baselines 3) distributional code: Compute $\beta$ from the distributions (i.e. compute a finite sum) and then change the double Q updates. That's it.
>
> * **Integration in other algorithms**: Yes, the main idea can be combined with other approaches in which an algorithm has a parameter that steers the over/underestimation *and* the parameter can be adapted on the fly. In that case we would suggest to locally change the parameter according to the sample variance. Here are two examples: (i) In TQC a number of top atoms is truncated. The number is a hyperparameter and must be chosen for every environment. We would suggest to chose the number of truncated atoms locally according to the sample variance. (ii) In randomized ensemble DQL we would suggest to make the number of chosen ensemble members for the updates locally dependent on the sample variance.

---

### Decision · Program_Chairs · 2025-05-01

**Decision:**

Accept (poster)

**Comment:**

The reviews alone would make this a very borderline paper, but actually evaluating the paper directly, it has strengths and weaknesses that largely balance out. Focusing on the essentials, the work motivates an insight into addressing over-estimation bias, provides a method to do so, and demonstrates that it is effective. However, when it comes to how well this translates into policy performance the story is much more mixed. Here the results are certainly not bad, but nor are they especially exceptional. That said, they are thorough and rigorous, and as is the overall treatment of the research problem itself, both of which should count for something.

So, while I think this is a paper I would be more supportive of at a different venue, I think it on balance teaches me more about the nature of the RL problem than a fair swath of the accepted RL papers at ICML.